# Biological Activities of Some New Secondary Metabolites Isolated from Endophytic Fungi: A Review Study

**DOI:** 10.3390/ijms22020959

**Published:** 2021-01-19

**Authors:** Ruihong Zheng, Shoujie Li, Xuan Zhang, Changqi Zhao

**Affiliations:** Gene Engineering and Biotechnology Beijing Key Laboratory, College of Life Science, Beijing Normal University, 19 XinjiekouWai Avenue, Beijing 100875, China; 201821200033@mail.bnu.edu.cn (R.Z.); 201831200027@mail.bnu.edu.cn (S.L.); 201631200003@mail.bnu.edu.cn (X.Z.)

**Keywords:** new secondary metabolites, endophytic fungi, structural feature, biological activity

## Abstract

Secondary metabolites isolated from plant endophytic fungi have been getting more and more attention. Some secondary metabolites exhibit high biological activities, hence, they have potential to be used for promising lead compounds in drug discovery. In this review, a total of 134 journal articles (from 2017 to 2019) were reviewed and the chemical structures of 449 new metabolites, including polyketides, terpenoids, steroids and so on, were summarized. Besides, various biological activities and structure-activity relationship of some compounds were aslo described.

## 1. Introduction

During the growth of microorganisms, some secondary metabolites biologically active are produced to make their lives better. Using chemical and biological methods, Elshafie et al. displayed that the cell-free culture filtrate of *Burkholderia gladioli* pv. *agaricicola* (Bga) Yabuuchi has a promising antibacterial activity against the two microorganisms B. megaterium and *E. coli* [1]. Camele et al. reported that the tested isolate of an endophytic bacterium *Bacillus mojavensis* showed antagonistic bacterial and fungal activities against several strains as well as biofilm formation ability [2]. Endophytes refer to the microorganisms that exist in various organs, tissues or intercellular space of plants, while the host plants generally do not show any symptoms of infection. Generally speaking, endophytes include endophytic fungi, endophytic bacterium and endophytic actinomycetes [3]. As a very important microbial resource, endophytes exist widely in nature. It is ubiquitous in various terrestrial and aquatic plants. Endophytes have been isolated from bryophytes, ferns, pteridophytes, hornworts, herbaceous plants and various woody plants. The region also ranges from tropical to arctic, from natural wild to agricultural industry ecosystem [4]. They have unique physiological and metabolic mechanisms, which enable them to adapt to the special environment inside plants, and at the same time, they can encode a variety of bioactive substances. In addition, endophytes coevolved with the host plants for a long time to produce some metabolic substances similar or identical to the host plants with medicinal value [5]. Some endophytes can even assist the host of medicinal plants to synthesize effective active compounds, the ground-breaking discovery provides a new method to produce the effective compounds which have similar effects with natural medicines isolated from plant tissues directly. At the same time, it has solved the problem of resource shortage and ecological destruction caused by slow growth of some natural plants and large amount of artificial exploitation [3]. The more beneficial thing is that some of them are environmentally friendly. Elshafie et al. have studied the fungus *Trichoderma harzianum* strain T22 (Th-T22) and indicated that Th-T22 showed significant mycoremediation ability in diesel-contaminated sand, suggesting that it can be used as a bioremediation agent for diesel spills in polluted sites [6]. Among the common endophytes, the endophytic fungi are most often isolated [4]. The first endophytic fungus was isolated from Perennial ryegrass (*Loliumtum eletum*) seeds by Vogle in 1898 [7]. Up to now, the study on endophytic fungi has a long history of more than 100 years, but the research on endophytic fungi of medicinal plants has not been formally carried out until the last 30 years, which has gradually attracted the attention of domestic and foreign scholars.

The multiformity of endophytes enable they can produce a variety of secondary metabolites. In recent years, the metabolites isolated from the endophytic fungi include alkaloids, steroids, terpenes, anthraquinones, cyclic peptides, flavonoids commonly [5]. Some secondary metabolites exhibit high biological activities. The antitumor, antibacterial, anti-inflammatory, antiviral, antifungal and other compounds have been produced by different endophytic fungi. Therefore, the chemical variety of secondary metabolites produced by endophytic fungi has advantage for new drug development [8].

In this review, 449 new secondary metabolites, together with their chemical structures and biological activities were summarized. The structure-activity relationships and absolute configureuration of some compounds have also been described. Among all new compounds, terpenoids account for the largest proportion (75%), followed by polyketones (36%). The proportion of different types of compounds in all new compounds is shown in Figure 1. These new compounds were isolated from various fungi associated with different tissues from different plants. As a result, their structures varied a lot, which leads to their multitudinous biological activities. In addition to common antimicrobial activity and anti-tumor activity, some compounds also showed anti-enzyme activity and inhibition of biofilm formation, inhibition of phytoplankton growth, and so on.

## 2. New Metabolites Isolated from Plant Endophytes

### 2.1. Terpenoids

#### 2.1.1. Sesquiterpenoids and Their Derivatives

Five new polyketide-terpene hybrid metabolites **1**–**5** (Figure 2) with highly functionalized groups, were isolated from the endolichenic fungus *Pestalotiopsis* sp. [9]. Co-cultivation of mangrove endophytic fungus *Trichoderma* sp. 307 and aquatic pathogenic bacterium *Acinetobacter johnsonii* B2 led to the production of two new furan-type isoeremophilane sesquiterpenes, microsphaeropsisin B **6** and microsphaeropsisin C **7** (Figure 2). Their absolute configureuration were assigned as 4S, 5R, 7R, 8S, 11S and 4R, 5R, 7R, 8S [10]. Following cultivation on rice medium, a new sesquiterpene, atrichodermone C **8** (Figure 2), was isolated from an endophytic fungal strain named *Trichoderma atroviride* which was isolated from the bulb of *Lycoris radiate* [11]. There is an endophytic fungus *Pestalotiopsis* sp. which was obtained from fruits of *Drepanocarpus lunatus* (Fabaceae). Co-culture of this fungus with *Bacillus subtilis* afforded two new sesquiterpenoids pestabacillins A **9** (Figure 2) and pestabacillins B **10** (Figure 2) [12]. Two new sesquiterpene-epoxycyclohexenone conjugates, nectrianolins A **11** (Figure 2) and nectrianolins B **12** (Figure 2), together with a sesquiterpene, nectrianolin C **13** (Figure 2), were isolated from the brown rice culture of *Nectria pseudotrichia* 120-1NP, an endophytic fungus isolated from *Gliricidia sepium*. It is of particular interest that **11** and **12** have a rearranged monocyclofarnesyl skeleton (which is uncommon to sesquiterpene-epoxycyclohexane conjugates) instead of a bicyclofarnesyl skeleton which is present in macrophorins, neomacrophorins, myrothecols, and craterellins [13]. It was found that endophytic *Nigrospora oryzae* stimulated the production of a new tremulane sesquiterpene nigrosirpexin A **14** (Figure 2) from *Irpex lacteus* [14]. Two novel sesquiterpenoids with an unprecedented tricyclo[4,4,2,1]hendecane scaffold, namely emericellins A **15** (Figure 2) and emericellins B **16** (Figure 2) representing a new skeleton, were isolated from the liquid cultures of an endophytic fungus *Emericella* sp. XL 029 associated with the leaves of *Panax notoginseng* [15]. Two trichothecene sesquiterpenoids, trichothecrotocins A **17** (Figure 2) and trichothecrotocins B **18** (Figure 2), and a pair of merosesquiterpenoid racemates, (+)-trichothecrotocin C **19** (Figure 2) and (−)-trichothecrotocin C **20** (Figure 2), were obtained from potato endophytic fungus *Trichothecium crotocinigenum* by bioguided isolation. Compounds **17** and **18** are trichothecenes possessing new ring systems. Compounds **19** and **20** possess novel 6/6−5/5/5 fused ring system [16]. Chemical investigation on the solid rice culture of *Trichoderma atroviride* S361, an endophyte isolated from *Cephalotaxus fortunei*, has afforded a new cyclohexenone sesquiterpenoid, trichodermadione B **21** (Figure 2) [17]. Seven new phenolic bisabolane sesquiterpenoids, (7R,10S)-7,10-epoxysydonic acid **22** (Figure 2), (7S,10S)-7,10-epoxysydonic acid **23** (Figure 2), (7R,11S)-7,12-epoxysydonic acid **24** (Figure 2), (7S,11S)-7,12-epoxysydonic acid **25** (Figure 2), 7-deoxy-7,14-didehydro-12-hydroxysydonic acid **26** (Figure 2), (Z)-7-deoxy-7,8-didehydro-12-hydroxysydonic acid **27** (Figure 2), and (E)-7-deoxy-7,8-didehydro-12-hydroxysydonic acid **28** (Figure 2), were obtained from the culture of an endophytic fungus *Aspergillus* sp. xy02 isolated from the leaves of a Thai mangrove *Xylocarpus moluccensis* [18]. Pestalustaines A **29** (Figure 2), one unique sesquiterpene possessing an unusual 5/6/7-fused tricyclic ring system was isolated from the plant-derived *Pestalotiopsis adusta* [19]. A new acorane sesquiterpene, 3β-hydroxy-β-acorenol **30** (Figure 2), possesses an acorane framework was separated from the extract of the green Chinese onion derived fungus *Fusarium proliferatum* AF-04 [20]. An examination of the endophytic fungus *Trichoderma asperellum* A-YMD-9-2 obtained from the marine red alga *Gracilaria verrucosa* led to the isolation of seven new chromanoid norbisabolane derivatives, trichobisabolins I–L **31**–**34** (Figure 2) and trichaspsides C–E **35**–**37** (Figure 2). The discovery of compounds **31**–**37** greatly diversifies the structures of norbisabolane sesquiterpenes [21]. Oxytropiols A–J **38**–**47** (Figure 2), ten undescribed highly oxygenated guaiane-type sesquiterpenoids, were isolated from the locoweed endophytic fungus *Alternaria oxytropis* [22]. Studies on the bioactive extract of mangrove endophytic fungus *Pleosporales* sp. SK7 led to the isolation of an abscisic acid-type sesquiterpene **48** (Figure 2), named (10S, 2Z)-3-methyl-5-(2,6,6-trimethyl-4-oxocyclohex-2-enyl)pent-2-enoicacid [23]. One new tremulane sesquiterpene, irpexlacte A **49** (Figure 2), was isolated from the endophytic fungus *Irpex lacteus* DR10-1 waterlogging tolerant plant *Distylium chinense* [24]. Trichocadinins B−G **50**–**55** (Figure 2), six new cadinane-type sesquiterpene derivatives, each with C-14 carboxyl functionality, were isolated from the culture extract of *Trichoderma virens* QA-8, an endophytic fungus obtained from the fresh inner tissue of the medicinal plant *Artemisia argyi* [25]. Chemical investigation of the EtOAc extract of the plant-associated fungus *Alternaria alternate* in rice culture led to the isolation of a new sesquiterpene (1R,5R,6R,7R,10S)-1,6-Dihroxyeudesm-4(15)-ene **56** (Figure 2) [26]. An investigation of a co-culture of the *Armillaria* sp. and endophytic fungus *Epicoccum* sp. YUD17002 associated with *Gastrodia elata* led to the isolation of five protoilludane-type sesquiterpenes named epicoterpenes A–E **57**–**61** (Figure 2). Compound **60** was the first example of an ent-protoilludane sesquiterpenoid scaffold bearing a five-membered lactone. Notably, none of the new compounds were produced by either of the two fungi when cultured alone under the same conditions [27]. A new sesquiterpene lactone, namely colletotrin **62** (Figure 2), was obtained from a rice culture of *Colletotrichum gloeosporioides*, an endophytic fungus isolated from the stem bark of Cameroonian medicinal plant *Trichilia monadelpha* (Meliaceae) [28]. Purpurolide A **63** (Figure 2), an unprecedent sesquiterpene lactone with a rarely encountered 5/5/5 spirocyclic skeleton, along with two new 6/4/5/5 tetracyclic sesquiterpene lactones purpurolide B and C **64**–**65** (Figure 2), were isolated from the cultures of the endophytic fungus *Penicillium purpurogenum* IMM003 [29]. Bioassay-guided fractionation of the crude extract of fermentation broth of one symbiotic strain *Fusarium oxysporum* ZZP-R1 derived from coastal plant *Rumex madaio Makino*, one traditional Chinese medicine used as a treatment of inflammation and toxication, yielded one novel compound, fusariumins D **66** (Figure 2). Chemical structure of **66** was determined as a sesquiterpene ester with a conjugated triene and an unusual oxetene ring by a combination of spectroscopic methods [30].

#### 2.1.2. Diterpenoids

One new cleistanthane-type diterpene zythiostromic acid C **67** (Figure 3), which structure was assigned as 3α,5α,7β,8β-tetrahydroxycleistanth-13(17),15-dien-18-oic acid, was isolated from the brown rice culture of *Nectria pseudotrichia* 120-1NP [31]. A fungal strain, *Drechmeria* sp., was isolated from the root of *Panax notoginseng*. Totally, seven new indole diterpenoids, drechmerins A–G **68**–**74** (Figure 3), were isolated from the fermentation broth of *Drechmeria* sp. [32]. A novel 1(2), 2(18)-diseco indole diterpenoid, drechmerin H **75** (Figure 3), was isolated from the fermentation broth of *Drechmeria* sp. together with a new indole diterpenoid, 2′-epi terpendole A **76** (Figure 3) [33]. An endophytic fungus, *Neosartorya fifischeri* JS0553, was isolated from *G. littoralis* plant. From the fungus, a new meroditerpenoid named sartorypyrone E **77** (Figure 3) was isolated [34]. Two new oxoindolo diterpene epimers, anthcolorin G **78** (Figure 3) and anthcolorin H **79** (Figure 3), isolated for the first time from a natural source, were isolated from the solid rice culture of the endophytic fungus *Aspergillus versicolor* [35]. A new isopimarane derivative which was named as xylaroisopimaranin A **80** (Figure 3) and the absolute configureurations was determined as 4S, 5R, 9R, 10R, 13R and 14S, was isolated from the plant endophytic fungus *Xylaralyce* sp. (HM-1) [36]. The endolichenic fungus *Apiospora montagnei* isolated from the lichen *Cladonia* sp. was cultured on solid rice medium, yielding a new diterpenoid libertellenone L **81** (Figure 3), compound **81** represented the first example of 6,7-seco-libertellenone derivative [37].

#### 2.1.3. Other Terpenoids

Eleven new ophiobolin-type sesterterpenoids, asperophiobolins A−K **82**–**92** (Figure 4), were isolated from the cultures of the mangrove endophytic fungus *Aspergillus* sp. ZJ-68. Asperophiobolins A–D (**82**−**85**) represented the first examples possessing a five-membered lactam unit between C-5 and C-21 in ophiobolin derivatives. The absolute configureuration of compands were defined as (2S,3R,5S,6R,11R,14R,15S) (**82**–**84**), (2S,3R,5S,6R,10S,11R,14R,15S) (**85**), (2S,6S,10S,11R,14R,15S,18R) (**87**), (2S,6R,10S,11R,14R,15S,18R) (**88**), (2S,6S,10S,11R,14R,15S,18S) (**89**), (2S,6R,10S,11R,14R,15S,18S) (**90**),(2S,3R,6R,10S,11R,14R,15S,18S) (**91**), (2R,3R,5R,6R,10S,11R,14R,15S) (**92**) [38]. From *Kadsura angustifolia* fermented by an associated symbiotic endophytic fungus, *Penicillium* sp. SWUKD4.1850, nine undescribed triterpenoids, kadhenrischinins A–H **93**–**100** (Figure 4), and 7β-schinalactone C **101** (Figure 4) were isolated and established. All these metabolites have been first detected in non-fermented *K. angustifolia*. Structurally, kadhenrischinins A–D (**93**–**96**) belong to the relatively rare class of highly oxygenated schitriterpenoids that contain a unique 3-one-2-oxabicyclo [3,2,1]-octane motif, while kadhenrischinins E–H (**97**–**100**) feature acyclopentane ring in a side chain rarely found in the family Schisandraceae [39]. Meroterpenoids with diverse ring systems including five new ones (**102**–**106**) (Figure 4), were isolated from *Phyllosticta capitalensis*, an endophytic fungus from *Cephalotaxus fortunei* Hook. Compound **102** was the first example with a 9,14-seco ring and a five-membered ring in guignardone derivatives. Compound **103** represented a novel guignardone derivative possessing a 5/7/6/5 ring system with CH2-7 attached to C-4 rather than C-6 in ring D [40]. Nine new meroterpenes, (7R,8R)-8-hydroxysydowic acid **107** (Figure 4), (7S,10S)-10-hydroxy-sydowic acid **108** (Figure 4), (7S,11R)-12-hydroxy-sydowic acid **109** (Figure 4), (7S,11R)-12-acetoxy-sydowic acid **110** (Figure 4), (7R,8R)-1,8-epoxy-11-hydroxy-sydonic acid **111** (Figure 4), 7-deoxy-7,14-didehydro-11-hydroxysydonic acid **112** (Figure 4), 7-deoxy-7,14-didehydro-12-acetoxy-sydonic acid **113** (Figure 4), and (E)-7-deoxy-7,8-didehydro-12-acetoxy-sydonic acid **114** (Figure 4), (7R)-11-hydroxy-sydonic acid methyl ester **115** (Figure 4), were isolated from the solid rice culture of the endophytic fungus *Aspergillus versicolor* [35]. Bioassay-guided fractionation of the crude extract of fermentation broth of one symbiotic strain *Fusarium oxysporum* ZZP-R1 derived from coastal plant *Rumex madaio Makino*, one traditional Chinese medicine used as a treatment of inflammation and toxication, yielded one novel compound, fusariumins C **116** (Figure 4). Chemical structure of **116** was determined as one meroterpene with cyclohexanone moiety [30]. A new monoterpentoid lithocarin D **117**, was isolated from the endophytic fungus *Diaporthe lithocarpus* A740 (Figure 4) [41].

### 2.2. Ketone Compounds

#### 2.2.1. Polyketides

An endophytic fungus, *Eupenicillium* sp. LG41, isolated from the Chinese medicinal plant *Xanthium sibiricum*, was subjected to epigenetic modulation using an NAD^+^-dependent histone deacetylase (HDAC) inhibitor, nicotinamide. Epigenetic stimulation of the endophyte led to enhanced production of two new decalin-derived polyketides with a double bond between C-3 and C-4, eupenicinicols C **118** (Figure 5) and D **119** (Figure 5) [42]. On the basis of One Strain/Many Compounds (OSMAC) strategy, five new polyketides, named phomopsiketones A–C **120**–**122** (Figure 5), (10S)-10-O-b-D-40-methoxymannopyranosyldiaporthin **123** (Figure 5), and clearanol **124** (Figure 5), were isolated from an endophytic fungus, *Phomopsis* sp. sh917, harbored in stems of *Isodon eriocalyx* var. laxiflora [43]. As naturally occurring polyketides, ten new salicyloid derivatives, namely vaccinols J–S **125**–**134** (Figure 5), were isolated from *Pestalotiopsis vaccinii* (cgmcc3.9199) endogenous with the mangrove plant *Kandelia candel* (L.) Druce (*Rhizophoraceae*) [44]. Twelve new polyketides, penicichrysogenins A–L **135**–**146** (Figure 5), were isolated from the solid substrate fermentation cultures of a Huperzia serrata endophytic fungus *Penicillium chrysogenum* MT-12. The structures of **135**–**139** were established as (2R)-6-hydroxy-2,4-dimethoxy-5-methylphthalide (**135**), 4,6-dihydroxy-5-hydroxymethylphthalide9 (**136**), 4,6-dihydroxy-5-methoxymethylphthalide (**137**), (2R)-4,5-dihydroxy-2,6-dimethoxy-2-pentylphthalide (**138**), (E)-4,5-dihydroxy-2-(4-hydroxypentylidene)-6-methoxyphthalide(**139**), respectively [45]. Three new polyketides, cylindrocarpones A–C **147**–**149** (Figure 5), were isolated from the endophytic fungus, *Cylindrocarpon* sp., obtained from the tropical plant *Sapium ellipticum* [46]. Six new xanthone-derived polyketides, named phomoxanthones F–K **150**–**155** (Figure 5), were isolated from *Phomopsis* sp. xy21, which was isolated as an endophytic fungus from the Thai mangrove *Xylocarpus granatum*. Phomoxanthone F **150** represented the first xanthone-derived polyketide containing a 10a-decarboxylated benzopyranone nucleus that was substituted by a 4-methyldihydrofuran-2(3H)-one moiety at C10a. Phomoxanthones G **151** and H **152** are highly oxidized xanthone-derived polyketides containing a novel 5-methyl-6-oxabicyclo [3.2.1] octane motif [47]. Compound **156** (Figure 5), 5,9-dihydroxy-2,4,6,8,10-pentamethyldodeca-2,6,10-trienal, a novel polyketide molecule was isolated from *Aspergillus flocculus* endophyte isolated from the stem of the medicinal plant *Markhamia platycalyx* [48]. Three new polyketides, (2S)-2,3-dihydro-5,6-dihydroxy-2-methyl-4H-1-benzopyran-4-one **157** (Figure 5), (2′R)-2-(2′-hydroxypropyl)-4-methoxyl-1,3-benzenediol **158** (Figure 5), and 4-ethyl-3-hydroxy-6-propenyl-2H-pyran-2-one **159** (Figure 5) were isolated from the culture broth of *Colletotrichum gloeosporioides*, an endophytic fungus derived from the mangrove *Ceriops tagal* [49]. Five polyketides, paralactonic acids A–E **160**–**164** (Figure 5) were isolated from *Paraconiothyrium* sp. SW-B-1, an endophytic fungus isolated from the seaweed, *Chondrus ocellatus Holmes* [50]. Four new polyketides, alternatains A–D **165**–**168** (Figure 5), were obtained from the solid substrate fermentation cultures of *Alternaria alternata* MT-47, an endophytic fungus isolated from the medicinal plant of *Huperzia serrata* [51]. From extracts of the plant associated fungus *Chaetosphaeronema achilleae* collected in Iran, two polyketides including a previously unreported isoindolinone named chaetosisoindolinone **169** (Figure 5) and a previously undescribed indanone named chaetosindanone **170** (Figure 5) were isolated [52]. During a survey of the secondary metabolites of endophytic fungi *Aspergillus porosus*, new polyketides with interesting structural features named porosuphenols A–D **171**–**174** (Figure 5) were found [53]. Chemical investigation of the EtOAc extract of the plant-associated fungus *Alternaria alternate* in rice culture led to the isolation of a novel liphatic polyketone, alternin A **175** (Figure 5), which possesses an unprecedented C25 liphatic polyketone skeleton [26]. Five new polyketides, colletotric B **176** (Figure 5), 3-hydroxy-5-methoxy-2,4,6-trimethylbenzoic acid **177** (Figure 5), colletotric C **178** (Figure 5), chaetochromone D **179** (Figure 5) and 8-hydroxy-pregaliellalactone B **180** (Figure 5), were isolated from thmangrove endophytic fungus *Phoma* sp. SYSU-SK-7 [54]. The EtOAc extract of *Phomopsis* sp. D15a2a isolated from the plant *Alternanthera bettzickiana* following fermentation on solid rice medium yielded three new polyketides, phomopones A−C **181**–**183** (Figure 5) [55]. Three new polyketides including two benzophenone derivatives, penibenzones A (**184**) and B (**185**) (Figure 5), and a new phthalide derivative, penibenzone C **186** (Figure 5), were isolated from the solid-substrate cultures of the endophytic fungus *Penicillium purpurogenum* IMM003 [56].

#### 2.2.2. Other Ketones

A new N-methoxypyridone analog 11S-hydroxy-14-methyl cordypyridone C **187** (Figure 6), was isolated from the co-culture of Hawaiian endophytic fungi *Camporesia sambuci* FT1061 and *Epicoccum sorghinum* FT1062 [57]. A novel endophyte *Rhytismataceae* sp. DAOMC 251461 produced two new dihydropyrones: (R)-4-hydroxy-5-octanoyl-6-oxo-3,6-dihydropyran-2-carboxylic acid (rhytismatone A) **188** (Figure 6) and (R)-methyl-4-hydroxy-5-octanoyl-6-oxo-3,6-dihydropyran-2-carboxylate (rhytismatone B) **189** (Figure 6) [58]. Five new bioactive 2-pyrone metabolites, phomaspyrones A–E **190**–**194** (Figure 6), were isolated from the culture broth of an endophytic fungus *Phomopsis asparagi* SWUKJ5.2020 of medicinal plant *Kadsura angustifolia*. The structures of **190**–**194** were identified as (S)-5-(1,2-dihydroxyethyl)-6-hydroxymethyl-4-methoxy-2H-pyran-2-one (**190**),(S)-5-(1-hydroxyethyl)-6-hydroxymethyl-4-methoxy-2H-pyran-2-one (**191**), (5S,8R)-5,8-dihydroxy-4-methoxy-5,6-dihydropyrano-[3,4-b]pyran-2(8H)-one (**192**), 4-methoxy-6-methyl-5-(2-oxobutyl)-2H-pyran-2-one (**193**), 6-(hydroxymethyl)-4-methoxy-5-(2-oxobutyl)-2H-pyran-2-one (**194**) respectively [59]. Extracts from an endophytic fungus *Dendrothyrium variisporum* isolated from the roots of the Algerian plant *Globularia alypum* produced two new minor furanone derivatives: methyl (5S)-5-[(10E,30Z)-hexa-1,3-dienyl]-5-methyl-4-oxo-2-methyl-4,5-dihydrofuran-3 carboxylate ((5S) cis-gregatin B) **195** (Figure 6), (5R)-5-[(10E,30Z)-hexa-1,3-dienyl]-5-methyl-4-oxo-2-[(4S,1E)-4-hydroxypent-1-enyl]-4,5-dihydrofuran-3carboxylate, (graminin D) **196** (Figure 6) [60]. Two new compounds isobenzofuranone A **197** (Figure 6) and indandione B **198** (Figure 6), were isolated from liquid cultures of an endophytic fungus *Alternaria* sp., which was obtained from the medicinal plant *Morinda officinalis*. Among them, the indandione 198 showed a rarely occurring indanone skeleton in natural products [61]. An endophytic fungal strain named *Trichoderma atroviride* was isolated from the bulb of *Lycoris radiata*. Following cultivation on rice medium, a new cyclopentenone derivative, atrichodermone B **199** (Figure 6), was isolated [11]. One previously undescribed isochromone derivative 6,8-dihydroxy-3-(2-hydroxypropyl)-7-methyl-1H-isochromen-1-one **200** (Figure 6), was isolated from the culture of the endophytic fungus *Eurotium chevalieri* KUFA 0006 [62]. One previously undescribed pyrone (simplicilopyrone) **201** (Figure 6) was isolated from the endophytic fungus *Simplicillium* sp. PSU-H41 [63]. Cytosporaphenones A–C, one new polyhydric benzophenone **202** (Figure 6) and two new naphtopyrone derivatives **203**–**204** (Figure 6), were isolated from *Cytospora rhizophorae*, an endophytic fungus from *Morinda officinalis* [64]. A novel pyrone derivative **205** (Figure 6) bearing two fused five-member rings, together with two new naphthalenone derivatives **206**–**207** (Figure 6), were obtained from the endophytic fungus *Fusarium* sp. HP-2, which was isolated from “Qi-Nan” agarwood [65]. Two new compounds penibenzophenones A-B **208**–**209** (Figure 6), were isolated from the EtOAc extract of the endophytic fungus *Penicillium citrinum* HL-5126 isolated from the mangrove *Bruguiera sexangula var. rhynchopetala* collected in the South China Sea [66]. Two new isochromanone derivatives, (3S,4S)-3,8-dihydroxy-6-methoxy-3,4,5-trimethylisochroman-1-one **210** (Figure 6) and methyl (S)-8-hydroxy-6-methoxy-5-methyl-4a-(3-oxobutan-2-yl)benzoate **211** (Figure 6), were isolated from the cultures of an endophytic fungus *Phoma* sp. PF2 obtained from *Artemisia princeps* [67]. Isoshamixanthone **212** (Figure 6), a new stereoisomeric pyrano xanthone was obtained from the endophytic fungal strain *Aspergillus* sp. ASCLA isolated from leaf tissues of the medicinal plant *Callistemon subulatus* [68]. From the endophytic fungus, *Cylindrocarpon* sp., obtained from the tropical plant *Sapium ellipticum*, a new pyrone cylindropyrone **213** (Figure 6) was isolated [46]. One new benzophenone derivative, named tenllone I **214** (Figure 6), was isolated from the endophytic fungus *Diaporthe lithocarpus* A740 [41].

### 2.3. Alkaloids and Their Derivatives

The endolichenic fungus *Apiospora montagnei* isolated from the lichen *Cladonia* sp. was cultured on solid rice medium, yielding a new pyridine alkaloid, 23-O-acetyl-N-hydroxyapiosporamide **215** (Figure 7) [37]. Chaetoindolin A **216** (Figure 7), a new indole alkaloid derivative was isolated from the endophytic fungus *Chaetomium globosum* CDW7 [69]. A synthetic α,β-unsaturated amide alkaloid (E)-tert-butyl(3-cinnamamidopropyl) carbamate **217** (Figure 7), newly identified as a natural product, was isolated from the EtOAc extract of the endophytic fungus *Penicillium citrinum* HL-5126 isolated from the mangrove *Bruguiera sexangula* var. *Rhynchopetala* [66]. A new alkaloid, 1, 2-dihydrophenopyrrozin **218** (Figure 7), was isolated from an axenic culture of the endophytic fungus, *Bionectria* sp., obtained from seeds of the tropical plant *Raphia taedigera* [70]. Two new pyridone alkaloids, cylindrocarpyridones A–B **219**–**220** (Figure 7), were isolated from the endophytic fungus, *Cylindrocarpon* sp., obtained from the tropical plant *Sapium ellipticum* [46]. From *Aspergillus versicolor*, an endophyte derived from leaves of the Egyptian water hyacinth *Eichhornia crassipes* (Pontederiaceae), one new compound aflaquinolone H **221** (Figure 7) belonging to dihydroquinolone alkaoids was obtained [71]. Two new spiroketal derivatives as alkaloids with an unprecedented amino group, 2′-aminodechloromaldoxin **222** (Figure 7) and 2′-aminodechlorogeodoxin **223** (Figure 7), were isolated from the plant endophytic fungus *Pestalotiopsis flavidula* [72]. The biotransformation of lycopodium alkaloid huperzine A (hupA), one of the characteristic bioactive constituents of the medicinal plant *Huperzia serrata*, by a fungal endophyte of the host plant was studied. Two previously undescribed compounds **224**–**225** (Figure 7), were isolated and identified [73]. Chemical investigation of the EtOAc extract of the plant-associated fungus *Alternaria alternate* in rice culture led to the isolation of a new indole alkaloid **226** (Figure 7) [26]. Bioactivity-guided isolation of the endophytic fungus *Fusarium sambucinum* TE-6L residing in *Nicotiana tabacum* L. led to the discovery of two new angularly prenylated indole alkaloids (PIAs) with pyrano[2,3-g]indole moieties, amoenamide C **227** (Figure 7) and sclerotiamide B **228** (Figure 7). Compound **227** containing the 8 bicyclo[2.2.2]diazaoctane core and indoxyl unit was rarely reported [74].

### 2.4. Penylpropanoids and Their Derivatives

A new isocoumarin (3R,4S,4aR,6R)-4,6,8-trihydroxy-3-methyl-3,4,4a,5,6,7-hexahydroisochromen-1-one **229** (Figure 8) was isolated from an endophyte *Mycosphaerellaceae* sp. DAOMC 250863 [58]. Using the bioassay-guided method, one new isocoumarin derivative, prochaetoviridin A **230** (Figure 8), was isolated from *C. globosum* CDW7, an endophyte from *Ginkgo biloba* [66]. A new isocoumarin derivative pestalotiopisorin B **231** (Figure 8), was isolated from *Pestalotiopsis* sp. HHL-101, an endophytic fungus obtained from Chinese mangrove plant *Rhizophora stylosa* [75]. In continuing search of fungal strain *Nectria pseudotrichia* 120-1NP, two new isocoumarins, namely, nectriapyrones A **232** (Figure 8) and B **233** (Figure 8) were identified [31]. Two new isocoumarin dimers **234**–**235** (Figure 8) were isolated from *Aspergillus versicolor*, an endophyte derived from leaves of the Egyptian water hyacinth *Eichhornia crassipes* (Pontederiaceae) [71]. Pestalustaines **236** (Figure 8), one unprecedented coumarin derivative bearing 6/6/5/5-fused tetracyclic ring system, was isolated from a plant-derived endophytic fungus *Pestalotiopsis adusta* [19]. Compounds **237** (Figure 8) and **238** (Figure 8), determined as two novel isocoumarin derivatives with a different butanetriol group at C-3, were produced by *T. harzianum* (*Trichoderma harzianum*) Fes1712 isolated from Rubber Tree *Ficus elastica* leaves [76]. Two pairs of new isocoumarin derivatives penicoffrazins B and C, **239**–**240** (Figure 8), were isolated from *Penicillium coffeae* MA-314, an endophytic fungus obtained from the fresh inner tissue of the leaf of marine mangrove plant *Laguncularia racemosa* [77]. A new dihydroisocoumarin, diaporone A **241** (Figure 8), was isolated from the ethyl acetate extract of the cultures of the endophytic fungus *Diaporthe* sp. [78].

### 2.5. Lactones

From the seeds of the traditional medicinal plant *Ziziphus jujuba* growing in Uzbekistan, the fungal endophyte *Alternaria* sp. was isolated. Extracts of this fungus yielded a new natural phthalide derivative 7-methoxyphthalide-3-acetic acid **242** (Figure 9) [79]. Three new lactone Derivatives isoaigialones, A, B, and C **243**–**245** (Figure 9), were isolated from the crude EtOAc extract of a *Phaeoacremonium* sp., an endophytic fungus obtained from the leaves of *Senna spectabilis*. **245** is epimeric at C-7 relative to compound **244** [80]. A new phytotoxic bicyclic lactone (3aS,6aR)-4,5-dimethyl-3,3a,6,6a-tetrahydro-2H-cyclopenta [b]furan-2-one **246** (Figure 9), was isolated from the ethyl acetate extract of fermentation broth of *Xylaria curta* 92092022 [81]. Three new lactones de-O-methyllasiodiplodins, (3R,7R)-7-hydroxy-de-O-methyllasiodiplodin **247** (Figure 9) and (3R)-5-oxo-deO-methyllasiodiplodin **248** (Figure 9), together with (3R)-7-oxo-de-O-methyllasiodiplodin **249** (Figure 9) were isolated from the co-cultivation of mangrove endophytic fungus *Trichoderma* sp. 307 and aquatic pathogenic bacterium *Acinetobacter johnsonii* B2 [10]. Two new lactones, pestalotiolactones A **250** (Figure 9) and B **251** (Figure 9), were isolated from the axenic culture of the endophytic fungus *Pestalotiopsis* sp., obtained from fruits of *Drepanocarpus lunatus* (Fabaceae) [12]. Active metabolites investigation of *Talaromyces* sp. (strain no. MH551540) associated with *Xanthoparmelia angustiphylla* afforded a new 3-methoxy-4,8-bihydroxymethyl-6-methyl-2,4,6-3en-δ-lactone, talaromycin A **252** (Figure 9) [82]. Introducing an alien carbamoyltransferase (*asm*21) gene into the *Streptomyces* sp. CS by conjugal transfer, as a result, one recombinatorial mutant named CS/*asm*21-4 was successfully constructed. From the extracts of the CS/*asm*21-4 cultured on oatmeal solid medium, a new macrolide hookerolide **253** (Figure 9) was obtained [83]. Four new aromatic butenolides, asperimides A–D **254**–**257** (Figure 9), were isolated from solid cultures of a tropical endophytic fungus *Aspergillus terreus*. Compounds **254**–**257** represent the first examples of butenolides with a maleimide core isolated from *Aspergillus* sp. [84]. In ongoing search for bioactive metabolites from the genus of *Aspergillus*, four new butenolides, namely terrusnolides A–D **258**–**261** (Figure 9) were isolated from an endophytic *Aspergillus* from *Tripterygium wilfordii*. Compound **258** was a butenolide derived by a triple decarboxylation. Furthermore, compounds **259**–**261** were the 4-benzyl-3-phenyl-5H-furan-2-one derivatives with an isopentene group fused to the benzene ring [85]. Chemical investigation on the culture extract of *H. fuscum* fermented on rice led to the isolation of one new 10-membered lactone 5,6-Epoxy-phomol **262** (Figure 9) [86]. Three new spirocyclic anhydride derivatives **263**–**265** (Figure 9) were isolated from the endophytic fungus *Talaromyces purpurogenus* obtained from fresh leaves of the toxic medicinal plant *Tylophora ovate* [87]. A new δ-lactone penicoffeazine A, **266** (Figure 9) was isolated from *Penicillium coffeae* MA-314, an endophytic fungus obtained from the fresh inner tissue of the leaf of marine mangrove plant *Laguncularia racemosa* [77]. On the basis of One Strain/Many Compounds (OSMAC) strategy, a new natural product **267** (Figure 9), was isolated from an endophytic fungus, *Phomopsis* sp. sh917, harbored in stems of *Isodon eriocalyx* var. *laxiflora* [43]. A chemical investigation on metabolites of *Phyllosticta* sp. J13-2-12Y isolated from the leaves of *Acorus tatarinowii* was carried out, which led to the isolation of four new phenylisotertronic acids, R-xenofuranone B **268** (Figure 9), S-xenofuranone B **269** (Figure 9), enantio-flflavipesin B **270** (Figure 9), and S-3-hydroxy-4,5-diphenylfuran-2(5H)-one **271** (Figure 9) [88]. An endophytic fungus *Pestalotiopsis microspora* isolated from the fruits of *Manilkara zapota* was cultured in potato dextrose broth media. Chromatographic separation of the EtOAc extract of the broth and mycelium led to the isolation of a new azaphilonoid named pitholide E **272** (Figure 9) [89].

### 2.6. Anthraquinones

An endophytic fungus *Penicillium citrinum* Salicorn 46 isolated from *Salicornia herbacea* Torr., Produced one new citrinin derivative, pencitrinol **273** (Figure 10) [90]. *Lachnum* cf. *pygmaeum* DAOMC 250335 was obtained from ascospores originating from a collection of apothecia occurring on a dead *P. rubens* twig, from this strain, a new chlorinated para-quinone, chloromycorrhizinone A **274** (Figure 10) was isolated [58]. The endolichenic fungus *Apiospora montagnei* isolated from the lichen *Cladonia* sp. was cultured on solid rice medium, yielding a new xanthone derivative 8-hydroxy-3-hydroxymethyl-9-oxo-9Hxanthene-1-carboxylic acid methyl ether **275** (Figure 10) [37]. One previously undescribed metabolite anthraquinone derivative acetylquestinol **276** (Figure 10), was isolated from the culture of the endophytic fungus *Eurotium chevalieri* KUFA 0006 [62]. New pulvilloric acid-type azaphilones **277**–**280** (Figure 10) were produced by *Nigrospora oryzae* co-cultured with *Irpex lacteus* [14]. A new shunt product spiciferone F **281** (Figure 10) together with two new analogs spiciferones G **282** (Figure 10) and H **283** (Figure 10) were isolated from endophytic fungus *Phoma betae* inhabiting in plant *Kalidium foliatum* (Pall.) [91]. Bioassay-guided fractionation of the dichloromethane extract of the fungus *Neofusicoccum austral* SYSU-SKS024 led to the isolation of three new ethylnaphthoquinone derivatives, neofusnaphthoquinone A **284** (Figure 10), 6-(1-methoxylethy1)-2,7-dimethoxyjuglone **285** (Figure 10), (3R,4R)-3-methoxyl-botryosphaerone D **286** (Figure 10), Neofusnaphthoquinone A **285** is the third example of the unsymmetrical naphthoquinone [92]. The EtOAc extract of strain *Nectria pseudotrichia* 120-1NP led to the identification of one new naphthoquinone, namely, nectriaquinone B **287** (Figure 10) [31]. Cytoskyrin C **288** (Figure 10), a new bisanthraquinone with asymmetrically cytoskyrin type skeleton, was isolated from an endophytic fungus ARL-09 (*Diaporthe* sp.) from *Anoectochilus roxburghii* [93]. Three new naphthomycins O–Q **289**–**291** (Figure 10), were obtained from the solid cultured medium of recombinatorial mutant strain CS/*asm21*-4 (By introducing an alien carbamoyltransferase (*asm21*) gene into the strain *Streptomyces* sp. CS (CS) by conjugal transfer) [83]. From the fermentation broth of the endophytic fungus *Xylaria* sp.SYPF 8246, one new compound, xylarianins B **292** (Figure 10) was isolated [94]. An undescribed substituted dihydroxanthene-1,9-dione, named funiculosone **293** (Figure 10), was isolated together from the culture filtrates of *Talaromyces funiculosus* (Thom) Samson, Yilmaz, Frisvad & Seifert (Trichocomaceae), an endolichenic fungus isolated from lichen thallus of *Diorygma hieroglyphicum* (Pers.) Staiger & Kalb (Graphidaceae), in India [95]. One new dihydroxanthenone derivative globosuxanthone E **294** (Figure 10) was obtained from the crude extracts of two endophytic fungi *Simplicillium lanosoniveum* (J.F.H. Beyma) Zare & W. Gams (*Sarocladium strictum*) PSU-H168 and PSU-H261 which were isolated from the leaves of *Hevea brasiliensis* [96]. Two new naphthoquinone derivatives, 6-hydroxy-astropaquinone B **295** (Figure 10) and astropaquinone D **296** (Figure 10) were isolated from *Fusarium napiforme*, an endophytic fungus isolated from the mangrove plant, *Rhizophora mucronata* [97].

### 2.7. Sterides

Two new steroids, (24R)-22, 23-dihydroxy-ergosta-4,6,8(14)-trien-3-one 23-β-d-glucopyranoside **297** (Figure 11), and xylarester **298** (Figure 11), were isolated from the extract of endophytic *Xylaria* sp. solid culture. Compound **298** has an unprecedent ergosta skeleton with a six-membered lactonic group in A ring [98]. An endophytic fungus, *Chaetomium* sp. M453 isolated from *Huperzia serrata* (Thunb. ex Murray) Trev yield four new steroids including three unusual C25 steroids, neocyclocitrinols E–G **299**–**301** (Figure 11), and 3β-hydroxy-5,9-epoxy-(22E,24R)-ergosta-7,22-dien-6-one **302** (Figure 11) [99]. Three new methylated Δ8-pregnene steroids, stemphylisteroids A–C **303**–**305** (Figure 11) were isolated from the medicinal plant Polyalthia laui-derived fungus *Stemphylium* sp.AZGP4-2. The discovery of those three steroids is a further addition to diverse and complex array of methylated steroids [100]. Three new ergosterol derivatives, namely, fusaristerols B [(22E,24R)-3-palmitoyl-19(10→6)-abeo-ergosta-5,7,9,22-tetraen-3β-ol] **306** (Figure 11), fusaristerols C [(22E,24R)-ergosta-7,22-diene-3β,6β,9α-triol] **307** (Figure 11), and fusaristerols D [(22E,24R)-ergosta-7,22-diene-3β,5α,6β,9α-tetraol 6-acetate] **308** (Figure 11), were isolated and characterized from the endophytic fungus *Fusarium* sp. isolated from *Mentha longifolia* L. (Labiatae) roots growing in Saudi Arabia [101]. A new ergosterol derivative, 23R-hydroxy-(20Z,24R)-ergosta-4,6,8(14),20(22)-tetraen-3-one **309** (Figure 11), was isolated from the co-culture between endophytic fungus *Pleosporales* sp. F46 and endophytic bacterium *Bacillus wiedmannii* Com1 both inhibiting in the medicinal plant *Mahonia fortunei*. This is the first example of isolation of a ergosterol derivative with a Δ20(22)-double bond in the side chain [102]. Two new sterol derivatives, namely ergosterimide B **310** (Figure 11) and demethylincisterol A5 **311** (Figure 11), were isolated from the rice fermentation culture of *Aspergillustubingensis* YP-2 [103].

### 2.8. Other Types of Compounds

An endophytic fungus *Talaromyces stipitatus* SK-4 was isolated from the leaves of a mangrove plant *Acanthus ilicifolius*. Its crude extract exhibited significant antibacterial activity was purified to afford two new depsidones, talaromyones A and B **312**–**313** (Figure 12) [104]. Four new amide derivatives, designated as cordycepiamides A–D **314**–**317** (Figure 12), were isolated from the EtOAc-soluble fraction of the 95% EtOH extract of long-grain rice fermented with the endophytic fungus *C. ninchukispora* BCRC 31900, derived from the seeds of medicinal plant *Beilschmiedia erythrophloia* Hayata [105]. One new 4-hydroxycinnamic acid derivatives, methyl 2-{(E)-2-[4-(formyloxy)phenyl]ethenyl}-4-methyl-3-oxopentanoate **318** (Figure 12), was isolated from an EtOAc extract derived from a solid rice medium of endophytic fungal strain *Pyronema* sp. (A2-1 & D1-2) [106]. When endophytic fungus *Phoma* sp. nov. LG0217 isolated from *Parkinsonia microphylla* cultured in the absence of the epigenetic modifier, it can produced a new metabolite, (S,Z)-5-(3′,4-dihydroxybutyldiene)-3-propylfuran-2(5H)-one **319** (Figure 12) [107]. One new citrinin derivatives, pencitrin **320** (Figure 12) was isolated from an endophytic fungus *P.*
*citrinum* 46 derived from *Salicornia herbacea* Torr by adding CuCl_2_ into fermentation medium [90]. Two new cytosporone derivatives **321**–**322** (Figure 12) were isolated from the endophytic fungus *Phomopsis* sp. PSU-H188 [108]. Extensive chemical investigation of the endophytic fungus, *Fusarium solani* JK10, harbored in the root of the Ghanaian medicinal plant *Chlorophora regia*, using the OSMAC (One Strain Many Compounds) approach resulted in the isolation of seven new 7–desmethyl fusarin C derivatives **323**–**329** (Figure 12) [109]. A new biphenyl derivative 5,5′-dimethoxybiphenyl-2,2′-diol **330** (Figure 12), was isolated from the mangrove endophytic fungus *Phomopsis longicolla* HL-2232 [110]. A new hexanedioic acid analogue, (2S,5R)-2-ethyl-5-methylhexanedioic acid **331** (Figure 12), was isolated from *Penicillium* sp. OC-4, an endophytic fungus associated with Orchidantha chinensis [111]. The endophytic fungus *Curvularia* sp. strain (M12) was isolated from a leaf of the medicinal plant *Murraya koenigii* and cultured on rice medium. Chromatographic analysis led to the isolation of four new compounds, murranofuran A **332** (Figure 12), murranolide A **333** (Figure 12), murranopyrone **334** (Figure 12), and murranoic acid A **335** (Figure 12) [112]. The cultivation of the mangrove-derived fungus *Rhytidhysteron rufulum* AS21B in acidic condition could change its secondary metabolite profile. Investigation of the culture broth extract led to the isolation and identification of two new spirobisnaphthalenes **336**–**337** (Figure 12) [113]. On the basis of One Strain/Many Compounds (OSMAC) strategy, one new natural product **338** (Figure 12), was isolated from an endophytic fungus, *Phomopsis* sp. sh917, harbored in stems of *Isodon eriocalyx* var. *laxiflora* [43]. Extracts from an endophytic fungus *Dendrothyrium variisporum* isolated from the roots of the Algerian plant *Globularia alypum* yielded three new anthranilic acid derivatives **339**–**341** (Figure 12) [60]. An endophytic fungal strain named *Trichoderma atroviride* was isolated from the bulb of *Lycoris radiata*. Following cultivation on rice medium, a novel 3-amino-5-hydroxy-5-vinyl-2-cyclopenten-1-one dimer, atricho dermone A **342** (Figure 12), was isolated. Compound **342** is the first example of cyclopentene dimer [11]. A new chaetoglobosin, penochalasin K **343** (Figure 12) bearing an unusual six-cyclic 6/5/6/5/6/13 fused ring system, was isolated from the solid culture of the mangrove endophytic fungus *Penicillium chrysogenum* V11 [114]. Three previously undescribed metabolites, including two prenylated indole 3-carbaldehyde derivatives **344**–**345** (Figure 12), an anthranilic acid derivative **346** (Figure 12) were isolated from the culture of the endophytic fungus *Eurotium chevalieri* KUFA 0006. The structures of compounds were established as 2-(2-methyl-3-en-2-yl)-1H-indole-3-carbaldehyde (**344**), (2,2-dimethylcyclopropyl)-1H-indole-3-carbaldehyde (**345**), 2[(2,2-dimethylbut-3-enoyl)amino]benzoic acid (**346**) [62]. Nine previously undescribed depsidones simplicildones A–I **347**–**355** (Figure 12) were isolated from the endophytic fungus *Simplicillium* sp. PSU-H41 [63]. Six new compounds including four tyrosine derivatives terezine M **356** and phomarosines A–C **357**–**359** (Figure 12), and two new hydantoin derivatives, (S)-5-isopropyl-3-methoxyimidazolidine-2,4-dione **360** (Figure 12) and (S)-5-(4-hydroxybenzoyl)-3-isobutyrylimidazolidine-2,4-dione **361** (Figure 12), were obtained from the investigation of the endophytic fungus *Phoma herbarum* PSU-H256, which was isolated from a leaf of *Hevea brasiliensis* [115]. New mellein derivative; 4-methylmellein **362** (Figure 12) was isolated from the ethyl acetate extract of the endophytic fungus *Penicillium* sp. isolated from the leaf of *Senecio flavus* (Asteraceae) [116]. One novel cytochalasin, named jammosporin A **363** (Figure 12) was isolated from the culture of the endophytic fungus *R. sanctae-cruciana*, harboured from the leaves of the medicinal plant *A. lebbeck* [117]. An endophytic fungus *Arthrinium arundinis* TE-3 was isolated and purified from the fresh leaves of cultivated tobacco (*Nicotiana tabacum* L.). Chemical investigation on this fungal strain afforded three new prenylated diphenyl ethers **364**–**366** (Figure 12) [118]. A novel indene derivative **367** (Figure 12), have been purified from an ethyl acetate extract of the plant-associated fungus *Aspergillus flavipes* Y-62, isolated from Suaeda glauca (Bunge) Bunge [119]. The endophytic fungus *Mycosphaerella* sp. (UFMGCB2032) was isolated from the healthy leaves of *Eugenia bimarginata*, a plant from the Brazilian savanna. Two novel usnic acid derivatives, mycousfuranine **368** (Figure 12) and mycousnicdiol **369** (Figure 12), were isolated from the ethyl acetate extract [120]. Intriguingly, incorporaion of Cu^2+^ into the PDB medium of the endophytic fungus, *Anteaglonium* sp. FL0768 enhanced production of metabolites and drastically affected the biosynthetic pathway resulting in the production of pentaketide dimers, palmarumycin CE4 **370** (Figure 12). The structure of palmarumycin CE4 **370** was established as (2β,4aα,5β,8β,8aα)-2,3,4a,5,8,8a-hexahydro-5-hydroxy-spiro [2,8-epoxynaphthalene]-1(4H)-2′-naphtho[1,8-de][1,3]dioxin-4-one [121]. Three new compounds, including rotational isomers **371**–**372** (Figure 12) and **373** (Figure 12) were isolated from the solid cultures of the endophytic fungus *Penicillium janthinellum* SYPF 7899, compound **372** is the rotamer of **371** [122]. The chemical assessment of endophyte *Phaeophleospora vochysiae* sp. nov from *Vochysia divergens*, revealed a new compound 3-(sec-butyl)-6-ethyl-4,5-dihydroxy-2-methoxy-6-methylcyclohex-2-enone **374** (Figure 12) [123]. Co-cultivation of fungus *Bionectria* sp. either with *Bacillus subtilis* or with *Streptomyces lividans* resulted in the production of two new o-aminobenzoic acid derivatives, bionectriamines A and B **375**–**376** (Figure 12) [70]. Chemical investigation on the solid rice culture of *Trichoderma atroviride* S361, an endophyte isolated from *Cephalotaxus fortunei*, has afforded a pair of novel N-furanone amide enantiomers, (−)-trichodermadione A **377** (Figure 12) and (+)-trichodermadione A **378** (Figure 12). The structure of **377** was identified as (4′R,2E)-N-(2-ethyl-5-methyl-3-oxo-2,3-dihydrofuran-2-yl)-5-hydroxy-3-methylpent-2-enamide [17]. Secondary metabolites were isolated from the fermentation broth of the endophytic fungus *Xylaria* sp.SYPF 8246, including four new compounds, xylarianins A–D **379**–**382** (Figure 12), three new natural products, 6-methox-ycarbonyl-2′-methyl-3,5,4′,6′-tetramethoxy-diphenyl ether **383** (Figure 12), 2-chlor-6-methoxycarbonyl-2′-rnethyl-3,5,4′,6′-tetramethoxy-diphenyl ether **384** (Figure 12), and 2-chlor-4′-hydroxy-6-methoxy carbonyl-2′-methyl-3,5,6′-trimethoxy-diphenyl ether **385** (Figure 12) [94]. Bysspectin A **386** (Figure 12), a polyketide-derived octaketide dimer with a novel carbon skeleton, and two new precursor derivatives, bysspectins B and C **387**–**388** (Figure 12), were obtained from an organic extract of the endophytic fungus *Byssochlamys spectabilis* that had been isolated from a leaf tissue of the traditional Chinese medicinal plant E*dgeworthia chrysantha* [124]. Fusarithioamide B **389** (Figure 12), a new aminobenzamide derivative with unprecedented carbon skeleton was separated from *Fusarium chlamydosporium* EtOAc extract isolated from *Anvillea garcinii* (Burm.f.) DC. Leaves (Asteraceae) [125]. The study of endophytic fungus *Annulohypoxylon stygium* (Xylariaceae family) isolated from *Bostrychia radicans* algae led to the isolation of a novel compound, 3-benzylidene-2-methylhexahydropyrrolo [1,2-α] pyrazine-1,4-dione **390** (Figure 12) [126]. A new 2H-benzindazole derivative, alterindazolin A **391** (Figure 12), has been isolated from cultures of the endophyte *Alternaria alternata* Shm-1obtained from the fresh wild body of *Phellinus igniarius*. The structure of **391** was elucidated for N-benzyl-3-[p-hydroxy phenyloxygen]-benz[e]indazole [127]. One new pentenoic acid derivative, named 1,1′-dioxine-2,2′-dipropionic acid **392** (Figure 12) and a new natural product, named 2-methylacetate-3,5,6-trimethylpyrazine **393** (Figure 12), were obtained from the *Cladosporium* sp. JS1-2, an endophytic fungus isolated from the mangrove *Ceriops tagal* collected in South China Sea [128]. Chemical assessment of the new species *Diaporthe vochysiae* sp. nov. (LGMF1583), isolated as endophyte of the medicinal plant *Vochysia divergens*, revealed two new carboxamides, vochysiamides A **394** (Figure 12) and B **395** (Figure 12) [129]. Two new eremophilane derivatives lithocarins B **396** (Figure 12) and **397** (Figure 12), were isolated from the endophytic fungus *Diaporthe lithocarpus* A740 [41]. Five new cytochalasans **398**–**402** (Figure 12) were isolated from the rice fermentation of fungus *Xylaria longipes* isolated from the sample collected at Ailao Moutain [130]. A new compound which was determined as 10-Ethylidene-2,4,9-trimethoxy-10,10a-dihydro-7,11-dioxa-benzo[b]heptalene-6,12-dione **403** (Figure 12) was isolated from *Penicillium citrinum* inhabiting *Parmotrema* sp. [131]. Investigation of the culture broth of *Periconia macrospinosa* KT3863 led to discover two new chlorinated melleins (3R,4S)-5-chloro-4-hydroxy-6-methoxymellein **404** (Figure 12), (R)-7-chloro-6-methoxy-8-O-methylmellein **405** (Figure 12) [132]. Two new compounds, lasdiplactone **406** (Figure 12) and lasdiploic acid **407** (Figure 12) were isolated from the chloroform extract of cell free filtrate of the endophytic fungus *Lasiosdiplodia pseudotheobromae*. The structure of **406** was characterized as (3S,4S,5R)–4–hydroxymethyl–3,5–dimethyldihydro–2–furanone [133]. Studies on the bioactive extract of mangrove endophytic fungus *Pleosporales* sp. SK7 led to the isolation of one new asterric acid derivative named methyl 2-(2-carboxy-4-hydroxy-6-methoxylphenoxy)-6-hydroxy-4-methyl-benzoate **408** (Figure 12) [23]. Chemical investigation of the mangrove-derived fungus *Aspergillus* sp. AV-2 following fermentation on solid rice medium led to the isolation of a new phenyl pyridazine derivative **409** (Figure 12) and a new prenylated benzaldehyde derivative, dioxoauroglaucin **410** (Figure 12) [134]. Three new furan derivatives, irpexlacte B–D **411**–**413** (Figure 12), were isolated from the endophytic fungus *Irpex lacteus* DR10-1 waterlogging tolerant plant *Distylium chinense*. Structures of compounds **411**–**413** were established as 5-(2α-hydroxypentyl) furan-2-carbaldehyde, 5-(1α-hydroxypentyl) furan-2-carbaldehyde, 5-(5-(2-hydroxypropanoyl) furan-2-yl) pentan-2-one, respectively [24]. Four new alkyl aromatics, penixylarins A–D **414**–**417** (Figure 12), were isolated from a mixed culture of the Antarctic deep-sea-derived fungus *Penicillium crustosum* PRB-2 and the mangrove-derived fungus *Xylaria* sp. HDN13-249. UPLC-MS data and an analysis of structural features showed that compounds **414** and **415** were produced by collaboration of the two fungi, while compounds **416**–**417** could be produced by *Xylaria* sp. HDN13-249 alone, but noticeably increased quantities by co-cultivation [135]. The co-culture of marine red algal-derived endophytic fungi *Aspergillus terreus* EN-539 and *Paecilomyces lilacinus* EN-531 induced the production of a new terrein derivative, namely asperterrein **418** (Figure 12) [136]. Fractionation and purification of the ethyl acetate extract of *Diaporthe lithocarpus*, an endophytic fungus from the leaves of *Artocarpus heterophyllus*, yielded one new compound, diaporthindoic acid **419** (Figure 12) [137]. A new diketopiperazine cyclo-(L-Phe-N-ethyl-L-Glu) **420** (Figure 12), was isolated from the cultures of an endophytic fungus *Aspergillus aculeatus* F027 [138]. Four novel compounds with g-methylidene-spirobutanolide core, fusaspirols A–D **421**–**424** (Figure 12), were isolated from the brown rice culture of *Fusarium solani* B-18. Compound **422** was found as the regioisomer of **421** [139]. One new polyacetylene glycoside **425** (Figure 12), one new brasilane-type sesquiterpenoid glycoside **426** (Figure 12), and two novel isobenzofuran-1(3H)-one derivatives **427**–**428** (Figure 12) were isolated from the solid culture of the endolichenic fungus *Hypoxylon fuscu* [86]. Chemical investigation of the crude extracts of both endophytic fungi *Simplicillium lanosoniveum* (J.F.H. Beyma) Zare & W. Gams PSU-H168 and PSU-H261 resulted in the isolation of three new compounds including two depsidones, simplicildones J and K **429**–**430** (Figure 12) and one dihydroxanthenone derivative, globosuxanthone E **431** (Figure 12) [96]. The apple juice supplemented solid rice media led to significant changes in the secondary metabolism of the endophytic fungus, *Clonostachys rosea* B5-2, and induced the production of four new compounds, (−)-dihydrovertinolide **432** (Figure 12), and clonostach acids A **433** (Figure 12), B **434** (Figure 12), and C **435** (Figure 12) [140]. Six new nonadride derivatives **436**–**441** (Figure 12) were isolated from the endophytic fungus *Talaromyces purpurogenus* obtained from fresh leaves of the toxic medicinal plant *Tylophora ovate* [87]. One new cyclic tetrapeptide, 18-hydroxydihydrotentoxin **442** (Figure 12), and a new amide, 6-hydroxyenamidin **443** (Figure 12) were obtained from the endophytic fungus *Phomopsis* sp. D15a2a isolated from the plant *Alternanthera bettzickiana* [55]. From an endophytic microorganism, *Aureobasidium pullulans* AJF1, harbored in the flowers of *Aconitum carmichaeli*, two unique lipid type new compounds (3R,5R)-3-(((3R,5R)-3,5-dihydroxydecanoyl)oxy)-5-hydroxydecanoic acid **444** (Figure 12), and (3R,5R)-3-(((3R,5R)-5-(((3R,5R)-3,5-dihydroxydecanoyl)oxy)-3-hydroxydecanoyl)oxy)-5-hydroxydecanoic acid **445** (Figure 12) were obtained [141]. The fungal strain *Alternaria alternata* JS0515 was isolated from *Vitex rotundifolia* (beach vitex). From the gungus one new altenusin derivative **446** (Figure 12), was isolated [142]. An investigation of a co-culture of the *Armillaria* sp. and endophytic fungus *Epicoccum* sp. YUD17002 associated with *Gastrodia elata* led to the isolation three aryl esters **447**–**449** (Figure 12) [27].

## 3. Biological Activity

### 3.1. Antimicrobial Activity

#### 3.1.1. Antifungal Activity

New polyketide-terpene hybrid metabolites **1** and **5** were tested for their inhibition activity following the NCCLS recommendations against six phytopathogenic fungi *Botrytis cinerea* (ACCC 37347), *Verticillium dahliae* (ACCC 36916), *Fusarium oxysporum* (ACCC 37438), *Alternaria solani* (ACCC 36023), *Fusarium gramineum* (ACCC 36249), and *Rhizoctonia solani* (ACCC36124) obtained from Agricultural Culture Collection of China (ACCC). The antifungal assay displayed that **1** and **5** exhibited pronounced biological effects against *F. oxysporum* with MIC (minimum inhibitory concentration) value of 8 g/mL, whereas **5** can potently inhibited *F. gramineum* at concentration of 8 g/mL, compared with the positive control ketoconazole (MIC value of 8 g/mL) [9].

Compounds **15**–**16** were evaluated for antifungal activities against six fungal strains, including Rhizoctonia solani, Verticillium dahliae Kleb, Helminthosporium maydis, Fusarium oxysporum, Botryosphaeria berengeriana and Colletotrichum acutatum Simmonds. Both compounds displayed moderate activities against three fungal strains Verticillium dahliae Kleb, Helminthosporium maydis, and Botryosphaeria dothidea with MIC values of 25–50 μg/mL [15].

The inhibitory activities of compounds **17**–**20** against four phytopathogenic fungi, including *Phytophthora infestane* (late blight), *Alternaria solani* (early blight), *Rhizoctonia solani* (black scurf), *Fusarium oxysporum* (blast), were evaluated. Compounds **17**–**20** all showed potent inhibitory activities toward *A. solani* and *F. oxysporum* with MIC value of 16 μg/mL, 32 μg/mL, 8 μg/mL, 8 μg/mL and 32 μg/mL, 16 μg/mL, 16 μg/mL, 16 μg/mL, respectively, while **19**–**20** weakly inhibited *P. infestans* and *R. solani* with MIC value of 128 μg/mL, 64 μg/mL and 128 μg/mL, 32 μg/mL, respectively. Hygromycin B was used as Positive control (MIC values of *P. infestans*, *A. solani*, *R. solani*, *and F. oxysporum* were 8 μg/mL, <4 μg/mL, 8 μg/mL, 64 μg/mL, respectively) [16].

Antifungal activity of compounds **50**–**55** against 14 plant-pathogenic fungi *Alternaria solani* QDAU-14 (AS), Bipolaris sorokiniana QDAU-7 (BS), *Ceratobasidium cornigerum* QDAU-8 (CC), *C. gloeosporioides* Penz QDAU-9 (CG), *Fusarium graminearum* QDAU-10 (FG), *F. oxysporum f.* sp. *cucumebrium* QDAU-16 (FOC), *F. oxysporum f.* sp. *momordicae* QDAU-17 (FOM), *F. oxysporum f.* sp. radicis lycopersici QDAU-5 (FOR), *F. solani* QDAU-15 (FS), *Glomerella cingulate* QDAU-2 (GC), *Helminthosporium maydis* QDAU-18 (HM), *Penicillium digitatum* QDAU-11 (PD), *P. piricola Nose* QDAU-12 (PP), and *Valsa mali* QDAU-13 (VM) were carried out by the microplate assay. Compound **50** exhibited inhibitory activity against the 13 test fungi with MIC values of 4 μg/mL (AS), 1 μg/mL (BS), 16 μg/mL (CC), 8 μg/mL (CG), 8 μg/mL (FG), 1 μg/mL (FOC), 2 μg/mL (FOM), 64 μg/mL (FOR), 4 μg/mL (FS), 1 μg/mL (GC), 8 μg/mL (PD), 4 μg/mL (PP), 16 μg/mL (VM), respectively, while compounds **50**–**55** showed activity against *Fusarium oxysporum f.* sp. cucumebrium with MIC values ranging from 1 to 64 μg/mL. **51** exhibited inhibitory activity against the 6 test fungi with MIC values of 32 μg/mL (AS), 8 μg/mL (BS), 32 μg/mL (FS), 4 μg/mL (GC), 8 μg/mL (PD), 4 μg/mL (PP), respectively. **52** exhibited inhibitory activity against the 4 test fungi with MIC values of 64 μg/mL (FOR), 1 μg/mL (GC), 8 μg/mL (PP), 32 μg/mL (VM), respectively. **53** exhibited inhibitory activity against the 3 test fungi with MIC values of 64 μg/mL (FOR), 16 μg/mL (PD), 1 μg/mL (PP), respectively. **54** exhibited inhibitory activity against Helminthosporium maydis with MIC value of 16 μg/mL. **55** exhibited inhibitory activity against *P. piricola* Nose with MIC values of 4 μg/mL (AS). Amphotericin B was used as the positive control against fungi with MIC values of 2 μg/mL (AS), 0.5 μg/mL (BS), 8 μg/mL (CC), 0.5 μg/mL (CG), 2 μg/mL (FG), 0.5 μg/mL (FOC), 1 μg/mL (FOM), 2 μg/mL (FOR), 4 μg/mL (FS), 0.5 μg/mL (GC), 2 μg/mL(HM), 2 μg/mL (PD), 2 μg/mL (PP), 8 μg/mL (VM), respectively [25].

Compounds **68**–**74** were assayed for their antifungal activities against *C. albicans*. Geneticin (G418), was used as positive control with the MIC value of 6.3 μg/mL. Compound **69** displayed inhibitory effect against *C. albicans* with an MIC value of 12.5 μg/mL, while compounds **68** and **74** exhibited weak inhibitory effect against *C. albicans* with MIC values of 100 μg/mL and 150 μg/mL [32].

Antifungal activities (Minimum inhibitory concentrations; MICs) of the isolated metabolite **170** were determined using a serial dilution assay against *Mucor hiemalis* DSM 2656. Compound **170** showed moderate to weak antifungal activity against *Mucor hiemalis* DSM 2656 with a MIC value of 33.33 μg/mL [52].

One fungus *Candida albicans* (ATCC 10231) was used for antifungal tests, the results showed that compound **177** exhibited significant antifungal activity against *C. albicans* with the MIC value of 2.62 μg/mL. The positive control for antifungal tests was used by ketoconazole with MIC value of 0.10 μg/mL [54].

The methylated dihydropyrone **189** and compound **274** were tested for in vitro antifungal activity using the Oxford diffusion assay against *M. violaceum* (*Microbotryum violaceum*) and *S. cerevisiae* (*Saccharomyces cerevisiae*), **189** and **274** exhibited moderate antifungal activity, inhibiting the growth of *S. cerevisiae* and *M. violaceum* at 25 μg/mL. Nystatin was the positive control for antifungal assays, previous studies had shown the MIC values of nystatin in the *S. cerevisiae* culture used was 4 µg/mL and for *M. violaceum* was 2 µg/mL [58].

Minimum Inhibitory Concentration (MIC) assays were used to assess antifungal activity of the compounds against anti-phytopathogenic activity against seven pathogenic fungi *Alternaria alternata* (Aa), *Botrytis cinerea* (Bc), *Cochliobolus heterostrophus* (Ch), *Colletotrichum lagenarium* (Cl), *Fusarium oxysporum* (Fo), *Gaeumannomyces graminis* (Gg), and *Thielaviopsis basicola* (Tb). Compound **227** showed potent and specific activity against 4 fungi with MIC values of 32 μg/mL(Bc), 16 μg/mL(Ch), 8 μg/mL(Fo), 8 μg/mL(Tb), respectively, whereas compound **228** showed moderate activity against 3 fungi with MIC values of 16 μg/mL(Bc), 32 μg/mL(Ch), 32 μg/mL(Fo) respectively. Prochloraz, a commercialized broad-156 spectrum fungicide widely used in agriculture, was used as positive antifungal control with MIC values of 8 μg/mL(Bc), 16 μg/mL(Ch), 8 μg/mL(Fo), 8 μg/mL(Tb), respectively. To the best of our knowledge, this is the first study to show that PIAs exhibit inhibitory activity against plant-pathogenic fungi [74].

Prochaetoviridin A **230** was evaluated for its antifungal activities against 5 pathogenic fungi *S. sclerotiorum*, *B. cinerea*, *F. graminearum*, *P. capsici* and *F. moniliforme* at the concentration of 20 µg/mL. It showed moderate antifungal activity with inhibition rates ranging from 13.7% to 39.0% [69].

Compounds **244** and **245** were evaluated against phytopathogenic fungi *Cladosporium cladosporioides* and *C. sphaerospermum* (*Cladosporium sphaerospermum*) using direct bioautography. The results showed that **244** exhibited antifungal activity, with a detection limit of 5 μg, for both fungi, while compound **245** displayed weak activity (detection limit > 5 μg), with a detection limit of 25 μg. Nystatin was used as a positive control, showing a detection limit of 1 μg [80].

Compound **266** was tested for antimicrobial activities against two plant-pathogenic fungi *Fusarium oxysporum* f. sp. *momordicae* nov. f. and Colletotrichum gloeosporioides, and exhibited potent activity against both strains with MIC values of 5 µM, which was close to that of the positive control, amphotericin B (MIC = 0.5 µM) [77].

Compounds **289**–**291** were assayed for antifungal activity against phytopathogenic fungi *M. grisea* and *F. verticillioides*, they showed evident inhibition of phytopathogenic fungi. The MIC values of compounds **289**–**291** were 200 μg/mL, 50 μg/mL and 50 μg/mL against *M. Grisea* and 200 μg/mL, 100 μg/mL and 100 μg/mL against *F. verticillioides*. Hygromycin B was the positive control against fungus with the MIC values of 50 μg/mL against both *M. Grisea* and *F. verticillioides* [83].

The purified metabolite **293** was tested for antimicrobial activity against selected pathogens namely *C. albicans*. Funiculosone (**293**) displayed antimicrobial activity inhibiting fungal pathogens. Funiculosone was able to inhibit the growth of *C. albicans* with an IC_50_ (50% inhibitory concentration) of 35 μg/mL [95].

Antifungal activity was determined against *C. neoformans* ATCC90113. The results showed that globosuxanthone E **294** displayed antifungal activity against *Cryptococcus neoformans* ATCC90113 with the MIC value of 32 μg/mL. Amphotericin B was used as a positive control for antifungal activity and exhibited an MIC value of 0.5 μg/mL [96].

The new compound, penochalasin K **343** was tested for its antifungal activity against four phytopathogenic fungi including *C. musae*, *C. gloeosporioides*, *P. italicm*, and *R. solani*. Compound **343** displayed excellent selective activities against the two phytopathogenic fungi *Colletotrichum gloeosporioides* (Penz) Sacc. (*C. gloeosporioides*), and *Rhizoctonia solani* Kühn *(R. solani*), with MIC values of 6.13 µM and 12.26 µM, respectively. Moreover, the activity towards *C. gloeosporioides* and *R. solani* were about ten-fold and two-fold better than those of the positive control carbendazim, respectively. Whereas only moderate or weak inhibitory activities were exhibited by compound **343** towards *Colletotrichum musae* (Berk. and M. A. Curtis) Arx. (*C. musae*) and *Penicillium italicum* Wehme (*P. italicm*). Carbendazim and the solvent were adopted as positive and negative control, respectively. The MIC values of Carbendazim against *C. gloeosporioides*, *R. solani*, *C. musae and P. italicm* were 65.38 µM, 32.69 µM, 32.69 µM and 16.34 µM [114].

The isolated compound **349** was evaluated for antifungal activities against *C. neoformans* and *P. marneffei*, it displayed weak antifungal activity against *C. neoformans* with MIC value of 32 μg/mL. Amphotericin B was used as positive control for fungi, displayed the MIC values of 1.0 μg/mL and 2.0 μg/mL against *C. neoformans* and *P. marneffei* [63].

Three fungi (*Aspergillus flavus*, *Fusarium oxysporum* and *Candida albicans*) were used in antifungal activity tests by disk diffusion method, the antifungal activity was recorded as clear zones of inhibition surrounding the disc (mm). Compound **362** showed antifungal activity against *F. oxysporum* (zone of inhibition was 6 mm) and variable activities against *A. flavus* and the yeast *C. albicans* (zone of inhibition was 5 mm). Nystatin (10 mg/disc) was used as standard antifungal (zone of inhibition against *A. flavus* and *F. oxysporum* were 12 mm and 17 mm) [116].

The antifungal activity against six commonly occurring plant-pathogenic fungi *Alternaria alternata*, *Cochliobolus heterostrophus*, *Gaeumannomyces graminis*, *Glomerella cingulata*, *Mucor hiemalis*, and *Thielaviopsis basicola* of compounds **364**–**365** were evaluated. Compounds **364** and **365** showed selective antifungal activity against *Mucor hiemalis* with minimum inhibitory concentration (MIC) values of 8 µg/mL and 4 µg/mL, respectively. Prochloraz was used as positive control with MIC value of 8 µg/mL against *Mucor hiemalis* [118].

In search for novel antifungal compounds, **368** and **369** were tested against *C. neoformans* and *C. gattii*. Compounds **368** and **369** exhibited moderate antifungal activities against *Cryptococcus neoformans* and *Cryptococcus gattii*, each with minimum inhibitory concentration values of 50.0 μg/mL and 250.0 μg/mL, respectively [120].

The antifungal activity of the compound **374** were evaluated against fungal strains *Phyllosticta citricarpa* LGMF06 and *Colletotrichum abscissum* LGMF1268 in order to select the best culture conditions to produce bioactive secondary metabolites. The isolated compound **374** displayed antifungal activity against the citrus phytopathogen *Phyllosticta citricarpa* with the inhibition zone of 30 mm. Amphotericin B was used as positive control with the inhibition zone of 37 mm [123].

The antifungal effect of **389** was assessed by agar disc diffusion assay towards *Candida albicans* (AUMC No. 418), *Geotrichium candidum* (AUMC No. 226), and *Trichophyton rubrum* (AUMC No. 1804) as fungi. It exhibited selective antifungal activity towards *C. albicans* (MIC 1.9 µg/mL and IZD 14.5 mm), comparing to the antifungal standard clotrimazole (MIC 2.8 µg/mL and IZD 17.9 mm), whilst, it had moderate activity against *G. candidum* (MIC 6.9 µg/mL and IZD 28.9 mm) [125].

Compound **418** was tested for antimicrobial activities against five plant-pathogenic fungi *A. brassicae*, *Colletotrichum gloeosprioides*, *Fusarium oxysporum*, *Gaeumannomyces graminis*, and *P. piricola*. It exhibited inhibitory activity against *A. brassicae* and *P. piricola* with the same MIC value of 64 µg/mL. The positive control against *A. brassicae* and *P. piricola* was amphotericin B with MIC values of 4 µg/mL and 8 µg/mL respectively [136].

Antifungal activity was determined against *C. neoformans* ATCC90113. Simplicildone K **430** and globosuxanthone E **431** displayed weak antifungal activity against *Cryptococcus neoformans* ATCC90113 with the same MIC values of 32 μg/mL. Amphotericin B was used as a positive control for antifungal activity and exhibited an MIC value of 0.5 μg/mL against *C. neoformans* ATCC90113 [96].

#### 3.1.2. Antibacterial Activity

The new compound **9** was evaluated for its antibacterial activities against *Mycobacterium tuberculosis*, *Staphylococcus aureus* (ATCC25923), *S. aureus* (ATCC700699), *Enterococcus faecalis* (ATCC29212), *E. faecalis* (ATCC51299), *E. faecium* (ATCC35667), *E. faecium* (ATCC700221) and *Acinetobacter baumannii* (ATCCBAA1605). It showed very weak inhibitory effect against *M. tuberculosis* (MIC > 50 µM) [12].

Compounds **15**–**16** were also evaluated for their antibacterial activity against twelve bacteria strains, including *Micrococcus lysodeikticus*, *Bacillus subtilis*, *Bacillus cereus*, *Micrococcus luteus*, *Staphyloccocus aureus*, *Bacillus megaterium*, *Bacterium paratyphosum B*, *Proteusbacillm vulgaris*, *Salmonella typhi*, *Pseudomonas aeruginosa*, *Escherichia coli*, and *Enterobacter aerogenes*. Compounds **15**–**16** displayed moderate activities against three bacterial strains (*Bacillus subtilis*, *Bacillus cereus* and *Escherichia coli*) with MIC values of 25–50 μg/mL [15].

Compounds **23**–**24**, **26** and **28** were evaluated for their antimicrobial activities against the Gram-positive strains *Staphylococcus aureus* ATCC 25923 and *Mycobacterium smegmatis* ATCC 607, Gram-negative strains *Escherichia coli* ATCC 8739 and *Pseudomonas aeruginosa* ATCC 9027, by the liquid growth inhibition in 96-well microplates. Compounds **23**–**24**, **26** and **28** displayed mild antibacterial activities against the Gram positive strain *Staphylococcus aureus* ATCC 25923 with IC_50_ values ranging from 31.5 to 41.9 µM [18].

New compounds **49**, **411**–**413** were evaluated for antibacterial activity against *P. aeruginosa* (CMCC(B)10,104). Compared with the positive control (Gentamicin, 0.18 µM), compounds **49**, **411**–**413** showed moderate activity with MIC values of 24.1 µM, 32.3 µM, 35.5 µM and 23.8 µM respectively [24].

Antimicrobial evaluation against one human pathogen *Escherichia coli* EMBLC-1 (EC), 10 marine-derived quatic bacteria *Aeromonas hydrophilia* QDIO-1 (AH), *Edwardsiella tarda* QDIO-2 (ET), *E. ictarda* QDIO-10, *Micrococcus luteus* QDIO-3 (ML), *Pseudomonas aeruginosa* QDIO-4 (PA), *Vibrio alginolyticus* QDIO-5, *V. anguillarum* QDIO-6 (VAn), *V. harveyi* QDIO-7 (VH), *V. parahemolyticus* QDIO-8 (VP), and *V. vulnificus* QDIO-9 (VV), was carried out by the microplate assay. Compound **50** showed activity with the same MIC value of 8 µg/mL against 4 bacteria ((EC) (AH) (PA) and (VH)) and the value of 4 µg/mL against *V. parahemolyticus*. Compound **51** showed activity with the MIC values of 16 µg/mL (EC), 8 µg/mL (PA) and 16 µg/mL (VH). Compound **52** showed activity with the MIC values of 8 µg/mL (EC), 8 µg/mL (AH), 4 µg/mL (PA), 2 µg/mL (VH) and 8 µg/mL (VP). While compound **55** had activity against aquatic pathogens *Edwardsiella tarda* and *Vibrio anguillarum* with MIC values of 1 μg/mL and 2 μg/mL, respectively, comparable to that of the positive control chloramphenicol (2 µg/mL (EC), 4 µg/mL (AH), 0.5 µg/mL (ET), 4 µg/mL (ML), 2 µg/mL (PA), 1 µg/mL (VAn), 1 µg/mL (VH), 4 µg/mL (VP), 1 µg/mL (VV)) [25].

Compounds **66** and **116** were evaluated for their antimicrobial activities against three human pathogenic strains (*Escherichia coli* ATCC 25922, *Staphyloccocus aureus* ATCC 25923 and *Candida albicans* ATCC 10231) by microbroth dilution method in 96-well culture plates. Bioassay results indicated that compound **116** displayed potent activity against *Staphyloccocus aureus* with an MIC value of 6.25 µM, which was equal to that of ampicillin sodium as a positive control, and compound **66** had a moderate inhibitory effect on *S. aureus* with an MIC value of 25.0 µM [30].

Compounds **70** and **74** were assayed for their antimicrobial activities against *S. aureus, B. cereus*, *B. subtillis*, *P. aeruginosa*, and *K. pneumonia*. The results showed that compounds **70** and **74** displayed weak antimicrobial effects with the same MIC value of 100 µg/mL against *B. subtillis* and *S. aureus*. Ampicillin was used as positive control with MIC values of 8 µg/mL and 3.5 µg/mL against *S. aureus* and *B. subtillis* [32].

Compound **119** was evaluated for antibacterial activities in vitro against Gram-Positive and Gram-Negative Bacteria (*Staphylococcus aureus* (DSM 799), *Escherichia coli* (DSM 1116), *Escherichia coli* (DSM 682), *Bacillus subtilis* (DSM 1088) and *Acinetobacter* sp. (DSM 586)). It was active against *Staphylococcus aureus* with an MIC value of 0.1 μg/mL. Streptomycin and Gentamicin were used as references against *Staphylococcus aureus* with MIC values of 5.0 μg/mL and 1.0 μg/mL, respectively. Comparison of **119** with **118** (>10.0 µg/mL against *Staphylococcus aureus*) and confirmed that the substitution at C-11 plays an important role in increasing the antibacterial activity against the selected bacterium [42].

The antibacterial activity of **157** and **159** was evaluated against five pathogenic bacteria of *Micrococcus tetragenus*, *Staphylococcus aureus*, *Streptomyces albus*, *Bacillus cereus*, and *Bacillus subtilis*. Compound **157** showed potent antimicrobial activity against *B. cereus* with the MIC value of 12.5 µg/mL, Compound **159** also showed potent antimicrobial activities against *B. subtilis*, *S. aureus*, and *S. albus* with the same MICs value of 12.5 µg/mL. Ciprofloxacin was used as a positive control with MIC values of 6.15 µg/mL, 5.60 µg/mL, 0.20 µg/mL, 1.50 µg/mL and 6.15 µg/mL against *M. tetragenus*, *B. cereus*, *B. subtilis*, *S. aureus* and *S. albus* [49].

The antimicrobial activity was determined by the paper disk diffusion method (100 μg compound in 8 mm paper disk), using meat peptone agar for *Staphylococcus aureus* and *Pseudomonas aeruginosa*, peptone yeast agar for *Candida albicans*, and potato dextrose agar for *Aspergillus clavatus*. **164** showed moderate antibacterial activity against *Staphylococcus aureus* NBRC 13276 (5: 24 mm) at a concentration of 100 μg/disk (MIC value: 3.2 μg/mL). Chloramphenicol was used for positive control against *S. aureus* (1 μg/mL) [50].

Antimicrobial activities (Minimum inhibitory concentrations; MICs) of the isolated metabolite **170** was determined using a serial dilution assay against *Bacillus subtilis* DSM 10, *Chromobacterium violaceum* DSM 30191, *Escherichia coli* DSM 1116, *Micrococcus luteus* DSM 1790, *Pseudomonas aeruginosa* DSM PA14, *Staphylococcus aureus* DSM 346, and *Mycobacterium smegmatis* DSM ATCC700084. Compound **170** showed moderate antibacterial activity against *Staphylococcus aureus* DSM 346 and *Bacillus subtilis* DSM 10, respectively, with a MIC value of 33.33 μg/mL. Oxytetracyclin was used as positive control with MIC values of 0.2 μg/mL and 4.16 μg/mL against *Staphylococcus aureus* DSM 346 and *Bacillus subtilis* DSM 10, respectively [52].

Antimicrobial tests were used for the disc diffusion method. Two Gram-positive methicillin-resistent *Staphylococcus aureus*, *Bacillus subtilis* (ATCC 6633), two Gram-negative *pseudomonas aeruginosa* (ATCC 9027), *Salmonella typhimurium* (ATCC 6539), were used. Compound **176** showed strong antibacterial activity against the *P. aeruginosa* and MRSA with the MIC values of 1.67 µg/mL and 3.36µg/mL, respectively. Compound **177** exhibited significant antibacterial activity against *B. subtilis* with the *MIC value* of 5.25 µg/mL. Positive control for antifungal tests were used by Ampicillin with the MIC values of 0.15 µg/mL, 0.15 µg/mL and 0.07 µg/mL against *P. aeruginosa*, MRSA (Methicillin-resistant *Staphylococcus aureus*) and *B. subtilis*, respetively. The results indicated that the methylester displayed improved biological activity and showed a selective antibacterial activity against *P. aeruginosa* and MRSA. Compound **176** exhibited more strong antimicrobial activity than compound **177** [54].

Antibacterial activity was determined against five pathogenic bacteria *Escherichia coli* (ATCC 25922), *Staphylococcus aureus* (ATCC 25923), *Bacillus cereus* (ATCC 11778), *Staphyloccocus epidermidis* (ATCC 12228) and *Staphylococcus albus* (ATCC 8799) by the microplate assay method. Compound **208** showed weak antibacterial activity against *Staphylococcus aureus* with a MIC value of 20 μg/mL. Ciprofloxacin was used as the positive control [66].

Antimicrobial activity testing of the compound **212** was carried out against a set of microorganisms using paper-disk diffusion assay. **212** exerted moderate-high activities (13 mm, 16 mm, 15 mm, 10 mm, 11 mm and 14 mm) against *Staphylococcus aureus*, *Pseudomonas aeruginosa*, *Candida albicans*, *Saccharomyces cerevisiae*, *Bacillus cereus* and *Bacillus subtilis* ATCC 6633. Gentamycin was used as positive control with the diameter of agar diffusion of 22 mm, 18 mm, 17 mm, 23 mm, 20 mm and 18 mm against the 5 bacteria as mentioned above [68].

Minimum Inhibitory Concentration (MIC) assays were used to assess antibacterial activity of the isolated compounds **227**–**228** against human pathogens (*Escherichia coli*, *Micrococcus luteus*, and *Pseudomonas aeruginosa*) and plant pathogen (*Ralstonia solanacearum*). Chloromycetin was used as a positive antibacterial control. Notably, compound **227** demonstrated potent activity against *P. aeruginosa* with an MIC value of 1 μg/mL, which was better than that of the positive control chloromycetin (MIC = 4 μg/mL). Compound **228** displayed activity against *Micrococcus luteus* and *Pseudomonas aeruginosa* with the same MIC value of 8 μg/mL (2 μg/mL and 4 μg/mL against *Micrococcus luteus* and *Pseudomonas aeruginosa* for Chloromycetin). In contrary to compounds **228** and the known compound **A** (Figure 13), **B** (Figure 13) showed stronger antibacterial activity (MIC values of 4, 4, 8, and 8 μg/mL against *E. coli*, *M. luteus*, *P. aeruginosa*, and *R. solanacearum*, respectively), indicating that hydroxylation at C-10 can augment antibacterial activity [74].

Compound **229** was tested for in vitro antimicrobial activity against 2 bacteria *B. subtilis* (ATCC 23857), and *E. coli* (ATCC 67878). Chloramphenicol was the antibacterial positive control. **229** showed modest antibiotic activity to *E. coli* with an MIC value of 100 µg/mL [58].

Antimicrobial activities were determined against four terrestrial pathogenic bacteria, including *Pseudomonas aeruginosa*, *Methicillinresistant Staphylococcus aureus*, *Bacillus subtilis* and *Escherichia coli* by the microplate assay method. Compound **231** exhibited modest antibacterial activity against *Escherichia coli* and *Pseudomonas aeruginosa* with 12.5 µg/mL, 50 µg/mL, respectively [75].

Antimicrobial activity was estimated by the inhibitory zone to five indicator microorganisms (*Bacillus subtilis* CMCC 63501, *Candida albicans* CMCC 98001, *Escherichia coli* CMCC 44102, *Pseudomonas aeruginosa* CMCC 10104 and *Staphylococcus aureus* CMCC 26003). Compounds **237** and **238** exhibited growth inhibitory activity against *E. coli* with MIC values of 32 µg/mL. Chloramphenicol was used as positive control with an MIC value of 4 µg/mL against *E. coli* [76].

Compound **241** was tested for antibacterial activity against *Bacillus subtilis* (ATCC 6633), *Staphylococcus aureus* (CGMCC 1.2465), *Streptococcus pneumoniae* (CGMCC 1.1692), *Escherichia coli* (CGMCC 1.2340), the results showed that **241** displayed modest antibacterial activity against *B. subtilis* with MIC value of 66.7 µM (the positive control gentamycin showed MIC value of 1.3 µM) [78].

Compound **246** was evaluated by the agar diffusion method against Gram-positive and Gram-negative bacteria, **246** showed moderate antibacterial activity against both *Pseudomonas aeruginosa* ATCC 15442 (13 mm) and *Staphylococcus aureus* NBRC 13276 (13 mm), respectively, at a concentration of 100 μg/disk [81].

Compounds **253**, **289**–**291** were assayed for their antibacterial activities against *Escherichia coli, Staphylococcus aureus,* and *Salmonella typhimurium*. All of the four compounds exhibited antibacterial activities against *Escherichia coli, Salmonella typhimurium,* and *Staphylococcus aureus* with the same MIC values of 25 µg/mL, 50 µg/mL and 25 µg/mL, respectively. Ampicillin was the positive control against bacteria, the MIC of ampicillin was lower than 0.78 μg/mL against *Salmonella typhimurium,* and *Staphylococcus aureus,* while the MIC value against *Escherichia coli* was 100 µg/mL [83].

The antimicrobial activity was determined by the paper disk diffusion method (100 μg compound in 8 mm paper disk), using meat peptone agar for *Staphylococcus aureus* and *Pseudomonas aeruginosa*. Comound **287** exhibited antibacterial activity against *S. aureus* and *P. aeruginosa* with MIC values (μg/mL) of >50 and 6.25. Chloramphenicol and kanamycin were used for positive control against *S. aureus* and *P. aeruginosa* (each 1 μg/mL), respectively [31].

Compound **293** was tested for antimicrobial activity against selected pathogens namely *S. aureus*, *E. coli* and *Pseudomonas aeruginosa* C. Gessard. Funiculosone (**293**) displayed antimicrobial activity inhibiting the bacterial pathogens. Funiculosone was able to inhibit the growth of *E. coli*, *S. aureus* and *C. albicans* with IC_50_ of 25 μg/mL and 58 μg/mL and 35 μg/mL respectively [95].

Compounds **295**–**296** were evaluated for antimicrobial activity against Gram-positive and Gram-negative bacteria. Compounds **295** and **296** showed moderate antibacterial activity against *S. aureus* NBRC 13276 and *P. aeruginosa* ATCC 15442 (MIC values of 6.3 µg/mL and 12.5 µg/mL for *S. aureus* NBRC 13276, 6.3 µg/mL and 6.3 µg/mL for *P. aeruginosa* ATCC 15442) [97].

Compounds **303**–**304** were evaluated for their antibacterial activities against six pathogenic bacteria including *M. tetragenus*, *S. aureus*, *S. albus*, *B. cereus*, *B. subtilis*, *E. coli*. Compound **303** showed antibacterial activity against *E. coli* with the MIC value of 6.25 μg/mL, and **304** exhibited a broad spectrum of antibacterial activities against six pathogenic with the MIC value ranging from 12.5 to 50 μg/mL (MIC values: 50 µg/mL for *M. tetragenus*, 25 µg/mL for *S. aureus*, >50 µg/mL for *S. albus*, 25 µg/mL for *B. cereus*, 12.5 µg/mL for *B. subtilis* and 50 µg/mL for *E. coli*). Ciprofloxacin was used as a positive control (MIC values: 0.313 µg/mL for *M. tetragenus* and *S. aureus*, 0.625 µg/mL for *S. albus*, *B. cereus*, *B. subtilis* and *E. coli*) [100].

The antibacterial activities of pure compound **309** was evaluated against Gram-positive bacteria *Staphylococcus aureus* and *Bacillus subtilis* and Gram-negative bacteria *Pseudomonas aeruginosa* and E*scherichia coli* using the disk diffusion assay. The new compound **309** showed inhibitory activity against *S. aureus* at 0.04 µg/paper disk, and the diameter of inhibition zone was 0.71 cm. The MIC for compound **309** against *S. aureus* was 100 µg/mL using the broth microdilution method, while streptomycin was employed as the positive control with an MIC of around 50 µg/mL [102].

Two Gram-positive *bacteria Bacillus subtilis* (ATCC6633) and *Staphylococcus aureus* ATCC (25923) were used. The antibacterial assay and the determination of the minimum inhibitory concentration (MIC) were determined according to continuous dilution method in the 96-well plates. Compound **313** showed antibacterial activity against *Bacillus subtilis* with an MIC value of 12.5 μg/mL. Ciprofloxacin was the positive control [104].

Compound **318** was tested for antibacterial activity against *Mycobacterium marinum* ATCCBAA-535. Although rifampin as positive control showed significantly in vitro antibacterial activity against *Mycobacterium marinum* ATCCBAA-535 with IC_50_ of 2.1 µM, compound **318** also exhibited potential inhibitory activity with IC_50_ of 64 µM [106].

The antibacterial activities of the isolated compounds **325**–**329** were evaluated against the soil bacterium *Acinetobacter* sp. BD4 (Gram–negative), the environmental strain of *Escherichia coli* (Gram–negative), as well as human pathogenic strains of *Staphylococcus aureus* (Gram–positive) and *Bacillus subtilis* (Gram–positive). The standard references employed were streptomycin (MIC values: 1.0 µg/mL against *Escherichia coli*, 10.0 µg/mL against *Acinetobacter* sp. BD4) and gentamicin (MIC values: 1.0 µg/mL against *Escherichia coli*, 5.0 µg/mL against *Acinetobacter* sp. BD4). Compounds **325**–**326** and **328**, demonstrated pronounced activity at 10.0 μg/mL against the soil bacterium *Acinetobacter* sp. BD4 comparable to streptomycin. Compounds **327** and **329** displayed antibacterial efficacies against *Escherichia coli* with the same MIC value of 5.0 µg/mL [109].

Antibacterial activity of the new compound **330** against *Vibrio parahaemolyticus* and *Vibrio anguillarum* was determined by the conventional broth dilution assay. **330** showed moderate inhibitory effects on *Vibrio parahaemolyticus* with an MIC value of 10 μg/mL. Ciprofloxacin was used as a positive control [110].

Antibacterial efficacies of the metabolite **339** were determined by serial dilution assay. Compound **339** showed strong activity against *Bacillus subtilis* and *Micrococcus luteus* with MIC values of 8.33 µg/mL and 16.66 µg/mL, respectively, while the MIC values of Oxytetracyclin used as the positive control against *Bacillus subtilis* and *Micrococcus luteus* were 4.16 µg/mL and 0.40 µg/mL, respectively. While the MIC value of compound **C** (Figure 13) against *Mucor hiemalis* (16.66 µg/mL) was the same as that of nystatin used as positive control. The two active metabolites are anthranilic acid derivatives with a phenylethyl core. Since metabolite **340**, which contains a phenylmethyl group instead of a phenylethyl residue, was not active, it was concluded that the phenylethyl moiety in compounds **339** and **C** is essential for their antimicrobial activity [60].

The isolated compound **347**, which was obtained in sufficient amounts, was evaluated for antimicrobial activities against *S. aureus* ATCC25923 and methicillin-resistant *S. aureus*. Simplicildone A **347** displayed weak antibacterial against *Staphylococcus aureus* with MIC value of 32 µg/mL. Vancomycin which was used as positive control for bacteria, displayed the MIC values of 0.5 µg/mL and 1.0 µg/mL against both *S. aureus* and methicillin-resistant *S. aureus* [63].

The antimicrobial activity of compound **367** was evaluated using the strains of methicillin-resistant *Staphylococcus aureus*, *Klebsiella pneumoniae*, *Pseudomonas aeruginosa*, *Bacillus subtilis*, and *Escherichia coli*. Compound **367** exhibited weaker activity in comparison to the positive control tetracycline against methicillin-resistant *S. aureus* (MRSA) with the MIC value of 128 µg/mL, and against *K. pneumoniae* and *P. aeruginosa* with equal MIC values of 32 µg/mL [119].

Compounds **371**–**373** were assayed for their antimicrobial activities against *Staphylococcus aureus*, *Bacillus subtilis*, *Pseudomonas aeruginosa*, *Klebsiella pneumonia* and *Escherichia coli*. Compounds **371**–**372** exhibited significant inhibitory activities against *B. subtilis* and *S. aureus* with MIC values of 15 µg/mL and 18 µg/mL, respectively. Compound **373** showed moderate inhibitory activities against *B. subtilis* (MIC 35 µg/mL) and *S. aureus* (MIC 39 µg/mL). Ampicillin (MIC values: 8 µg/mL, 3.5 µg/mL, 10 µg/mL, 10 µg/mL and 2.5 µg/mL against the 5 bacteria mentioned above) and kanamycin (MIC values: 4 µg/mL, 1.0 µg/mL, 8 µg/mL, 9 µg/mL and 4 µg/mL against the 5 bacteria mentioned above) served as the positive control. In addition, morphological observation showed the rod-shaped cells of *B. subtilis* growing into long filaments, which reached 1.5- to 2-fold of the length of the original cells after treatment with compounds **371**–**372**. The coccoid cells of *S. aureus* exhibited a similar response and swelled to a 2-fold volume after treatment with compounds **371**–**372** [122].

The antimicrobial activity of the compound **374** was evaluated against the Gram-positive bacteria *Staphylococcus aureus* (ATCC 25923), methicillin-resistant *Staphylococcus aureus* (MRSA) (BACHC-MRSA). The resulting inhibition zones were measured in millimeters. **374** displayed antibacterial activity against sensitive and resistant *S. aureus*, the diameter of inhibition zone was 14 mm, Ampicillin was antibacterial control with the diameter of inhibition zone of 30 mm [123].

Compounds **388** was tested for its antimicrobial activities against *Escherichia coli* ATCC 25922, *Staphyloccocus aureus* ATCC 25923, *Staphylococcus epidermidis* ATCC 12228, and *Mycobacterium Smegmatis* MC 2155 ATCC70084. Compound **388** was active against *Escherichia coli* ATCC 25922 and *Staphyloccocus aureus* ATCC 25923 with MIC values of 32 µg/mL and 64 µg/mL, respectively. Levofloxacin was used as a positive control with MIC value of 0.12 µg/mL [124].

Fusarithioamide B **389** has been assessed for antibacterial activities towards various microbial strains (*Staphylococcus aureus* (AUMC No. B-54) and *Bacillus cereus* (AUMC No. B-5) as Gram-positive bacteria, *Escherichia coli* (AUMC No. B-53), *Pseudomonas aeurginosa* (AUMC No. B-73), and *Serratia marscescens* (AUMC No. B-55) as Gram-negative bacteria) by disc diffusion assay. It possessed high antibacterial potential towards *E. coli* (Inhibition zone diameter (IZD): 25.1 ± 0.60 mm, MIC value: 3.7 ± 0.08 µg/mL), *B. cereus* (Inhibition zone diameter (IZD): 23.0 ± 0.36 mm, MIC value: 2.5 ± 0.09 µg/mL), and *S. aureus* (Inhibition zone diameter (IZD): 17.4 ± 0.09 mm, MIC value: 3.1 ± 0.11 µg/mL) compared to ciprofloxacin used as antibacterial standard (Inhibition zone diameter (IZD): 15.3 ± 0.07 mm, MIC value: 3.4 ± 0.32 µg/mL for *S. aureus*, Inhibition zone diameter (IZD): 21.2 ± 0.51 mm, MIC value: 2.9 ± 0.20 µg/mL for *B. cereus*, Inhibition zone diameter (IZD): 25.6 ± 0.22 mm, MIC value: 3.9 ± 0.06 µg/mL for *E. coli*) [125].

The new compounds were evaluated for their antibacterial activities against five terrestrial pathogenic bacteria, including *S. aureus* (ATCC 27154), *Staphylococcus albus* (ATCC 8799), *B. cereus* (ATCC 11778), *Escherichia coli* (ATCC 25922), and *Micrococcus luteus* (ATCC 10240) by the microplate assay method. The result showed that Compounds **392**–**393** showed moderate antibacterial activities against *Staphylococcus aureus* with the MIC values of 25.0 µg/mL and 12.5 µg/mL, respectively. Ciprofloxacin was used as positive control with the MIC value of 0.39 µg/mL [128].

The MIC of compound **395** against *Staphylococcus aureus* (MSSA), Methicillin resistant *Staphylococcus aureus* (MRSA) and *Klebsiella pneumoniae carbapenemase-producing* (KPC) was performed. Vochysiamide B **395** displayed considerable antibacterial activity against the Gram-negative bacterium *Klebsiella pneumoniae* (KPC), a producer of carbapenemases, MIC of 80 μg/mL in comparison with positive controls meropenem and gentamicin with MIC values of 45 μg/mL and 410 μg/mL against KPC [129].

The antimicrobial activities of compounds were tested against six microorganisms by the microdilution method, including *Mycobacterium phlei*, *Bacillus subtilis*, *Vibrio parahemolyticus*, *Escherichia coli*, *Pseudomonas aeruginosa*, and *Proteus vulgaris*. Among them, compound **416** showed promising activity against *M. phlei* with the same MIC values as positive control ciprofloxacin of 12.5 µM, which indicated the antituberculosis potential. Compound **415** showed activities against *B. subtilis* with MIC value of 100 µM. Compound **416** showed activities against *M. phlei* with MIC value of 6.25 µM. Ciprofloxacin was positive control shared same MIC values of 1.56 µM against *Mycobacterium phlei* and *Bacillus subtilis* [135].

Compound **418** was tested for antimicrobial activities against two human pathogens (*E. coli* and *S. aureus*), seven aquatic bacteria (*Aeromonas hydrophila*, *Edwardsiella tarda*, *Micrococcus luteus*, *Pseudomonas aeruginosa*, *Vibrio alginolyticus*, *Vibrio harveyi*, and *Vibrio parahaemolyticus*). Compound **418** exhibited inhibitory activity against *E. coli*, and *S. aureus* with same MIC values of 32 µg/mL. Positive control was chloramphenicol which with MIC values of 2 µg/mL and 1 µg/mL against *E. coli*, and *S. aureus* [136].

Antibacterial activity was evaluated against *S. aureus* and methicillin-resistant *S. aureus*. Simplicildone K **430** exhibited antibacterial activity against *Staphylococcus aureus* and methicillin-resistant *S. aureus* with equal MIC values of 128 μg/mL. Vancomycin was used as a positive control for antibacterial activity and displayed equal MIC values of 0.5 μg/mL against both *S. aureus* and methicillin-resistant *S. aureus* [96].

#### 3.1.3. Antiviral Activity

Anti-enterovirus 71 (EV71) was assayed on Vero cells with the CCK-8 (DOjinDo, Kumamoto, Japan) method. The 50% inhibitory concentration (IC_50_) of the testing compound was calculated using the GraphPad Prism software. Ribavirin was used as the positive control with an IC_50_ value of 177.0 µM. Vaccinol J **125** exhibited in vitro anti-EV71 with IC_50_ value of 30.7 µM, and the inhibition effect was stronger than positive control ribavirin [44].

Anti-HIV activities of compound **150** was tested in vitro by HIV-I virus-transfected 293 T cells. At the concentration of 20 μM, **150** showed a weak inhibitory rate of 16.48 ± 6.67%. Efavirenz was used as the positive control, with an inhibitory rate of 88.54 ± 0.45% at the same concentration [47].

### 3.2. Cytotoxic Activity or Anticancer

Nectrianolins A–C **11**, **12**, and **13** were evaluated for their in vitro cytotoxicity against HL60 (human leukemia 60) and HeLa cell lines by the MTT method using a published protocol. Compounds **11**, **12**, and **13** exhibited cytotoxic activity against the HL60 cell line with IC_50_ values of 1.7 µM, 1.5 µM and 10.1 µM, respectively. Additionally, compounds **11**, **12**, and **13** exhibited cytotoxicity against the HeLa cell line with IC_50_ values of 34.7 µM, 16.6 µM and 52.1 µM, respectively [13].

Compounds **29** and **236** were evaluated for their cytotoxic activities against three human tumor cell lines HeLa, HCT116 (Human colon cancer tumor cells), and A549 (Human lung cancer cells), both of them exhibited weak to moderate cytotoxic activities with IC_50_ values ranging from 21.09 to 55.43 µM (**29**: 58.75 ± 1.77 µM, 47.75 ± 1.68 µM, 29.58 ± 1.47 µM, **236**: 21.18 ± 1.33 µM, 21.04 ± 1.32 µM, 37.33 ± 1.57 µM against HeLa, HCT116 and A549 respectively) [19].

The cytotoxic activity of the isolated compounds **78**–**79** and **113**–**114** were tested against Hela cells. Compound **79** showed weak cytotoxic activities against Hela cells with IC_50_ value of 43.7 ± 0.43 µM. Compound **78** did not show significant cytotoxic activity. As the oxoindoloditerepene epimers, the 3α-epimer **79** was clearly more cytotoxic than the 3β-epimer **78**, suggesting that their cytotoxic activity depended on their stereochemistry. The acetoxy derivatives **113** and **114** showed weak cytotoxic activities against Hela cells with IC_50_ values of 83.8 ± 5.2 µM and 53.5 ± 2.1 µM respectively [35].

Since many triterpenoids isolated from plants of the family Schisandraceae are reported to reduce the risk of liver diseases and cancer, compounds **93**–**100** were evaluated for in vitro cytotoxicity against human hepaticellular liver carcinoma cell (HepG2), according to the MTT method, with cisplatin as the positive control (IC_50_ value of 9.8 ± 0.21 µM). Compounds **93**–**100**, showed moderate cytotoxic activity with IC_50_ values ranging from 14.3 to 21.3 µM (IC_50_ values of compounds **93**: 15.6 µM, **94**: 16.1 µM, **95**: 16.4 µM, **96**: 15.4 µM, **97**: 17.9 µM, **98**: 18.8 µM, **99**: 14.3 µM, **100**: 21.3 µM). It should be noted that those metabolites **93**–**100** produced during fermentation showed stronger cytotoxicity to HepG2 cell line than that of nigranoic acid, the main component of non-fermented *K. angustifolia* [39].

The in vitro cytotoxicity of compound **119** against the human acute monocytic leukemia cell line (THP-1) was evaluated using a resazurin-based assay and an ATPlite assay. Compound **119** demonstrated marked cytotoxicity against the human acute monocytic leukemia cell line (THP-1) with the IC_50_ value of 8.0 µM [42].

The in vitro cytotoxicity assay was performed with some cancer cells including the mouse fibroblast cell line L929, cervix carcinoma cell line KB-3-1, human breast adenocarcinoma MCF-7, human prostate cancer PC-3, squamous carcinoma A431, human lung carcinoma A549 and ovarian carcinoma SKOV-3. Compounds **169**, **170** showed significant cytotoxicity against the mouse fibroblast cell line L929 and the cervix carcinoma cell line KB-3-1, with IC_50_ values ranging from 6.3 to 23 µg/mL (**169**: 23 µg/mL against the mouse fibroblast cell line L929, 22 µg/mL against cervix carcinoma cell line KB-3-1 **170**: 6.3 µg/mL against the mouse fibroblast cell line L929, 11 µg/mL against cervix carcinoma cell line KB-3-1). Compound **170** showed the strongest cytotoxicity among the metabolites tested against human breast adenocarcinoma MCF-7 cells with IC_50_ value of 1.5 µg/mL. Besides, compound **170** showed cytotoxicity against squamous carcinoma A431, human lung carcinoma A549 and ovarian carcinoma SKOV-3 with IC_50_ values of 6.5 µg/mL, 16 µg/mL and 6.5 µg/mL. Epothilon B was used as positive control (IC_50_ values against 7 cancer cells mentioned above were 0. 8 ng/mL, 0. 06 ng/mL, 0.04 ng/mL, 1.1 ng/mL, 0.1 ng/mL, 2 ng/mL and 0.12 ng/mL) [52].

Standard MTT assays employing MDA-MB-435 and A549 cell lines were performed. The IC_50_ was determined by a 50% reduction of the absorbance in the control assay. Compound **176** exhibited cytotoxicity against MDA-MB-435 and A549 cell lines with IC_50_ values of 16.82 and 20.75 µM, respectively. The positive control was used by Epirubicin (EPI) with IC_50_ values of 0.26 and 5.60 µM against MDA-MB-435 and A549 cell lines [54].

All isolated new compounds **190**–**194** were evaluated for their cytotoxic activities against various cancer cell lines, which include A549, Raji, HepG2, MCF-7, HL-60 and K562. Compounds **190**–**194** displayed in vitro inhibitory activities against the six tumor cell lines to various degrees. Among them, compound **192** showed the most potent cytotoxicity against all evaluated cell lines with IC_50_ values of 1.2, 2.0, 1.6, 2.2, 1.0 and 1.2 μg/mL, respectively, which were even stronger than an anti-tumor agent DDP used as positive control (IC_50_ values against six cell lines: 2.8 μg/mL, 2.1 μg/mL, 2.6 μg/mL, 2.4 μg/mL, 2.1 μg/mL and 2.2 μg/mL). Compounds **193** and **194** also exhibited moderate growth inhibition against six tested cell lines with IC_50_ 6.3–26.8 μg/mL for **193** and IC_50_ 3.1–24.4 μg/mL for **194**. However, compounds **190** and **191** were effective only against HL-60 and K562 cell lines (IC_50_ value: **190**: 24.1 μg/mL, 10.7 μg/mL **191**: 24.2 μg/mL, 23.1 μg/mL). These results indicated that the keto or hemiketal functionality (e.g., **192**–**195**) would play an important role in cytotoxic activity. Additionally, the activity profile reflected that the hydroxyl-substituted position had a different impact on cytotoxic activity. 2-Pyrones were more active as cytotoxic agents if the alkyl chain at C-6 was oxygenated but the addition of the hydroxyl subunit to C-8 and C-9 significantly decreased the activity [59].

The isolated compound **202** was preliminary evaluated for its cytotoxicities against MCF-7, NCI-H460, HepG-2, and SF-268 cell lines with cisplatin as the positive control. The new compound **202** exhibited weak growth inhibitory activity against the tumor cell lines MCF-7 and HepG-2 with IC_50_ values of 70 and 60 µM, respectively [64].

Cytotoxic activities of compound **209** against HeLa, MCF-7 and A549 cell lines were evaluated by the MTT method. Adriamycin was used as a positive control. The results showed that **209** displayed cytotoxic activity against A549 cell lines with IC_50_ value of 15.7 μg/mL [66].

Compound **221** was assessed for its antiproliferative activities against the mouse lymphoma (L5178Y) cell line using the in vitro cytotoxicity (MTT) assay and kahalalide F as a standard antiproliferative agent (IC_50_ = 4.30 µM). Results revealed the new compound, aflaquinolone H (**221**), exhibited moderate antiproliferative activity (IC_50_ = 10.3 µM) which highlights the role of the hydroxyl group at C-21 for the antiproliferative activity [71].

Compounds **222**–**223** were evaluated for in vitro inhibition of cell proliferation by the MTT method using a panel of four human cancer cell lines: NCI-H460 (non-small cell lung cancer), SF-268 (CNS glioma), MCF-7 (breast cancer), and PC-3 (prostate adenocarcinoma) cells. Compounds **222** and **223** showed moderate cytotoxicity against four human cancer cell lines with IC_50_ values of 18.63 ± 1.82, 20.23 ± 2.15, 23.53 ± 2.33 and 20.48 ± 2.04 µM, and 16.47 ± 1.63, 17.57 ± 2.12 20.79 ± 2.39 and 19.43 ± 2.02 µM, respectively, while compound **D** (Figure 13) was found to be inactive (>50 µM), which suggested -NH_2_ group might play a very important role for their cytotoxicity. Doxorubicin (Adriamycin) was used as positive control in this assay (IC_50_ values against the 4 human cancer cell lines: 0.43 ± 0.12 µM, 0.61 ± 0.09 µM, 0.41 ± 0.11 µM and 0.25 ± 0.08 µM respectively) [72].

Compound **241** was also tested for cytotoxicity against SH-SY5Y (human glioma cell lines), HeLa (cervical epithelial cells), HCT116 (human colon cancer cells), HepG2 (human hepatocellular carcinoma cells), A549 (human lung cancer cells), and MCF7 (human breast cancer cells). Compound **241** showed weak cytotoxic effects against HeLa cells with IC_50_ value of 97.4 μM, while the positive control cisplatin showed IC_50_ value of 21.1 µM [78].

The cytotoxicity of compound **244** against a human cervical tumor cell line (HeLa) was tested using the MTT assay. Compound **244** presented an IC_50_ value of 100 μmol/L. Camptothecin was used as positive control and presented an IC_50_ of 0.12 μmol/L [80].

The cytotoxicities against HBE, THLE, and MDA-MB-231 of compound **252** were evaluated by MTT method. **252** exhibited selective cytotoxicities against MDA-MB-231 with IC_50_ of 24.6 ±1.3 µg/mL [82].

Compounds **262**,**426**–**427** were evaluated for their cytotoxicity against a human leukemia cell line (K562), a colon adenocarcinoma cell line (SW480), and a human liver carcinoma cell line (HepG2). Compounds **262** and **427** showed moderate cytotoxic activity against all the tested cell lines with IC_50_ ranging from 12.0 to 28.3 µM (IC_50_ values against K562, SW480, and HepG2 cells: **262**: 15.9 (13.1–19.3) µM, 12.0 (8.8–16.4) µM, 28.3 (23.2–34.6) µM **427**: 20.6 (14.0–30.3) µM, 20.3 (16.8–24.4) µM, 20.4 (16.4–25.4) µM). In addition, compound **426** showed moderate cytotoxicity towards K562 cells with an IC_50_ value of 18.7 µg/mL. Cisplatin was used as the positive control with IC_50_ values of 3.8, 5.5, and 6.8 µM toward K562, SW480, and HepG2 cells, respectively [86].

Compound **281** was evaluated cytotoxic activities against three cancer cell lines HCT 116, HeLa, and MCF7, and displayed strong biological effect against MCF7 with halfmaximal inhibitory concentration (IC_50_) value at 7.73 ± 0.11 µM compared with the cis-platinum (14.32 ± 1.01 µM) [91].

The isolated compound **287** was examined for cytotoxic activity by MTT assay. Camptothecin was used as positive control for HL60 with IC_50_ = 23.6 nM. **287** exhibited cytotoxicity against human promyelocytic leukemia HL60 cells with IC_50_ value of 1.33 µM. The higher cytotoxicity of **287** and **E** (Figure 13) compared to that of the related compounds **F** (Figure 13) and **G** (Figure 13) was attributed to their increased cell membrane permeability due to the presence of the hydroxyl group [69].

Compound **288** was investigated for its cytotoxicities against SMMC-7721 cell by MTT method. The results showed that **288** inhibited SMMC-7721 cells proliferation in a dose-dependent manner (100 µM, 50 µM, 25 µM, 12.5 µM, 6.25 µM), with IC_50_ of 61 + 2.2 µM [31].

The cytotoxicities of compound **297** were tested by using human promyelocytic leukemia HL-60, human hepatoma SMMC-7721, non-small cell lung cancer A-549, breast cancer MCF-7 and human colorectal carcinoma SW4801 cell lines, **297** showed cytotoxicity against MCF-7 with the ratio of inhibition at 72% for a concentration at 40 µM (IC_50_ of positive control Taxol < 0.008 µM) [98].

The cytotoxicities of compound **311** were evaluated against the A549 and HepG2 cell lines by the MTT method. Newly isolated compound **311** showed weak activities with IC_50_ values of 11.05 µM and 19.15 µM, respectively, against the tested cell lines. Doxorubicin was used as a reference (0.94 µM and 1.16 µM) [103].

The obtained compound **320** was evaluated for its cytotoxic activities against A549 human lung cancer cells and HepG2 human liver cancer cells. Compound **320** exhibited potent cytotoxic activities towards A549 human lung cancer cells and HepG2 human liver cancer cells with IC_50_ values of 23.73 ± 3.61 µM and 35.73 ± 2.15 µM, respectively [90].

The anti-tumor activities of compounds **336**–**337** were evaluated against Ramos and H1975 cell lines. **337** displayed the most promising anti-tumor activity against both Ramos and H1975 cell lines with IC_50_ values of 0.018 µM and 0.252 µM, respectively. Compound **337** may be more effective in anti-tumor activity against Ramos and H1975 than stand drug Ibrutinib and afatinib, with IC_50_ values of 28.7 µM and 1.97 µM. These findings suggest that compound **337** might be promising lead for leukemia and lung cancer treatments. In addition, **336** also displayed anti-tumor activity against both Ramos and H1975 cell lines with IC_50_ values of 17.98 and 7.3 µM, respectively [113].

Compound **343** was evaluted for the cytotoxicities against three human tumor cell lines, including a human breast cancer cell line (MDA-MB-435), a human gastric cancer cell line (SGC-7901), and a human lung adenocarcinoma epithelial cell line (A549) by MTT method. It is notable that penochalasin K **343** exhibited remarkable broad-spectrum inhibitory activities against all the tested cell lines (IC_50_ values against MDA-MB-435, SGC-7901 and A549: 4.65 ± 0.45 µM, 5.32 ± 0.58 µM and 8.73 ± 0.62 µM). Epirubicin was used as a positive control with IC_50_ values of 0.56 ± 0.06 µM, 0.37 ± 0.11 µM and 0.61 ± 0.05 µM against MDA-MB-435, SGC-7901 and A549 [114].

The cytotoxicity was evaluated by the [3H] thymidine assay using breast cancer (MCF-7) and colon cancer (COLO-205) cell lines. Doxorubicin (10 µg), was used as a positive control with ED_50_ (50% effective dose) value of 1.8 µg/mL against MCF-7 cell line. Compound **362** showed cytotoxic activity against MCF-7 cell line with ED_50_ value of >10 µg/mL [116].

Compound **363** was evaluated for its cytototoxicity against different cancer cell lines MOLT-4, A549, MDA-MB-231and MIA PaCa-2 by MTT assay. Interestingly, compound **363** showed considerable cytotoxic potential against the human leukaemia cancer cell line (MOLT-4) with IC_50_ value of 20 µmol/L, it was not as active as the positive control flavopiridol (IC_50_ value of 0.2 µmol/L) [117].

Cytotoxicity against four tumor cell lines (A549, HeLa, MCF-7, and THP-1) of compound **365** was evaluated. In the cytotoxic assay, compound **365** displayed weak in vitro cytotoxicity against the THP-1 cell line, with IC_50_ value of 40.2 µM [118].

The cytotoxic effect of **389** was evaluated in vitro towards ovarian (SK-OV-3), epidermoid (KB), malignantmelanoma (SK-MEL), human breast adenocarcinoma (MCF-7), colorectal adenocarcinoma (HCT-116), and ductal (BT-549) carcinomas. Doxorubicin (positive control) and DMSO (negative control) were used. It had selective and potent effect towards BT-549, MCF-7, SKOV-3, and HCT-116 cell lines with IC_50_s 0.09 ± 0.05, 21 ± 0.07, 1.23 ± 0.03, and 0.59 ± 0.01 µM, respectively, compared to doxorubicin (IC_50_s 0.045 ± 0.11, 0.05 ± 0.01, 0.321 ± 0.21, and 0.24 ± 0.04 µM, respectively). Fusarithioamide B (**389**) may provide a lead molecule for future developing of antitumor and antimicrobial agents [125].

In the cancer cell line cytoxicity assays, compound **395** displayed low activity against human non-small cell lung A549 and human prostate PC3 cell lines (A549: EC_50_ (concentration for 50% of maximal effect) = 86.4 μM for 395, PC3: EC_50_ = 40.25 μM for **395**. 1.5 mM hydrogen peroxide was used as positive control (100% dead cells), 0.1% dimethyl sulfoxide was used as negative control (100% live cells) [129].

Compounds **396**–**397** were evaluated for their cytotoxic activity against four human tumor cell lines (SF-268, MCF-7, HepG-2 and A549) by the SRB (Sulforhodamine B) method. As a result, compounds **396**, **397** showed weak inhibitory activities against the four tumor cell lines with IC_50_ values ranging from 30 to 100 µM (IC_50_ values against SF-268, MCF-7, HepG-2 and A549 **396**: 41.68 ± 0.88 µM, 37.68 ± 0.3 µM, 48.33 ± 0.1 µM and 53.36 ± 0.91 µM, **397**: 69.46 ± 7.08 µM, 97.71 ± 0.72 µM, 79.43 ± 0.63 µM and 0 ≥ 100 µM). Cisplatin was used as a positive control with IC_50_ values of 3.39 ± 0.29 µM, 3.19 ± 0.12 µM, 2.42 ± 0.14 µM and 1.56 ± 0.08 µM against the four human tumor cell lines [41].

The in vitro cytotoxicity assay was performed according to the MTS method in 96-well microplates. Five human tumor cell lines were used: human myeloid leukemia HL-60, human hepatocellular carcinoma SMMC-7721, lung cancer A-549, breast cancer MCF-7, and human colon cancer SW480, which were obtained from ATCC (Manassas, VA, USA). Cisplatin was used as the positive control for the cancer cell lines (IC_50_ values against HL-60, A-549, SMMC-7721, MCF-7, and SW480 cell: 4.05 ± 0.11, 19.40 ± 0.71, 14.91 ± 0.36, 22.96 ± 0.58 and 23.15 ± 0.22 μM). Compound **447** demonstrated moderate cytotoxicity against HL-60, A-549, SMMC-7721, MCF-7, and SW480 cell with IC_50_ values of 15.80, 15.93, 19.42, 19.22, and 23.03 μM, respectively [27].

### 3.3. Other Activities

α-Glucosidase inhibitors are helpful to prevent deterioration of type 2 diabetes and for the treatment of the disease in the early stage, so the α-glucosidase inhibitory effects of the isolated compounds were evaluated. As a result, compounds **247**, **248** exhibited potent α-glucosidase inhibitory activity with IC_50_ values of 25.8 µM, 54.6 µM, respectively, which were much better than acarbose (IC_50_ of 703.8 µM) as a positive control. Compounds **7** and **249** showed moderate inhibitory activity against α-glucosidase with IC_50_ values of 188.7 µM and 178.5 µM, respectively. The results indicated that the configureuration at C-5 in compounds **6** and **7** might affect α-glucosidase inhibitory activity. Moreover, the methoxy group at C-15 in the lasiodiplodin derivatives decreased the activity (**248** vs. **H** (Figure 13)). For compounds **247**, **I** (Figure 13), **J** (Figure 13), and **K** (Figure 13), compounds **247** and **I** showed potent α-glucosidase inhibitory effects, whereas **J** and **K** were inactive, which attested that the position of the hydroxyl group had a significant impact on the activity [10].

AChE inhibitory activities of the compound **14** were assayed by the spectrophotometric method. Compound **14** indicated anti-AChE activity with inhibition ratio at 35% in the concentration of 50 μM. Tacrine (Sigma, purity > 99%) was used as a positive control of inhibition ratio at 52.63% with the concentration of 0.333 μM [14].

The inhibition of the marine phytoplankton *Chattonella marina*, *Heterosigma akashiwo*, *Karlodinium veneficum*, and *Prorocentrum donghaiense* by **31**–**37** were assayed. The results showed that **32**–**34** were more active to *C. marina*, *K. veneficum*, and *P. donghaiense* than **31** and **35**–**37** (IC_50_ against *C. Marina*, *H. akashiwo*, *K. veneficum* and P. donghaiense: **31**: 11, 4.6, 12 and 23 μg/mL **32**: 1.2, 4.3, 1.3 and 5.7 μg/mL **33**: 3.3, 9.2, 1.5 and 6.8 μg/mL **34**: 0.93, 7.8, 2.7 and 4.9 μg/mL **35**: 6.7, 2.9, 6.6 and 10 μg/mL **36**: 5.4, 5.8, 8.4 and 14 μg/mL **37**: 3.7, 6.9, 9.4 and 12 μg/mL). A structure-activity relationship analysis revealed that the phenyl group in **32**–**34** may contribute to their inhibitory ability, but the isomerization at C-9 and/or C-11 of **32**–**37** only has slight influences on their activities. K2Cr2O7 was used as positive control with IC_50_ values of 0.46, 0.98, 0.89 and 1.9 μg/mL, respectively [21].

The biological effects of compound **38** were evaluated on the seedling growth of *Arabidopsis thaliana*, and **38** displayed an effect on the root growth but no remarkable inhibition of leaf growth in *Arabidopsis thaliana* [22].

The antioxidant activity was estimated by using adapted 2, 2′-diphenyl-b-picrylhydrazyl (DPPH) method. Ascorbic acid (IC_50_ = 2.0 μM) and methanol were used as positive and negative controls, respectively. **49** and **413** showed remarkable antioxidant activity with IC_50_ values of 2.50 and 5.75 μM respectively [24].

The biological activity properties of compounds **63**–**65** were evaluated for inhibitory activity against pancreatic lipase. Compounds **63**–**65** displayed potent inhibition in the assay with IC_50_ values of 2.83 ± 0.52, 5.45 ± 0.69, and 6.63 ± 0.89 μM, respectively, compared to the standard kaempferol (1.50 ± 0.21 μM) [29].

Nuclear transcription factor (PXR) can regulate a suite of genes involved in the metabolism, transport, and elimination of their substances, such as CYP3A4 and MRP, therefore, it is regarded as an important target to treat cholestatic liver disorders. So compound **76** was assayed for agonistic effects on PXR. Compound **76** displayed the significant agonistic effect on PXR with EC_50_ value of 134.91 ± 2.01 nM [33].

Brine shrimp inhibiting assay was assayed. Compound **80** displayed brine shrimp inhibiting activities with IC_50_ value of 10.1 μmol/mL. The SDS (sodium dodecyl sulfate) was employed as positive control and its inhibiting ratio was 95% for brine shrimp and LC_50_ 0.6 μmol/mL [36].

Monitoring the NO level in LPS-activated cells has become a common approach for evaluating the potential anti-inflammatory activities of compounds. Isolates **82**–**92** were evaluated for their inhibitory activity against NO production in LPS-activated RAW 264.7 marcrophages, while indomethacin was used as a positive control. Compounds **89**–**91** exhibited inhibitory effects with IC_50_ values of 21, 24 and 16 μM, respectively, which are lower than that of the positive control indomethacin (IC_50_ = 38 ± 1 μM), while compound **85** exhibited moderate inhibition with an IC_50_ value of 42 μM. Preliminary structure–activity relationships revealed that the analogues with the S absolute configureuration at C-18 (e.g., **89**–**91**) significantly enhanced the activity, as exemplified by compound **89** showing inhibition against NO production in RAW 264.7 marcrophage cells with an IC_50_ value of 21 μM, whereas compound **87** exerted less than 40% inhibition at 50 μM. In addition, all isolated compounds (**82**–**92**) were tested for their inhibitory activity of Mycobacterium tuberculosis protein tyrosine phosphatase B (MptpB). Compound **89** displayed inhibition with an IC_50_ value of 19 μM, comparable to the positive control (oleanolic acid, IC_50_ = 22 ± 1 μM). Compounds **83**, **85**, **86** and **90** showed moderate inhibitory activity of MptpB with IC_50_ values of 39 ± 2 μM, 42 ± 3 μM, 28 ± 1 μM and 35 ± 1 μM, respectively [38].

Compounds **135**–**146** were evaluated for their inhibitory effects on the NO production in LPS-stimulated RAW264.7 microglial cells using Griess assay. Meanwhile, the effects of compounds **135**–**146** on cell proliferation/viability were measured using the MTT method. As a result, compounds **138**, **139**, **142**, **143**, **145** and **146** exhibited inhibitory activity against NO production with IC_50_ values in the range of 56.3–98.4 μM (IC_50_ values of compounds on LPS-stimulated NO production in RAW264.7 macrophage cells **138**: 85.2 ± 4.3 µM, **139**: 98.4 ± 5.6 µM, **142**: 95.9 ± 3.4 µM, **143**: 64.8 ± 1.3 µM, **145**: 60.0 ± 3.1 µM, **146**: 56.3 ± 1.1 µM). Indomethacin was used as a positive control (IC_50_ = 33.6 ± 1.4 μM) [45].

Measurement of ATP release of thrombin-activated platelets of the isolated compound **168** was investigated by applying D. S. Kim’s method. Compound **168** exhibited inhibitory activities on ATP release of thrombin-activated platelets with IC_50_ value of 57.6 ± 3.2 μM. Staurosporine served as the positive control with IC_50_ value of 3.2 ± 0.6 μM [51].

The inhibition of biofilm formation against *Staphylococcus aureus* DSM 1104 was tested in 96-well tissue microtiter plates. The compounds were tested in concentrations of up to 256 µg/mL. MeOH and cytochalasin B were used as negative and positive control, respectively. Minimum Inhibitory Concentration (MIC) value of 256 µg/mL was observed for metabolite **169** and it showed a weak inhibition of biofilm formation of 20.78% at 256 µg/mL [52].

A colorimetric α-glucosidase (Sigma-Aldrich Co. CAS number: 9001-42-7, E.C 3.2.1.20) assay of compounds **176**–**180** was performed. 1-deoxynojirimycin (St. Louis, MO, USA) was used as a positive control. In addition, The DPPH radical scavenging assay of these compounds was also conducted with 96-well plates using a revised method. The positive control was used by Vitamin C. Compounds **176**–**178** showed significant α-glucosidase inhibitory activity with IC_50_ values of 35.8 μM, 53.3 μM and 60.2 μM, respectively, compared to 62.8 μM for the positive control (1-deoxynojirimycin). Moreover, compound **179** exhibited radical scavenging activity against DPPH with EC_50_ value of 68.1 μM, the EC_50_ value of positive control ascorbic acid was 22.3 μM [54].

The tested compounds **200**, **276**, **344**–**346** were investigated for their capacity to inhibit biofilm formation in the reference strains of *S. aureus*, *E. faecalis* and *E. coli*. Aacetylquestinol **276**, **345** and **200** were found to cause a significant reduction inbiofilm production by *E. coli* ATCC 25922 with the percentage of biofilm formation: 50.6 ± 17.6%, 23.7 ± 24.8% and 57.6 ± 8.1%, respectively. On the other hand, emodin **344** and **345** showed inhibition of biofilm production in *S. aureus* ATCC 25923 (21.1 ± 11.5% and 21.8 ± 18.9%). Interestingly, **345**, which is the most effective in inhibiting biofilm formation in *E. coli* ATCC 25922, also caused nearly 80% reduction of the biofilm production in *S. aureus* ATCC 25923 [62].

Compound **207** was evaluated for its acetylcholinesterase (AChE) inhibitory activity using the Ellman colorimetric method, it showed weak AChE inhibitory activity with the inhibition ratio of 11.9% at the concentration of 50 μmol/mL [65].

The anti-inflammatory activities of the isolated compounds **210**–**211** were evaluated by measuring the inhibitory activity of nitric oxide (NO) production levels in the lipopolysaccharide (LPS)-induced RAW264.7 macrophage cells. **210**–**211** exhibited moderate inhibitory activities on NO production in LPS-stimulated RAW264.7 cells without cell cytotoxicities [67].

The transformed products **224**–**225** and the parent compound **L** (Figure 13) were evaluated for the neuroprotective activity using the LPS-induced neuro-inflammation injury assay. **224**–**225** exhibited moderate neuroprotective activity by increasing the viability of U251 cell lines with EC_50_ values of 35.3 ± 0.9 nM and 32.1 ± 0.9 nM, respectively, while **L** (EC_50_ = 8.3 ± 0.4 nM) exhibited comparable activity with the positive control ibuprofen (EC_50_ = 19.4 ± 0.7 nM). The transformed products **224**–**225** and **L** all exhibited considerable neuroprotective activity in the invitro LPS-induced neuro-inflammation injury assay, suggesting that the hupA moiety shared by these compounds may be used as a lead structure for the development of neuroprotective drugs [73].

The artificial insect mixed drug method was used to determine the insecticidal activities of compound **228**. Compound **228** displayed remarkable insecticidal activities against first instar larvae of the cotton bollworm Helicoverpa armigera with mortality rates of 70.2%. Commercially-available matrine was used as positive control, causing 87.4% mortality rate under the same conditions. Acute cytotoxicity towards hatching rate, malformation and mortality of zebrafish embryos or larvae were also performed. Compounds **227** and **228** significantly decreased the hatching rate of zebrafish embryos, compound **228**, used at concentrations of 5–100 μg/L, decreased the hatching rate of zebrafish embryos to below 20% [74].

The potential phytotoxicity of **246** against lettuce seedlings (*Lactuca sativa* L.) was studied. Aqueous solutions of **246** ranging between 25 and 200 μg mL^−1^, were assayed for its effects on seed germination, root length, and shoot length of the lettuce. Compound **246** showed the most robust inhibitory effect on root growth. Compound **246** inhibited root growth by 50% at a concentration of 25 µg/mL. In addition, the highest concentration of **246** (200 µg/mL) strongly exerted an inhibitory effect on seed germination (90% inhibition) [81].

Compounds **256**–**257** were investigated for their inhibitory activities against the LPS-activated production of NO in RAW264.7 cells using the Griess assay with indomethacin as a positive control (IC_50_ = 37.5 ± 1.6 μM). The effects of compounds on cell proliferation/viability were determined using MTT method, and none of the test compounds exhibited cytotoxicity at their effective concentrations. Compounds **256** and **257** showed strong inhibitory effects on the production of NO, with IC_50_ values of 0.78 ± 0.06 and 1.26 ± 0.11 μM, respectively [84].

In vitro anti-inflammatory effects of compounds **258**–**261** were evaluated in lipopolysaccharide (LPS)-stimulated RAW264.7 macrophages. **258**–**261** exhibited excellent inhibitory effects on the production of interleukin-1β (IL-1β), tumor necrosis factor-α (TNF-α), and nitric oxide (NO) in LPS-induced macrophages with the IC_50_ values ranging from 16.21 ± 1.62 μM to 35.23 ± 3.32 μM, from 19.83 ± 1.82 μM to 42.57 ± 4.56 μM, from 16.78 ± 1.65 μM to 38.15 ± 3.67 μM, respectively, similar with the positive control indomethacin. Those results indicated that, terrusnolides A–D (**258**–**261**) might play a significant role as a lead compound in the study of anti-inflammatory agents. In addition, compounds **258**–**261** were also investigated for the inhibitory activities against BACE1 by M-2420 method and acetylcholin esterase (AchE) using Ellman’s method. Compound **260** exhibited weak AchE inhibitory activity with IC_50_ value of 32.56 ± 3.16 µM, compound **261** exhibited weak BACE1 inhibitory activity with IC_50_ value of 37.45 ± 4.56 µM. LY2811376 and Donepezil were used as the positive control in BACE1 and AchE inhibitory assay with IC_50_ values of 0.25 ± 0.04 µM and 0.05 ± 0.01 µM, respectively [85].

The Indoleamine 2,3-dioxygenase (IDO) inhibitory activity assay of compounds **284**–**286** were carried out. The results showed that compound **285** possessed significant inhibitory activity against IDO with IC_50_ value of 0.11 μM. Epacadostat, as the positive control, was one of the most potent IDO inhibitors with IC_50_ value of 0.05 μM. For compounds **284** and **286**, they showed relatively strong inhibitory activity with IC_50_ values of 1.47 μM and 6.36 μM, respectively [92].

NF-κB has been considered as an attractive therapeutic target for the cancer research. Compound **288** was investigated for its effects on NF-κB pathway by reporter gene assay. The results showed that it could activate the NF-κB pathway with increments in the relative luciferase activity at a concentration of 50 μM [93].

The phytotoxic activities of **295** and **296** were investigated by seed germination test on lettuce (*Lactuca sativa* L.) with 2,4-dichlorophenoxyacetic acid (0.3 µg/mL) as the positive control. Compounds **295** and **296** each inhibited the growth of both roots and hypocotyls at 30 µg/mL. Furthermore, **295** suppressed seed germination at 100 µg/mL [97].

Acetylcholinesterase (AChE) inhibitory activities of the compound **302** were assayed by the spectrophotometric method developed by Ellman with modification. **302** showed weak AChE inhibitory activity (The percentage inhibition was at 20%~60% in 50 μM) [99].

The 5-lipoxygenase (5-LOX) inhibitory potential of **306**–**308** from *Fusarium* sp. was assessed in an attempt to explore their activity against 5-LOX. It is noteworthy that **306** displayed prominent 5-LOX inhibitory activity with IC_50_ value of 3.61 μM, compared to that of indomethacin (IC_50_ = 1.17 μM), while **307** and **308** had moderate activity with IC_50_ values of 7.01 μM and 4.79 μM, respectively [101].

α-Glucosidase inhibitory activity was performed in the 96-well plates and acarbose was used as the positive compound. In the inhibitory assay against α-glucosidase, compound **313** displayed moderate activities [104].

The anti-inflammatory activities of selected isolated 4 compounds **314**–**317** were evaluated as inhibitory activities against lipopolysaccharide (LPS) induced nitric oxide (NO) production in RAW264.7 cell lines. Compound **317** showed the most NO inhibitory effects, with the inhibition of 17.4% NO production in LPS stimulated RAW264.7 cells at 10 μM. At the same concentration, compound **315** significantly inhibited the NO production, with 11.2% inhibitory rate. Compound **314** showed weak NO inhibitory effects at 10 μM, with inhibitory rates of 6.5%. At the same concentration, quercetin, the positive control, inhibited NO production to 12.9% [105].

The Superoxide anion radical scavenging activity of compound **331** was investigated. It displayed strong antioxidant activity with EC_50_ value of 1.08 mg/mL on superoxide anion racdicals. Ascorbic acid (Vc) was used as positive control with EC_50_ value of 0.33 mg/mL [111].

Compounds **333** and **334** were subjected to motility inhibitory and zoosporicidal activity tests against *P. capsici* (*Phomopsis capsici*). Compounds **333** and **334** showed more than 50% motility inhibitory activity (IC_50_) at a concentration of 50−100 μg/mL [112].

Human carboxylesterases (hCE 1 and hCE 2) are the important enzymes that hydrolyze chemicals with functional groups, such as a carboxylic acid ester and amide, and they are known to play vital roles in drug metabolism and insecticide detoxication. The isolated compounds **379**–**385** were assayed for their inhibitory activities against hCE 2. Loperamide was used as a positive control with IC_50_ value of 1.31 ± 0.09 µM. Compounds **379**, and **383**–**385** displayed significant inhibitory activities against hCE 2 with IC_50_ values of 10.43 ± 0.51, 6.69 ± 0.85, 12.36 ± 1.27, 18.25 ± 1.78 µM, respectively [94].

The inhibitory effects on human carboxylesterases (hCE1, hCE2) of compound **386** were evaluated. The results demonstrated that bysspectin A **386** was a novel and highly selective inhibitor against hCE2 with the IC_50_ value of 2.01 µM. Docking simulation also demonstrated that active compound **386** created interaction with the Ser-288 (the catalytic amino-acid in the catalytic cavity) of hCE2 via hydrogen bonding, revealing its highly selective inhibition toward hCE2 [124].

Compounds **392**–**393** were also evaluated for growth inhibition activity against newly hatched larvae of *H. armigera Hubner*. Compounds **392** and **393** showed growth inhibition activities against newly hatched larvae of *H. armigera Hubner* with the IC_50_ values of 150 and 100 µg/mL, respectively. Azadirachtin was used as positive control with the IC_50_ value of 25 µg/mL [128].

Antioxidant activity of the compound **403** was determined by DPPH assay and compared with the positive control BHT. Compound **403** showed moderate antioxidant activities with IC_50_ value of 120.1 ± 11.7 μg/mL [131].

The new compounds **406**–**407** were subjected for determination of the xanthine oxidase (XO) inhibitory activity using microtiter plate based NBT assay. Allopurinol was used as a positive control with IC_50_ value of 0.18 ± 0.02 µg/mL. **406** and **407** showed XO inhibitory activity with IC_50_ values of 2.81 ± 0.71 and 0.41 ± 0.1 µg/mL, respectively. The oxidized form of **406** also showed high XO inhibition with IC_50_ value of 0.35 ± 0.13 µg/mL [133].

Compound **421** was tested for osteoclastic differentiation activity using murine macrophage derived RAW264.7 cells. **421** significantly increased the number of mature osteoclasts at the comparable levels to the positive control of kenpaullone, compared to the negative control (DMSO), suggesting that **421** activated a signaling pathway in osteoclastic differentiation [139].

Phtotoxicity assay against lettuce seedlings of compound **432** was carried out using a published protocol. The new compound (−)-dihydrovertinolide **432** exhibited phytotoxicity against lettuce seedlings at a concentration of 50 mg/L [140].

All new compounds were tested for in vitro anti-inflammatory activities against nitric oxide production in liposaccharide (LPS)-induced RAW264.7 cells, and dexamethasone was used as the positive control. Compound **436** showed significant inhibitory activity against NO production in LPS-induced RAW264.7 cells with an IC_50_ value of 1.9 μM. They were also evaluated for in vitro antidiabetic activities based on the inhibition of alpha-glucosidase, PTP1b, and XOD. Compounds **437** and **441** showed moderate inhibitory activities toward XOD and PTP1b, respectively, at 10 μM with inhibition rates of 67% and 76% [87].

New compound **447** was tested for acetylcholinesterase (AChE) inhibitory activities using the Ellman method with tacrine as the positive control. The results revealed that compound **447** showed weak AChE inhibitory activity wth IC_50_ value of 23.85 ± 0.20 μM. Tacrine are the positive control used to estimate AChE inhibitory activity with IC_50_ value of 0.26 ± 0.02 μM [27].

All information about the new compounds are briefly summarized in the Table 1 below.

## 4. Conclusions

From 2017–2019, a total of 449 new secondary metabolites isolated from plant endophytic fungi using different culture method like common culture, co-culture with bacteria, addition of metal ions and so on, were summarized in this review. These compounds have a variety of unique structures, the difference in structure leads to various biological activities of these compounds. Some of these metabolites display significant antimicrobial effects, cytotoxic activities, antioxidant activities and other biological activities, which indicate that they have potential to be agents to treat some diseases. In this review, structure-activity relationships of some compounds were also reviewed.

According to genome sequencing, a lot of microorganisms have the potential to produce secondary metabolites with novel structures. However, many fungal gene clusters may be silent under standard laboratory growth conditions. As a result, some pathways to yield secondary metabolites cannot be expressed. Therefore, activating these pathways means that we can get more novel compounds. The approach of microorganism co-culture, involving the cultivation of two or more microorganisms in the same lab environment can do a favour for us. Interestingly, 29 new compounds summarized above were obtained through co-culture of bacteria and fungi or two fungi. Besides, by adding CuCl_2_ into fermentation medium of an endophytic fungus *P. citrinum* 46, two compounds were isolated. The results showed that adding Cu^2+^ into medium to activate silent fungal metabolic pathways can increase the discovery of new compounds.

Because the compounds mentioned above were isolated from endophytic fungi in different parts of different plants in different regions, they have a variety of structures and biological activities. In addition to anti-tumor and anti-microbial activities, some compounds also exhibit unique biological activities. Among them, 7 compounds showed weak to moderate AChE inhibitory activity. Some compounds exhibited moderate to potent α-glucosidase inhibitory activity compared with those of positive control. By using adapted 2,2′-diphenyl-b-picrylhydrazyl (DPPH) method, a few of compounds were found to show moderate to remarkable antioxidant activity. Some of them also showed weak to significant inhibitory activity against NO production in LPS-induced RAW264.7 cells. The biological activity properties of 18 compounds were evaluated for inhibitory activity against some enzymes like pancreatic lipase, the 5-lipoxygenase (5-LOX), the Indoleamine 2,3-dioxygenase (IDO), Mycobacterium tuberculosis protein tyrosine phosphatase B (MptpB), the xanthine oxidase (XO) and so on, they showed weak to high inhibition.

Endophytic fungi isolated from different parts of plants are a huge treasure house on account of the discovery of novel secondary metabolites with biological activities and unique structures. Since the endophyte resources were discovered, more and more researches have been conducted on them. Just from my review article, the new secondary metabolites isolated from plant endophytes during the three years from 2017 to 2019 were counted. Among them, 38 articles were published in 2017, 136 new compounds were obtained; 39 articles were published in 2018, 117 new compounds were obtained; 57 articles were published in 2019, and 196 new compounds were obtained. It can be discovered that in the past three years, the research trend of plant endophytes and their metabolites have increased year by year. The more new compounds obtained, the greater the possibility of screening compounds with excellent biological activity. This is also an important significance for researchers to study plant endophytes. Through this review, i hope to arouse more people’s interest and attention in this field and screen out compounds with good biological activities to create a better life for mankind by utilizing endophytes resources.

## Figures and Tables

**Figure 1 ijms-22-00959-f001:**
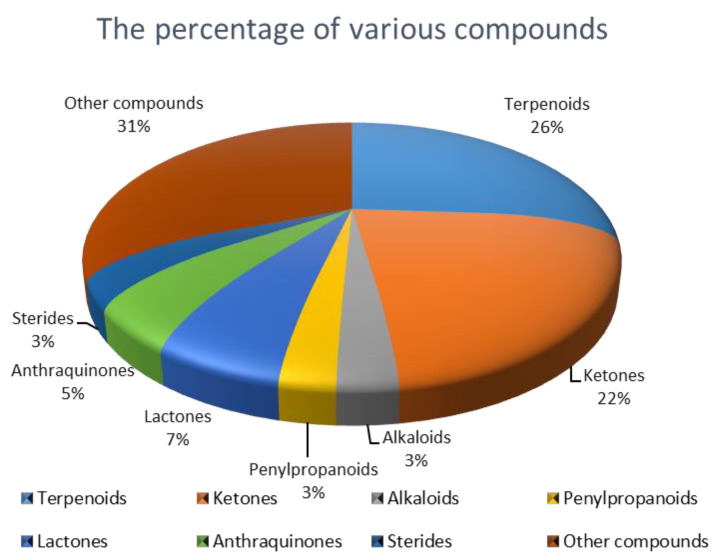
Percentage of metabolites synthesized by endophytes.

**Figure 2 ijms-22-00959-f002:**
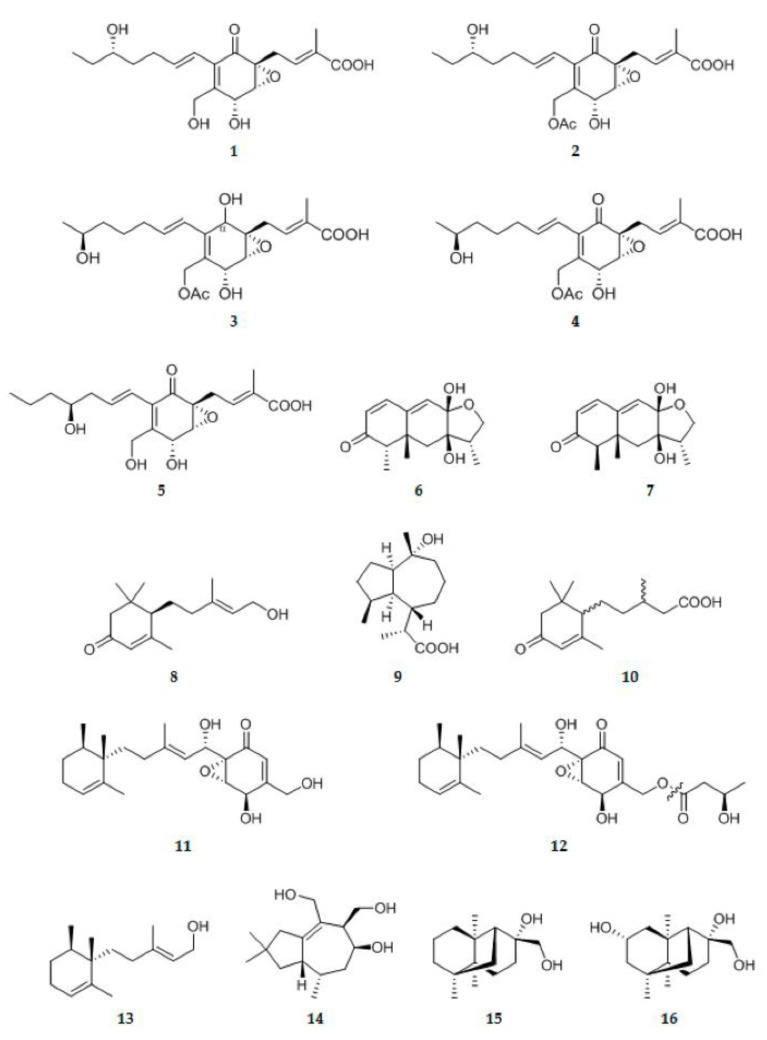
Chemical structures of sesquiterpenoids and derivatives.

**Figure 3 ijms-22-00959-f003:**
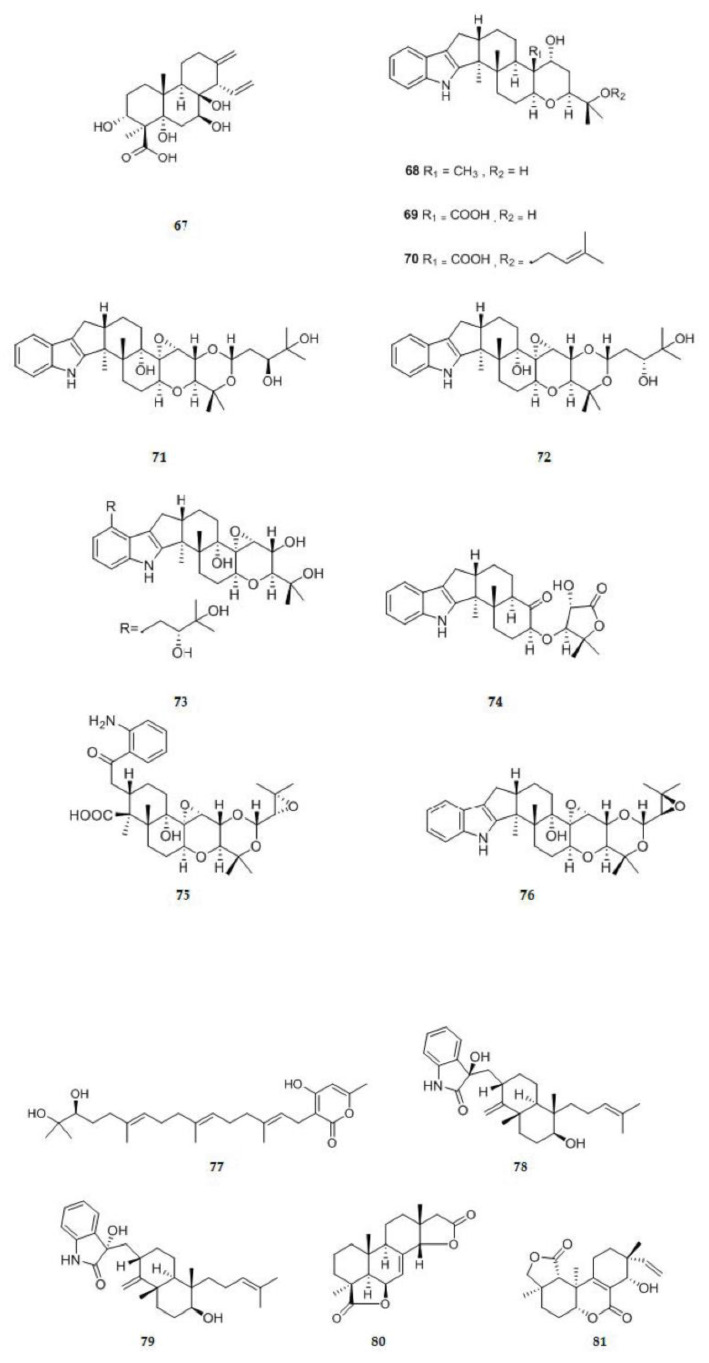
Chemical structures of diterpenoids and derivatives.

**Figure 4 ijms-22-00959-f004:**
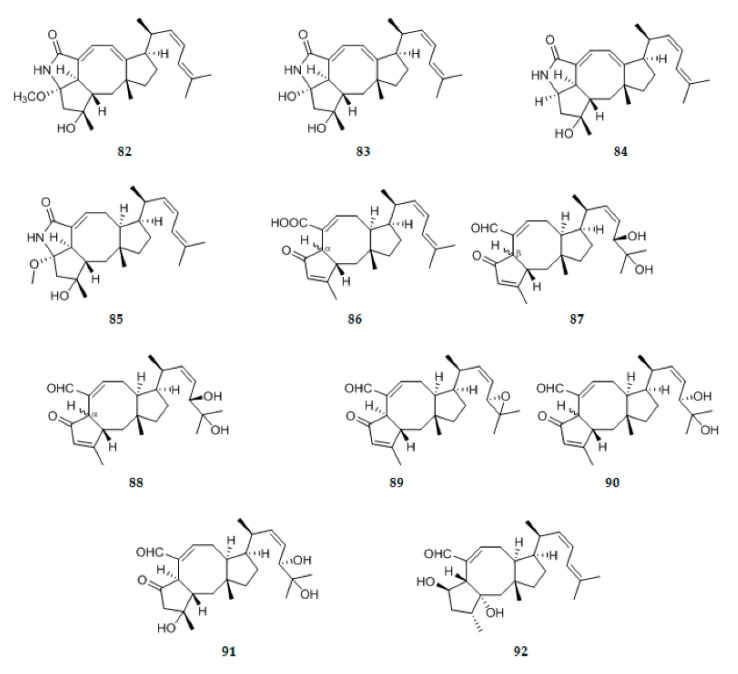
Chemical structures of other terpenoids and derivatives.

**Figure 5 ijms-22-00959-f005:**
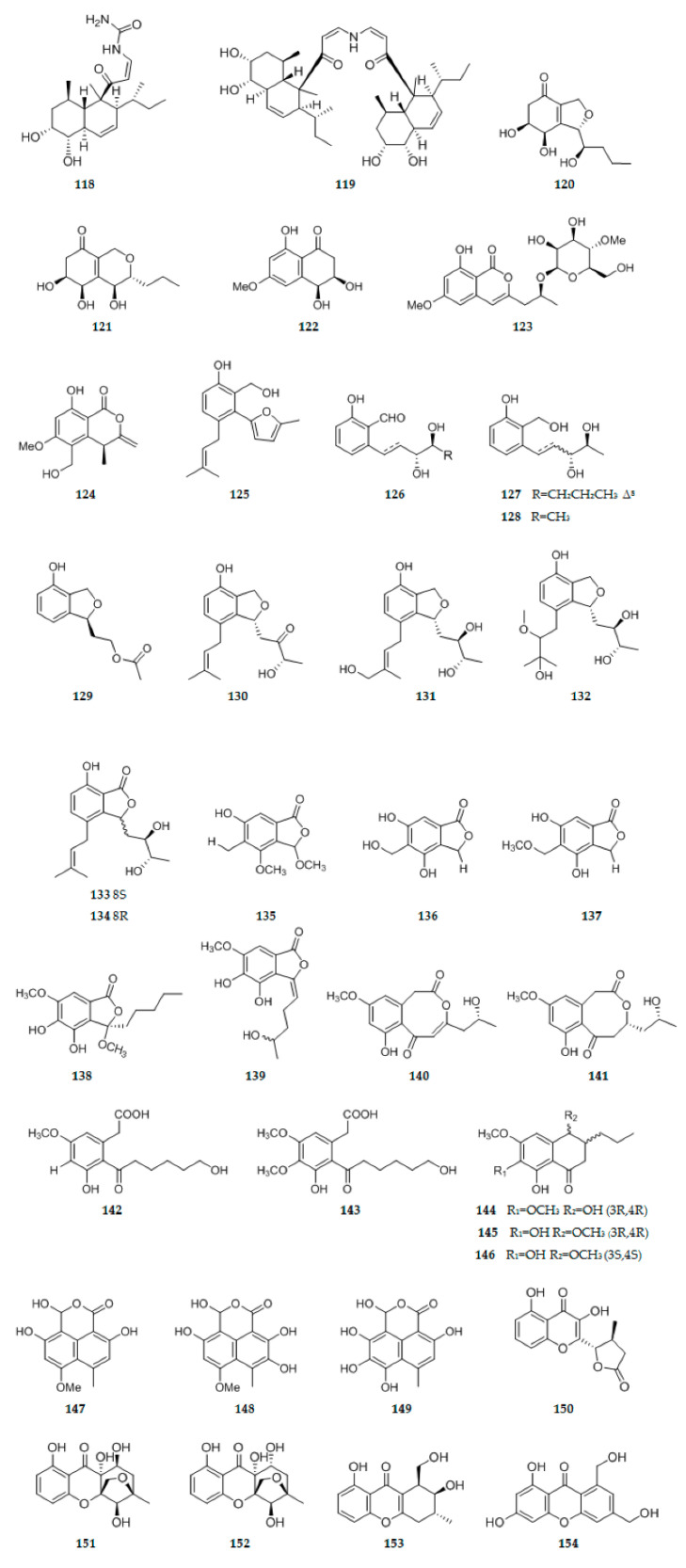
Chemical structures of polyketides.

**Figure 6 ijms-22-00959-f006:**
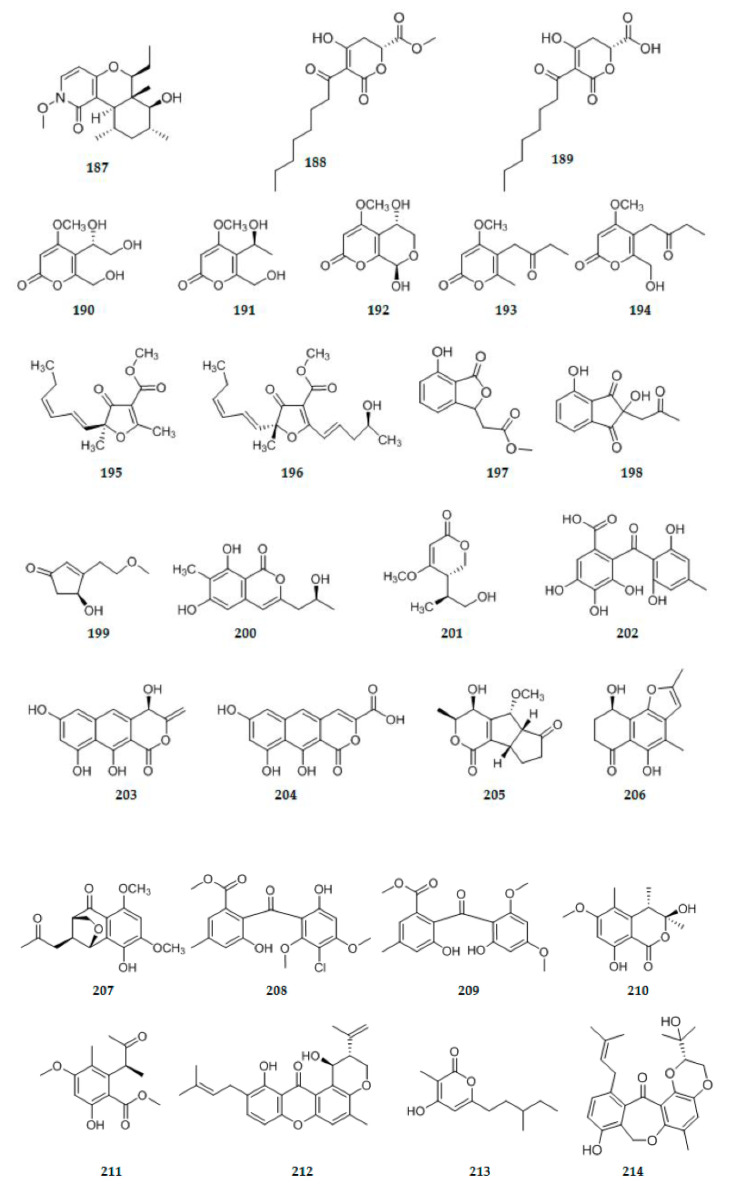
Chemical structures of other ketones.

**Figure 7 ijms-22-00959-f007:**
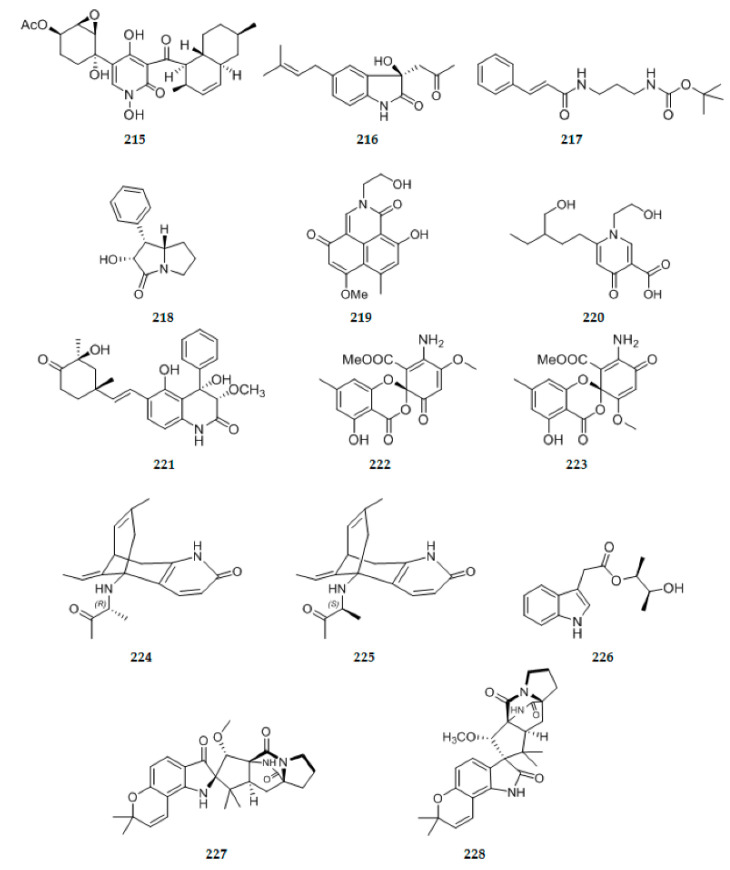
Chemical structures of alkaloids and their derivatives.

**Figure 8 ijms-22-00959-f008:**
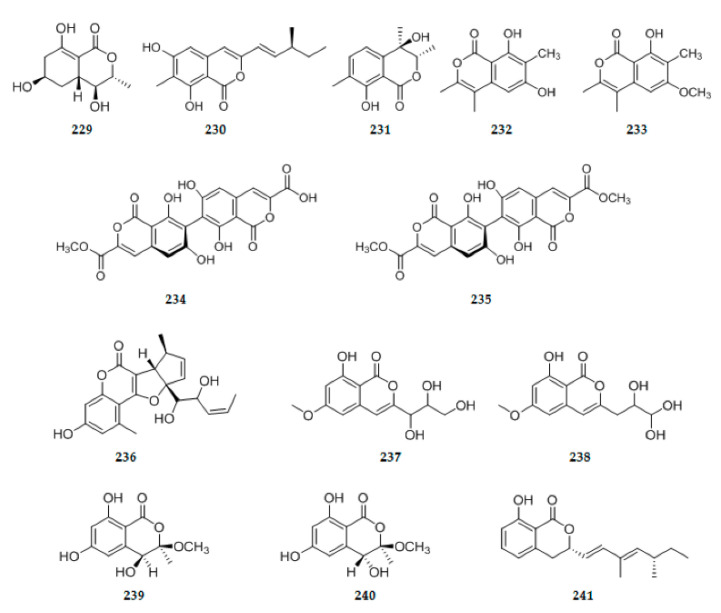
Chemical structures of penylpropanoids and their derivatives.

**Figure 9 ijms-22-00959-f009:**
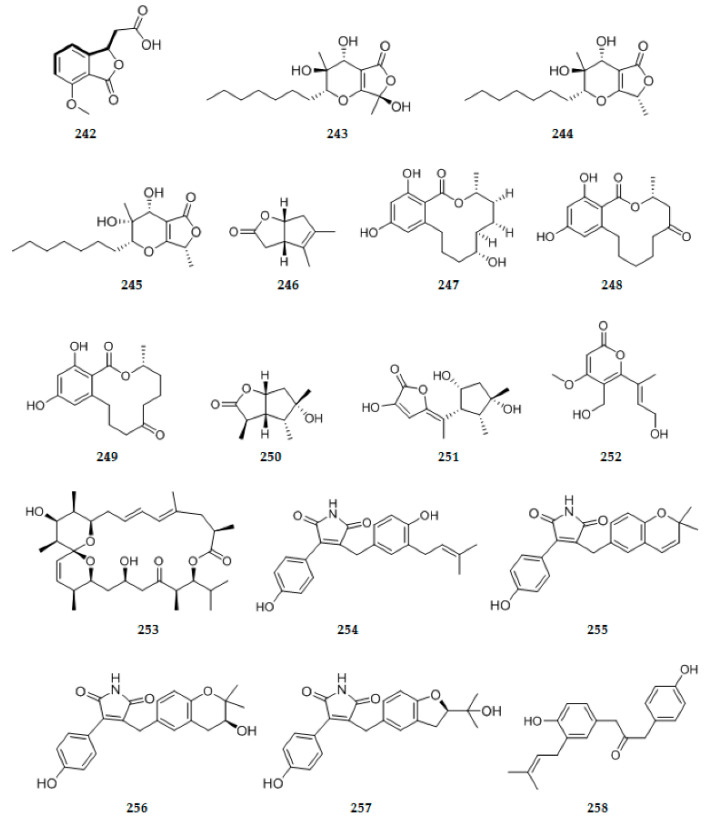
Chemical structures of lactones.

**Figure 10 ijms-22-00959-f010:**
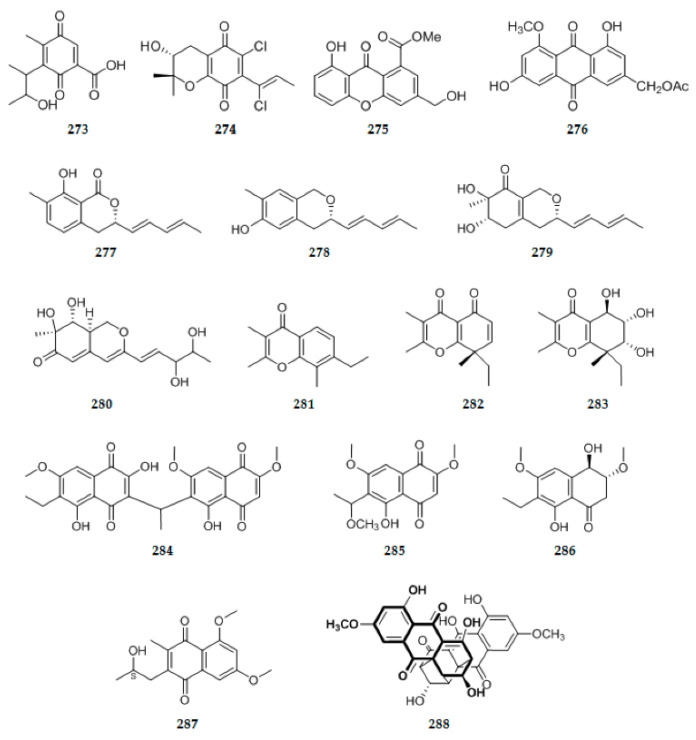
Chemical structures of anthraquinones.

**Figure 11 ijms-22-00959-f011:**
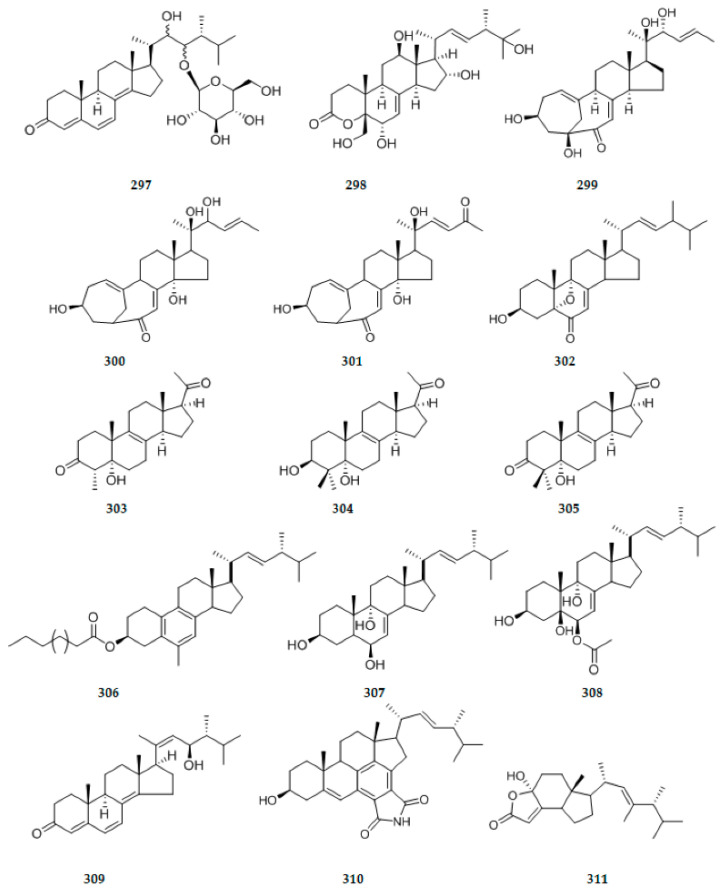
Chemical structures of sterides.

**Figure 12 ijms-22-00959-f012:**
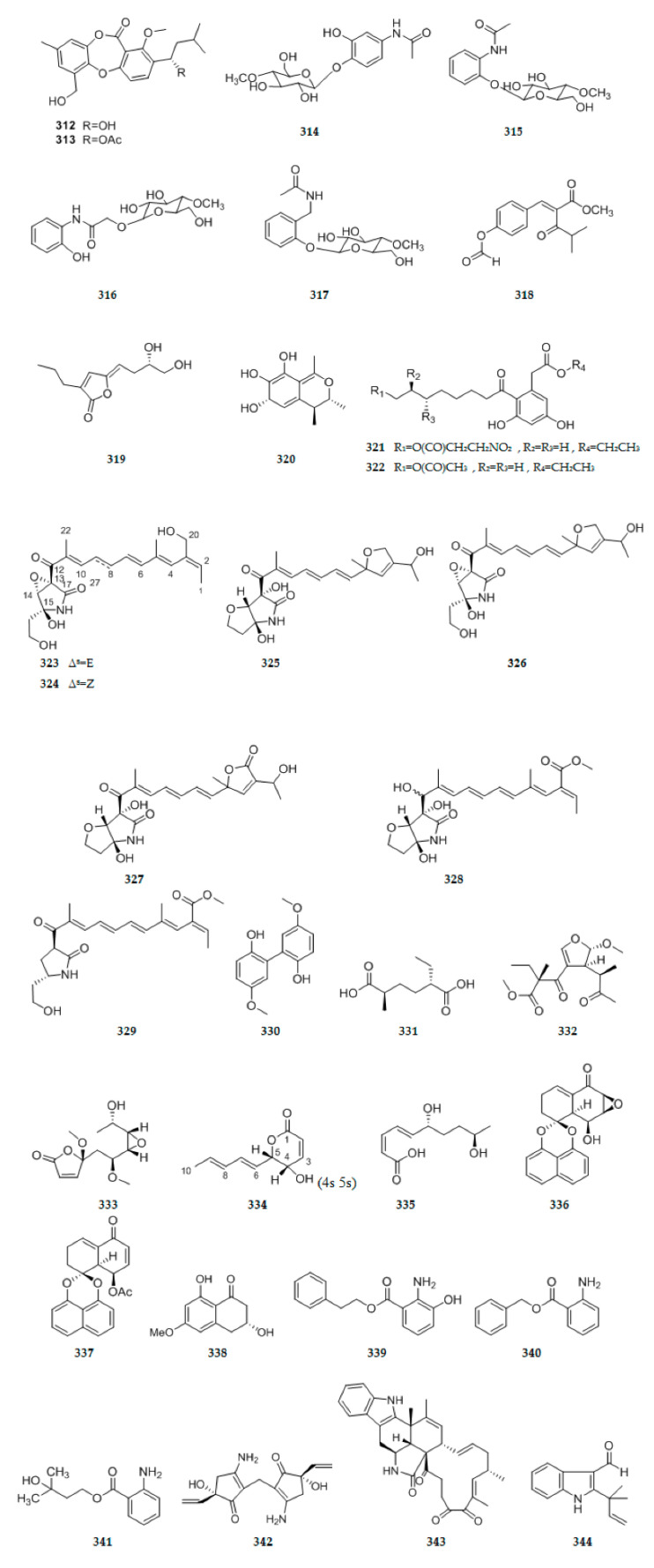
Chemical structures of other new compounds.

**Figure 13 ijms-22-00959-f013:**
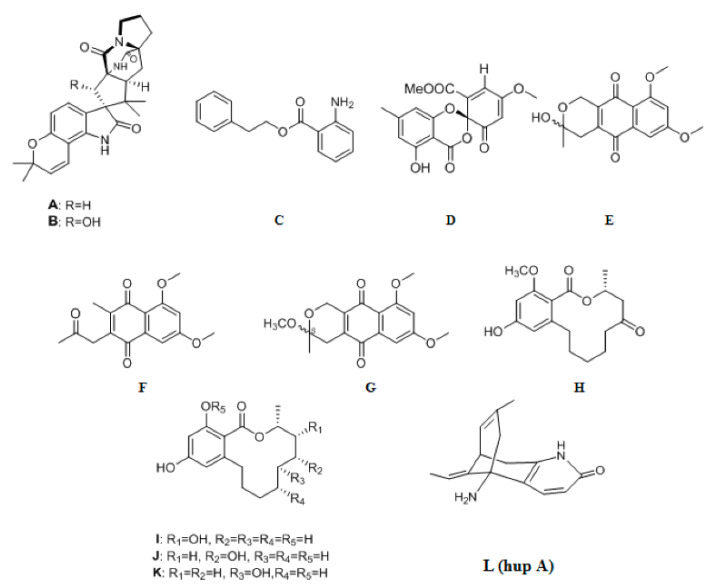
Chemical structures of known compounds.

**Table 1 ijms-22-00959-t001:** Brief summary of new compounds.

Compound	Molecular Formula	Degree of Unsaturation	Color and Morphology	Endophytic Fungus	Host Plant	Site and Nation	Biological Activity	Ref.
Terpenoids Sesquiterpenoids and derivatives								
**1**	C_19_H_26_O_7_	7	brown oil	*Pestalotiopsiss*p.T	lichen *Cetraria islandica* (L.) Ach.	Yunnan Province, China	Inhibit the growth of plant pathogenic fungus (1,5)	[9]
**2**	C_21_H_28_O_8_	8						
**3**	C_21_H_30_O_8_	7						
**4**	C_21_H_28_O_8_	8						
**5**	C_19_H_26_O_7_	7						
**6**	C_15_H_20_O_4_	6	white powder	Co-culture Strain 307: *Trichoderma* sp. the stem bark of *Clerodendrum inerme* Bacterium B2: *Acinetobacter johnsonii* From an aquaculture pond		Guangdong Province, China	Show moderate inhibitory activity against α-glucosidase (7)	[10]
**7**	C_15_H_20_O_4_	6						
**8**	C_15_H_24_O_2_	4	colorless gum	*Trichoderma atroviride*	bulb of *Lycoris radiata*.	Hubei Province China	Inactive	[11]
**9**	C_15_H_26_O_3_	3	white amorphous powder	Co-culture *Pestalotiopsis* sp. fruits of *Drepanocarpus lunatus* (Fabaceae) *Bacillus subtilis*			Weak antibacterial activities (9)	[12]
**10**	C_15_H_24_O_3_	4	colorless oi					
**11**	C_22_H_32_O_5_	7	colorless crystal	*Nectria pseudotrichia* 120-1NP	Inner tissue of *Gliricidia sepium* healthy stem		Cytotoxicity (11–13)	[13]
**12**	C_26_H_38_O_7_	8	yellow oil					
**13**	C_15_H_26_O	3	yellow oil					
**14**	C_15_H_26_O_3_	3		Co-culture *Nigrospora oryzae* *Irpex lacteus*	seeds of *Dendrobium officinale*	Yunnan Province, China	Anti-AChE activity	[14]
**15**	C_15_H_26_O_2_	3	white powder	*Emericella* sp. XL 029	leaves of *Panax notoginseng*	Hebei province, China	Antifungal activity Antibacterial activity (15,16)	[15]
**16**	C_15_H_26_O_3_	3	colorless oil					
**17**	C_19_H_24_O_4_	8	colorless oil	*Trichothecium crotocinigenum*			Antiphytopathogenic activity (17–20)	[16]
**18**	C_19_H_25_ClO_5_	7	colorless crystals					
**19**	C_22_H_28_O_5_	9	colorless crystals					
**20**								
**21**	C_15_H_22_O_4_	5	colorless oil	*Trichoderma atroviride* S361	Bark of *Cephalotaxus fortunei*	Zhejiang province, China	Inactive	[17]
**22**	C_15_H_20_O_4_	6	white amorphous powder	*Aspergillus* sp. xy02	leaves of mangrove *Xylocarpus moluccensis*	Trang Province, Thailand	Antibacterial activity (23–24,26,28)	[18]
**23**								
**24**								
**25**								
**26**			colorless oil					
**27**								
**28**								
**29**	C_15_H_24_O_3_	4	colorless oil	*Pestalotiopsis adusta*	stem bark of medicinal plant *Sinopodophyllum hexandrum* (Royle) Ying	Qinling Mountains China	Weak to moderate cytotoxic activity	[19]
**30**	C_15_H_26_O_2_	3	colorless oil	*F. proliferatum* AF-04	green Chinese onion	Lanzhou, China		[20]
**31**	C_14_H_24_O_3_	3	colorless crystals	*Trichoderma asperellum* A-YMD-9–2	marine Red alga *Gracilaria verrucosa*	Yangma Island, Yantai, China	Potent inhibition of several marine phytoplankton species 31–37	[21]
**32**	C_14_H_20_O_2_	5	colorless oil					
**33**								
**34**								
**35**	C_22_H_37_NO_7_	5	colorless oil					
**36**								
**37**								
**38**	C_15_H_24_O_4_	4	crystal powder	*Alternaria oxytropis*	desert plant locoweed *Oxytropis glabra*	Inner Mongolia, China	Displayed an effect on the root growth in *Arabidopsis thaliana* (38)	[22]
**39**			colourless oil					
**40**	C_15_H_22_O_4_	5	colourless oil					
**41**	C_15_H_24_O_5_	4	crystal powder					
**42**	C_15_H_22_O_3_	5	colourless oil					
**43**	C_15_H_24_O_5_	4						
**44**	C_15_H_26_O_4_	3						
**45**								
**46**	C_15_H_26_O_3_	3						
**47**								
**48**	C_15_H_22_O_3_	5	colorless crystal	*Pleosporales* sp. SK7	mangrove plant *Kandelia candel*	Guangxi Province, China		[23]
**49**	C_15_H_22_O_4_	5	yellowish needle crystals	*Irpex lacteus* DR10-1	waterlogging tolerant plant *D. chinense*	Chongqing China	Antioxidant activity Antibacterial activity	[24]
**50**	C_15_H_16_O_3_	8	colorless crystals	*Trichoderma virens* QA-8	fresh inner tissue of the medicinal plant *Artemisia argyi*	Hubei Province, China	Antibacterial (50–52,55) Antifungal activity (50–55)	[25]
**51**	C_15_H_16_O_4_	8	colorless oil					
**52**	C_15_H_22_O_2_	5	amorphous powder					
**53**	C_15_H_22_O_3_	5	amorphous powder					
**54**	C_15_H_24_O_3_	4	colorless oil					
**55**	C_14_H_16_O_4_	7	amorphous powder					
**56**	C_15_H_26_O_2_	3	colorless needle	*Alternaria alternate*	leaves of *Psidium littorale* Raddi	Fujian Province, China		[26]
**57**	C_15_H_22_O_3_	5	colorless oil	*Epicoccum* sp. YUD17002 & *Armillaria* sp.	rhizomes of the underground portion of *Gastrodia elata*	Yunnan Province, China		[27]
**58**	C_15_H_24_O_4_	4						
**59**	C_15_H_22_O_2_	5						
**60**	C_15_H_22_O_3_	5						
**61**	C_15_H_24_O_4_	4	white amorphous powder					
**62**	C_29_H_42_O_9_	9	sticky and optically active oi	*Colletotrichum gloeosporioides*	Cameroonian medicinal plant *Trichilia monadelpha* (Meliaceae)	Yaounde, Central region, Cameroon		[28]
**63**	C_17_H_22_O_7_	7	white powder	*Penicillium purpurogenum* IMM003	leaf tissue of the medicinal plant *Edgeworthia chrysantha*.	China	Show significant inhibitory activity against pancreatic lipase	[29]
**64**	C_17_H_20_O_7_	8	colorless crystals					
**65**	C_16_H_20_O_6_	7						
**66**	C_16_H_24_O_3_	4	yellow oil	*Fusarium oxysporum* ZZP-R1	coastal plant *Rumex madaio Makino*	Putuo Island (Zhoushan, China)	Moderate antibacterial effect	[30]
Terpenoids Diterpenoids								
**67**	C_20_H_30_O_6_	6	colorless oil	*Nectria pseudotrichia* 120-1NP	healthy stem of *Gliricidia sepium*	Yogyakarta, Indonesia		[31]
**68**	C_28_H_39_NO_3_	10	amorphous powder	*Drechmeria* sp.	root of *Panax notoginseng*	Yunnan, China	Display inhibitory effect (69) Weak antimicrobial effects. (68,70,74)	[32]
**69**	C_28_H_37_NO_5_	11						
**70**	C_33_H_45_NO_5_	12						
**71**	C_32_H_43_NO_7_	12						
**72**	C_32_H_43_NO_7_	12						
**73**	C_33_H_45_NO_7_	12						
**74**	C_27_H_33_NO_5_	12						
**75**	C_32_H_33_NO_9_	17	amorphous powder	*Drechmeria* sp.	root of *Panax notoginseng*	Yunnan province, China	Display the significant agonistic effect on pregnane X receptor (PXR) (76)	[33]
**76**	C_32_H_41_NO_6_	13						
**77**	C_26_H_40_O_5_	7	colorless oil	*Neosartorya fifischeri* JS0553	Plant *G. littoralis*	Suncheon, Korea		[34]
**78**	C_28_H_39_NO_3_	10	Pale yellow oil	*Aspergillus versicolor*	fruits of the mangrove *Avicennia marina*	Red Sea, Egypt	Weak cytotoxic activity (79)	[35]
**79**								
**80**	C_20_H_26_O_4_	8	colorless crystals	*Xylaralyce* sp.	healthy leaves of *Distylium chinense*	China	Display brine shrimp inhibiting activity	[36]
**81**	C_20_H_26_O_5_	8	colorless crystals	*Apiospora montagnei*	lichen *Cladonia* sp.			[37]
Terpenoids Other terpenoids								
**82**	C_26_H_37_NO_3_	9	colorless oil	*Aspergillus* sp. ZJ-68	fresh leaves of the mangrove plant *Kandelia candel*	Guangdong Province, China.	Exhibit inhibitory effects on lipopolysaccharide-induced nitric oxide production in RAW 264.7 macrophage cells (89–91) Show comparable inhibition of Mycobacterium tuberculosis protein tyrosine phosphatase B (89)	[38]
**83**	C_25_H_35_NO_3_	9						
**84**	C_25_H_35_NO_2_	9						
**85**	C_26_H_39_NO_3_	8						
**86**	C_25_H_34_O_3_	9						
**87**	C_25_H_36_O_4_	8						
**88**	C_25_H_36_O_4_	8						
**89**	C_25_H_36_O_4_	8						
**90**	C_25_H_34_O_3_	9						
**91**	C_25_H_38_O_5_	7						
**92**	C_25_H_38_O_3_	7						
**93**	C_30_H_40_O_6_	11	yellowish needle crystals	*Kadsura angustifolia & Penicillium* sp. SWUKD4.1850	fresh healthy branches of *K. angustifolia*	China	Moderate cytotoxic activity (93–100)	[39]
**94**	C_30_H_40_O_6_	11	white needle crystals					
**95**	C_30_H_40_O_6_	11	white amorphous solid					
**96**	C_30_H_40_O_6_	11						
**97**	C_32_H_44_O_7_	11	white amorphous powder					
**98**	C_30_H_42_O_6_	10	white powder					
**99**	C_34_H_46_O_8_	12	yellow amorphous solid					
**100**	C_31_H_44_O_6_	10	yellow amorphous solid					
**101**	C_30_H_46_O_6_	8	white amorphous powder					
**102**	C_17_H_26_O_5_	5	colorless oil	P*hyllosticta capitalensis*	leaves of *Cephalotaxus fortunei* Hook	Shanxi Province, China		[40]
**103**	C_17_H_24_O_5_	6						
**104**	C_17_H_22_O_5_	7						
**105**	C_22_H_32_O_6_	7						
**106**	C_17_H_26_O_5_	5						
**107**	C_15_H_20_O_5_	6		*Aspergillus versicolor*			Show weak cytotoxic activities against Hela cells. (113–114)	[35]
**108**	C_15_H_20_O_5_	6						
**109**	C_15_H_20_O_5_	6	rose-colored oil					
**110**	C_17_H_22_O_6_	7						
**111**	C_15_H_20_O_5_	6						
**112**	C_15_H_20_O_4_	6	colorless oil					
**113**	C_17_H_22_O_5_	7						
**114**	C_17_H_22_O_5_	7						
**115**	C_16_H_24_O_5_	5						
**116**	C_21_H_32_O_3_	6	yellow oil	*Fusarium oxysporum* ZZP-R1	coastal plant *Rumex madaio Makino*	Putuo Island (Zhoushan, China)	Antimicrobial activity	[30]
**117**	C_12_H_20_O_4_	3	yellow oil	*Diaporthe lithocarpus* A740	from the twigs of medicinal plant *Morinda officinalis*	Guangdong province China		[41]
Ketones								
**118**	C_20_H_32_O_4_N_2_	6	white powder	*Eupenicillium* sp. LG41	Chinese medicinal plant *Xanthium sibiricum*	China	Cytotoxic activity Antimicrobial activity (Antibacterial)	[42]
**119**	C_38_H_59_O_6_N	10						
**120**	C_12_H_18_O_5_	4	colorless crystals	*Phomopsis* sp. sh917	fresh stems of *I. eriocalyx* var*. laxiflflora*	Kunming, China		[43]
**121**	C_12_H_18_O_5_	4	colorless powders					
**122**	C_11_H_12_O_5_	6	Brown needles					
**123**	C_20_H_26_O_10_	8	colorless needles					
**124**	C_13_H_14_O_5_	7	brown solids					
**125**	C_17_H_20_O_3_	8	white amorphous powder	P*estalotiopsis vaccinii* (cgmcc3.9199)	branch of mangrove plant *Kandelia candel* (L.) Druce (Rhizophoraceae)	coastal and estuarine areas of southern China	Anti-enterovirus 7l (EV71)	[44]
**126**	C_14_H_18_O_4_	6	colorless oil					
**127**	C_12_H_16_O_4_	5						
**128**	C_12_H_18_O_4_	4						
**129**	C_12_H_14_O_4_	6						
**130**	C_17_H_22_O_4_	7						
**131**	C_17_H_24_O_5_	6						
**132**	C_18_H_28_O_6_	5						
**133**	C_17_H_22_O_5_	7						
**134**	C_17_H_22_O_5_	7						
**135**	C_11_H_12_O_5_	6	pale yellow powder	*Penicillium chrysogenum* MT-12	*Huperzia serrata* (Thunb. ex Murray) Trev.	Fujian Province, China	Exhibit inhibition of nitric oxide production in lipopolysaccharide (LPS)-stimulated RAW264.7 macrophage cells (138,139,142,143,145,146)	[45]
**136**	C_9_H_8_O_5_	6						
**137**	C_10_H_10_O_5_	6	yellow powder					
**138**	C_15_H_20_O_6_	6						
**139**	C_14_H_16_O_6_	7						
**140**	C_15_H_16_O_6_	8	pale yellow powder					
**141**	C_15_H_18_O_6_	7						
**142**	C_15_H_20_O_6_	6						
**143**	C_16_H_22_O_7_	6	yellow powder					
**144**	C_15_H_20_O_5_	6						
**145**	C_15_H_20_O_5_	6						
**146**	C_15_H_20_O_5_	6						
**147**	C_14_H_12_O_6_	9	yellow powder	*Cylindrocarpon* sp.	fresh roots of *Sapium ellipticum*	Haut Plateaux region, Cameroon		[46]
**148**	C_14_H_12_O_7_	9						
**149**	C_13_H_10_O_7_	9						
**150**	C_14_H_12_O_6_	9	yellow crystals	*Phomopsis* sp. xy21	leaves of the Thai mangrove *Xylocarpus granatum*	Trang Province, Thailand	Weak anti-HIV activity (150)	[47]
**151**	C_15_H_16_O_7_	8	colorless crystals					
**152**	C_15_H_16_O_7_	8	White amorphous solid					
**153**	C_15_H_16_O_5_	8						
**154**	C_15_H_12_O_6_	10						
**155**	C_15_H_10_O_7_	11						
**156**	C_17_H_28_O_3_	4	white powder	*Aspergillus flocculus*	stem of the medicinal plant *Markhamia platycalyx*			[48]
**157**	C_10_H_10_O_4_	6	colorless crystals	*Colletotrichum gloeosporioides*	mangrove *Ceriops tagal*	Hainan Province China	Show potent antibacterial activity (157,159)	[49]
**158**	C_10_H_14_O_4_	4	brown oil					
**159**	C_10_H_12_O_3_	5	white powder					
**160**	C_14_H_18_O_4_	6	amorphous powder	*Paraconiothyrium* sp. SW-B-1	the seaweed, *Chondrus ocellatus* Holmes	Yamagata Prefecture, Japan	Show moderate antibacterial activity (164)	[50]
**161**	C_14_H_18_O_5_	6						
**162**	C_14_H_16_O_6_	7						
**163**	C_12_H_16_O_6_	5						
**164**	C_22_H_20_O_4_	13						
**165**	C_14_H_16_O_6_	7	pale brown, amorphous powder	*Alternaria alternata* MT-47	medicinal plant of *Huperzia serrata*	Fujian Province, China	Exhibit inhibitory activity on the ATP release of thrombin-activated platelets (168)	[51]
**166**	C_15_H_12_O_8_	10	pale yellow amorphous powder					
**167**	C_18_H_18_O_9_	10	white amorphous powder					
**168**	C_18_H_20_O_9_	9						
**169**	C_10_H_11_NO_4_	6	white gum	*Chaetosphaeronema achilleae*	shoots	English Yew (Taxus baccata), Iran	Weak antifungal activity and antibacterial activity (170) Cytotoxicity (169,170) Biofilm formation (169)	[52]
**170**	C_10_H_10_O_5_	6						
**171**	C_27_H_38_O_6_	9	colorless oil	*Aspergillus porosus*	algal			[53]
**172**	C_27_H_38_O_6_	9						
**173**	C_26_H_36_O_6_	9						
**174**	C_26_H_36_O_6_	9						
**175**	C_25_H_38_O_3_	7	colorless oil	*Alternaria alternate*	leaves of *Psidium littorale* Raddi	Fujian Province, China		[26]
**176**	C_29_H_30_O_10_	15	amorphous powder	*Phoma* sp. SYSU-SK-7	healthy branch of the marine *Kandelia candel*	Guangxi Province, China	Show strong antibacterial activity (176) Exhibit significant antifungal and antibacterial activity (177) Show significant α-glucosidase inhibitory activity (176–178) Cytotoxicity (176) Exhibit radical scavenging activity against DPPH (179)	[54]
**177**	C_11_H_14_O_4_	5	white solid					
**178**	C_21_H_24_O_7_	10						
**179**	C_13_H_12_O_5_	8						
**180**	C_11_H_16_O_3_	4	colourless oil					
**181**	C_10_H_14_O_3_	4		*Phomopsis* sp. D15a2a	leaves of *Alternanthera bettzickiana* (Amaranthaceae)	Anambra state of Nigeria		[55]
**182**	C_11_H_16_O_4_	4						
**183**	C_11_H_16_O_4_	4						
**184**	C_23_H_26_O_7_	11		*Penicillium purpurogenum* IMM003	fresh healthy leaves of *Edgeworthia chrysantha*	Zhejiang Province, China		[56]
**185**	C_22_H_26_O_6_	10						
**186**	C_10_H_8_O_5_	7						
**187**	C_18_H_27_NO_4_	6	colorless gum	*Camporesia sambuci* FT1061 & *Epicoccum sorghinum* FT1062	healthy fruit of the plant *Rhodomyrtus tomentosa*	the Big Island in Hawaii		[57]
**188**	C_14_H_20_O_6_	5	light yellow solid	*Rhytismataceae* sp. DAOMC 251461	healthy *P. mariana* needles	New Brunswick, Canada.	Exhibit moderate antifungal activity (189)	[58]
**189**	C_15_H_22_O_6_	5						
**190**	C_9_H_12_O_6_	4	colorless plate	*Phomopsis asparagi* SWUKJ5.2020	fresh, healthy branches of medicinal plant *Kadsura angustifolia*	Yunnan province China	Exhibit notable cytotoxicity (192–194)	[59]
**191**	C_9_H_12_O_5_	4						
**192**	C_9_H_10_O_6_	5	colorless crystals					
**193**	C_11_H_14_O_4_	5	colorless plates					
**194**	C_11_H_14_O_5_	5						
**195**	C_14_H_18_O_4_	6	colorless oil	*Dendrothyrium variisporum*	roots of the Algerian plant *Globularia alypum*	Ain Touta, Batna 05000 (Algeria)		[60]
**196**	C_18_H_24_O_5_	7						
**197**	C_11_H_10_O_5_	7	colorless oil	*Alternaria* sp.	twigs of *Morinda offificinalis*	Guangdong province China		[61]
**198**	C_12_H_11_O_5_	8	yellow oil					
**199**	C_8_H_12_O_3_	3	colorless gum	*Trichoderma atroviride*	bulb of *Lycoris radiata*	Hubei Province China		[11]
**200**	C_13_H_14_O_5_	7	yellow viscous liquid	*Eurotium chevalieri* KUFA 0006	healthy twig of *Rhizophora mucronata* Poir	Chanthaburi Province, Eastern Thailand	Prevent biofilm formation	[62]
**201**	C_9_H_14_O_4_	3	colorless gum	*Simplicillium* sp. PSU-H41	leaf of *Hevea brasiliensis*	Songkhla Province Thailand		[63]
**202**	C_15_H_12_O_8_	10	yellowish crystal	*Cytospora rhizophorae*	*Morinda offificinalis*	Guangdong province China	Exhibit weak growth inhibitory activity against the tumor cell lines (202)	[64]
**203**	C_14_H_10_O_6_	10	brown gum					
**204**	C_14_H_8_O_7_	11	yellowish green powder					
**205**	C_13_H_16_O_5_	6	yellow gum	*Fusarium* sp. HP-2	Chinese agarwood “Qi-Nan”	Hainan Province China	Show weak acetylcholinesterase inhibitory activity (207)	[65]
**206**	C_14_H_14_O_4_	8	red crystals					
**207**	C_16_H_18_O_6_	8	red solid					
**208**	C_18_H_17_ClO_7_	10	yellowish powder	*Penicillium citrinum* HL-5126	mangrove *Bruguiera sexangula var. rhynchopetala*	South China Sea	Display cytotoxic activity (209) Show weak antibacterial activity (208)	[66]
**209**	C_18_H_18_O_7_	10						
**210**	C_13_H_16_O_5_	6	amorphous white powder	*Phoma* sp. PF2	*Artemisia princeps*		Show moderate inhibitory activities on nitric oxide levels (210–211)	[67]
**211**	C_14_H_18_O_5_	6						
**212**	C_25_H_26_O_5_	13	polar yellow solid	*Aspergillus* sp. ASCLA	healthy leaf tissue of the medicinal plant *Callistemon subulatus*		Exert moderate-high activities against *Staphylococcus aureus*	[68]
**213**	C_12_H_18_O_3_	4	white powder	*Cylindrocarpon* sp.	fresh roots of *Sapium ellipticum*	Haut Plateaux region, Cameroon		[46]
**214**	C_25_H_28_O_6_	12	yellow oil	*Diaporthe lithocarpus* A740	twigs of medicinal plant *Morinda officinalis*.	Guangdong province China		[41]
Alkaloids and their derivatives								
**215**	C_26_H_33_O_8_N	11		*Apiospora montagnei*	lichen *Cladonia* sp.			[37]
**216**	C_16_H_19_NO_3_	8	colorless amorphous solid	*Chaetomium globosum* CDW7				[69]
**217**	C_17_H_24_N_2_O_3_	7	colorless crystals	*Penicillium citrinum* HL-5126	mangrove *Bruguiera sexangula var. rhynchopetala*	South China Sea		[66]
**218**	C_13_H_15_NO_2_	7	colorless powder	*Bionectria* sp.	seeds of the tropical plant *Raphia taedigera*	Haut Plateaux region, Cameroon		[70]
**219**	C_16_H_15_NO_5_	10	yellow powder	*Cylindrocarpon* sp.	fresh roots of *Sapium ellipticum*	Haut Plateaux region, Cameroon		[46]
**220**	C_14_H_21_NO_5_	5	white powder					
**221**	C_26_H_29_NO_6_	13	pale yellow amorphous solid	*Aspergillus versicolor*	leaves of the Egyptian water hyacinth *Eichhornia crassipes*	Egypt	Exhibit moderate antiproliferative activity	[71]
**222**	C_17_H_15_NO_8_	11	white amorphous solid	*Pestalotiopsis flavidula*	branches of *Cinnamomum camphora*	Yunnan province china	Moderate cytotoxicity (222–223)	[72]
**223**								
**224**	C_19_H_24_N_2_O_2_	9	white amorphous powder	*Irpex lacteus*-A	medicinal plant *Huperzia serrata*	Fujian Province China	Show moderate neuroprotective activity (224–225)	[73]
**225**	C_19_H_24_N_2_O_2_	9						
**226**	C_14_H_17_NO_3_	7	colorless solid	*Alternaria alternate*	leaves of *Psidium littorale* Raddi	Fujian Province, China		[26]
**227**	C_27_H_31_N_3_O_5_	14	brilliant yellowish oil	*Fusarium sambucinum* TE-6L	fresh leaves of cultivated tobacco (*N. tabacum* L.). *N. tabacum* L.	Hubei province China	Show potent inhibitory effects (227–228) Exhibit remarkable larvicidal activity (228)	[74]
**228**	C_27_H_31_N_3_O_5_	14	white solid					
Penylpropanoids and their derivatives								
**229**	C_10_H_14_O_5_	4	clear solid	*Mycosphaerellaceae* sp. DAOMC 250863	healthy needles from *Picea rubens* (red spruce) and *P. mariana* (black spruce)	Eastern Canada	Show modest antibiotic activity to *E. coli*	[58]
**230**	C_16_H_18_O_4_	8	light-yellow powder	*C. globosum* CDW7	*Ginkgo biloba*	China	Show moderate antifungal activity	[69]
**231**	C_12_H_14_O_4_	6	colorless amorphous solid	*Pestalotiopsis* sp. HHL-101	fresh twigs of the mangrove plant *Rhizophora stylosa*	Hainan Island, China	Exhibit moderate antibacterial activity	[75]
**232**	C_12_H_12_O_4_	7	white amorphous powder	*Nectria pseudotrichia* 120–1NP	healthy stem of *Gliricidia sepium*	Yogyakarta, Indonesia		[31]
**233**	C_13_H_14_O_4_	7						
**234**	C_21_H_12_O_12_	16	off-white amorphous solid	*Aspergillus versicolor*	leaves of the Egyptian water hyacinth *Eichhornia crassipes*	Egypt		[71]
**235**	C_22_H_14_O_12_	16	yellowish amorphous powder					
**236**	C_21_H_22_O_6_	11	colorless crystals	*Pestalotiopsis adusta*	stem bark of wild rare medicinal plant *Sinopodophyllum hexandrum* (Royle) Ying	Qinling Mountains China	Show weak to moderate cytotoxic activity	[19]
**237**	C_13_H_14_O_7_	7	white solid powder	*T. harzianum* Fes1712	Rubber Tree *Ficus elastica* Leaves	China	Exhibit inhibitory activity against Gram-negative bacteria (237–238)	[76]
**238**								
**239**	C_11_H_12_O_6_	6	white amorphous powder	*Penicillium coffeae* MA-314	fresh inner tissue of the leaf of marine mangrove plant *Laguncularia racemosa*	Hainan island, China		[77]
**240**								
**241**	C_18_H_22_O_3_	8	yellow oil	*Diaporthe* sp.	branches of *Pteroceltis tatarinowii* Maxim	Nanjing province, China	Show modest antibacterial activity Weak cytotoxicity	[78]
Lactones								
**242**	C_11_H_10_O_5_	7	yellowish brown solid	*Alternaria* sp.	seeds of the plant *Ziziphus jujuba*	Uzbekistan		[79]
**243**	C_16_H_26_O_6_	4	white, amorphous powder	*Phaeoacremonium* sp.	leaves of *Senna spectabilis*	Araraquara Cerrado area, Sao Paulo state, Brazil.	Exhibit antifungal activity (244–245) Cytotoxicity (244)	[80]
**244**	C_16_H_26_O_5_	4						
**245**								
**246**	C_9_H_12_O_2_	4	amorphous powder	*Xylaria curta* 92092022	barks	Taiwan China	Show moderate antibacterial and phytotoxic activities	[81]
**247**	C_16_H_22_O_5_	6	white powder	*Trichoderma* sp. 307 & *Acinetobacter johnsonii* B2	Strain 307, stem bark of *Clerodendrum inerme*	Guangdong Province, China	Exhibit potent α-glucosidase inhibitory activity (247–248) show moderate inhibitory activity against α-glucosidase (249)	[10]
**248**	C_16_H_20_O_5_	7						
**249**			colorless needles					
**250**	C_10_H_16_O_3_	3	colorless oil	*Pestalotiopsis* sp.	fruits of *Drepanocarpus lunatus* (Fabaceae)			[12]
**251**	C_13_H_18_O_5_	5						
**252**	C_11_H_14_O_5_	5	colorless crystals	*Talaromyces* sp.	*Xanthoparmelia angustiphylla*	Stockholm, Sweden	Exhibit selective cytotoxicities	[82]
**253**	C_32_H_50_O_7_	8	yellow powder	Mutant CS/*asm*21-4	*Maytenus hookeri*	China	Exhibit antibacterial activity	[83]
**254**	C_22_H_21_NO_4_	13	light yellow gum	*Aspergillus terreus*	Yongxing Island fresh, healthy leaves of S. *maritima* L.	South China Sea, China	Show strong inhibitory effects on the production of NO (256–257)	[84]
**255**	C_22_H_19_O_4_	14						
**256**	C_22_H_21_NO_5_	13						
**257**	C_22_H_21_NO_5_	13						
**258**	C_20_H_22_O_3_	10	yellow oil	*Aspergillus* sp.	root of *Tripterygium wilfordii*	Wuhan, China	Exhibited weak AchE and BACE1 inhibitory activity (260–261) Showed excellent inhibitory effects on the production of IL-1β, TNF-α, and NO (258–261)	[85]
**259**	C_24_H_26_O_6_	12	yellow oil					
**260**	C_24_H_26_O_6_	12	colorless oil					
**261**	C_23_H_26_O_6_	11						
**262**	C_22_H_36_O_8_	5	oil	*H. fuscum*	lichen *Usnea* sp.	Yunnan, China	Exhibit moderate cytotoxicity	[86]
**263**	C_26_H_34_O_12_	10	white powder	*Talaromyces purpurogenus*	fresh leaves of the toxic medicinal plant *Tylophora ovata*	China		[87]
**264**	C_28_H_36_O_12_	11						
**265**	C_26_H_40_O_9_	7						
**266**	C_11_H_18_O_3_	3	yellow oil	*Penicillium coffeae* MA-314	fresh inner tissue of the leaf of marine mangrove plant *Laguncularia racemosa*	Hainan island, China	Exhibit potent antifungal activity	[77]
**267**	C_12_H_12_O_5_	7	brown solids	*Phomopsis* sp.	stems of *Isodon eriocalyx* var. *laxiflflora*	Kunming, China		[43]
**268**	C_17_H_14_O_3_	11	white amorphous powder	*Phyllosticta* sp. J13-2-12Y	leaves of *Acorus tatarinowii*	Guangxi Province, China		[88]
**269**								
**270**	C_19_H_16_O_5_	12	colorless oil					
**271**	C_16_H_12_O_3_	11	colorless crystal					
**272**	C_22_H_26_O_6_	10	luminous yellow oil	*Pestalotiopsis microspora*	fruits of *Manilkara zapota*	Kandy, Sri Lanka		[89]
Anthraquinones								
**273**	C_12_H_14_O_5_	6	yellow amorphous powder.	*Penicillium citrinum* Salicorn 46	*Salicornia herbacea* Torr.	China		[90]
**274**	C_14_H_14_O_4_Cl_2_	7	yellow oil	*Lachnum* cf. *pygmaeum* DAOMC 250335	dead *P. rubens* twig	NB, Canada	Inhibit the growth of *M. violaceum,*	[58]
**275**	C_16_H_12_O_6_	11		*Apiospora montagnei*	lichen *Cladonia* sp.			[37]
**276**	C_18_H_14_O_7_	12	yellow crystal	*Eurotium chevalieri* KUFA 0006	healthy twig of *Rhizophora mucronata* Poir.	Chanthaburi Province, Eastern Thailand	Cause a significant reduction in biofilm production	[62]
**277**	C_15_H_16_O_3_	8		*Nigrospora oryzae* co-cultured with *Irpex lacteus*	seeds of *Dendrobium offifficinale*	Yunnan Province China		[14]
**278**	C_15_H_18_O_2_	7						
**279**	C_15_H_20_O_4_	6						
**280**	C_15_H_20_O_6_	6						
**281**	C_14_H_16_O_2_	7		*Phoma betae*	*Kalidium foliatum* (Pall.)	China	Cytotoxic activities (281)	[91]
**282**	C_14_H_16_O_3_	7						
**283**	C_14_H_20_O_5_	5						
**284**	C_27_H_24_O_10_	16	red powder	*Neofusicoccum austral* SYSU-SKS024	branches of the mangrove plant *Kandelia candel*	Guangxi province, China	Show inhibitory effects against Indoleamine 2,3-dioxygenase (IDO)	[92]
**285**	C_15_H_16_O_6_	8	yellow powder					
**286**	C_14_H_18_O_5_	6	white powder					
**287**	C_16_H_18_O_5_	8	yellow amorphous powder	*Nectria pseudotrichia* 120-1NP	healthy stem of *Gliricidia sepium*	Yogyakarta, Indonesia	Exhibit antibacterial activity Exhibit cytotoxicity	[31]
**288**	C_30_H_22_O_12_	20	yellow powder	ARL-09 (*Diaporthe* sp.)	*Anoectochilus roxburghii*	China	Cytotoxicity Effects on NF-κB signaling pathway	[93]
**289**	C_40_H_45_NO_10_S	19	red powder	CS/*asm*21-4	callus of Chinese medicinal plant *Maytenus hookeri*	China	Show moderate antimicrobial activities (antibacterial activities and antifungal activity) (289–291)	[83]
**290**	C_40_H_49_NO_12_	17	yellow powder					
**291**	C_40_H_44_NO_8_Cl	19						
**292**	C_12_H_18_O_6_	4	colorless oil	*Xylaria* sp. SYPF 8246	root of *Panax notoginseng*	Yunnan, China		[94]
**293**	C_15_H_14_O_6_	9		*Talaromyces funiculosus*	lichen thallus of *Diorygma hieroglyphicum*	India	Display antimicrobial activity	[95]
**294**	C_16_H_14_O_7_	10	yellow gum	*Simplicillium lanosoniveum* Zare & W. Gams PSU-H168 and PSU-H261	leaves of *Hevea brasiliensis*	Songkhla Province, Thailand	Display antifungal activity	[96]
**295**	C_17_H_18_O_7_	9	red amorphous powder	*Fusarium napiforme*	mangrove plant, *Rhizophora mucronata*	Makassar, Indonesia	Exhibit moderate antibacterial activity (295–296) Phytotoxic (295–296)	[97]
**296**	C_16_H_16_O_6_	9	orange amorphous powder					
Sterides								
**297**	C_34_H_52_O_8_	9	faint yellow oil	*Xylaria* sp.	leaves of *Panax notoginseng*	Yunnan province China	Show cytotoxicity (297)	[98]
**298**	C_28_H_44_O_7_	7	semitransparent oil					
**299**	C_25_H_36_O_5_	8	colorless needle	*Chaetomium* sp. M453	Chinese herbal medicine *Huperzia serrata*	Yunnan Province, China	Show weak acetylcholinesterase inhibitory activity (302)	[99]
**300**	C_25_H_36_O_5_	8	colorless amorphism					
**301**	C_25_H_34_O_5_	9						
**302**	C_28_H_42_O_3_	8	yellow oil					
**303**	C_22_H_32_O_3_	7	colorless crystals	*Stemphylium* sp. AZGP4–2	root of *Polyalthia laui*	Hainan Province China	Show antibacterial activity against *Escherichia coli* (303) Exhibit antibacterial activity (304)	[100]
**304**	C_23_H_36_O_3_	6						
**305**	C_23_H_34_O_3_	7	colorless needle crystals					
**306**	C_44_H_72_O_2_	9	white amorphous powder	*Fusarium* sp.	*Mentha longifolia* L. (Labiatae) roots	Saudi Arabia	Possessed 5-LOX inhibitory potential (306–308)	[101]
**307**	C_28_H_46_O_3_	6						
**308**	C_30_H_48_O_5_	7						
**309**	C_28_H_40_O_2_	9	colorless powder	*Pleosporales* sp. F46 and *Bacillus wiedmannii*. Com1	medicinal plant *Mahonia fortunei*	Qingdao, China.	Exhibit moderate antibacterial efficacy	[102]
**310**	C_32_H_41_NO_3_	13	white power	*Aspergillustubingensis* YP-2	bark of *Taxus yunnanensis*	Yunnan Province, China	Show weak cytotoxicities (311)	[103]
**311**	C_22_H_34_O_3_	6						
Other types of compounds								
**312**	C_21_H_24_O_6_	10	colorless oil	*Talaromyces stipitatus* SK-4	leaves of a mangrove plant *Acanthus ilicifolius*	Guangxi Province, China	Show antibacterial activity and inhibitory against α-glucosidase (313)	[104]
**313**	C_23_H_26_O_7_	11						
**314**	C_15_H_21_NO_8_	6	whitish needles	C. *ninchukispora* BCRC 31900	seeds of medicinal plant *Beilschmiedia erythrophloia* Hayata	Taiwan China	Show anti-inflammatory effects through inhibition of NO production (317,314–315)	[105]
**315**	C_15_H_21_NO_7_	6						
**316**	C_16_H_23_NO_7_	6						
**317**	C_15_H_21_NO_8_	6	yellowish solid					
**318**	C_15_H_16_O_5_	8	white amorphous powder	*Pyronema* sp. (A2-1 & D1-2)	*Taxus mairei*	Hubei province, China	Exhibit moderate antibiotic activity	[106]
**319**	C_11_H_16_O_4_	4	yellow oil	*Phoma* sp. nov. LG0217	branches of *Parkinsonia microphylla*	Tucson, Arizona		[107]
**320**	C_12_H_16_O_4_	5	colorless amorphous powder	*Penicillium citrinum* Salicorn 46	*Salicornia herbacea* Torr	China	Exhibit potent cytotoxic activity	[90]
**321**	C_21_H_29_NO_9_	8	colorless gum	*Phomopsis* sp. PSU-H188	midrib of *Hevea brasiliensis*	Trang Province, Thailand		[108]
**322**	C_20_H_28_O_7_	7						
**323**	C_21_H_27_O_6_N	9	yellow amorphous solid	*Fusarium solani* JK10	root of the Ghanaian medicinal plant *Chlorophora regia*	Eastern Region of Ghana	Exhibit antibacterial efficacies (325–326,328)	[109]
**324**								
**325**	C_21_H_27_O_7_N	9						
**326**								
**327**	C_21_H_25_O_8_N	10						
**328**	C_22_H_29_O_7_N	9	pale yellow amorphous solid					
**329**	C_22_H_29_O_5_N	9	yellow amorphous solid					
**330**	C_14_H_14_O_4_	8	colourless oil	*Phomopsis longicolla* HL-2232	fresh healthy leaf of *Brguiera sexangula var. rhynchopetal*a	South China Sea	Show moderate antibacterial activities	[110]
**331**	C_9_H_16_O_4_	2	white needles	*Penicillium* sp. OC-4	leaves of *Orchidantha chinensis*	Guangdong Province, China	Display strong antioxidant activity	[111]
**332**	C_16_H_24_O_6_	5	colorless, amorphous solid	*Curvularia* sp.	leaf of the medicinal plant *Murraya koenigii*	Bangladesh	Exhibit zoospore motility impairment activity (333–334)	[112]
**333**	C_12_H_18_O_6_	4						
**334**	C_10_H_12_O_3_	5	colorless crystals					
**335**	C_10_H_16_O_4_	3	colorless oil					
**336**	C_20_H_16_O_5_	13	yellow viscous oil	*Rhytidhysteron rufulum* AS21B	leaves of *Azima armentosa*	Samutsakhon province, Thailand	Display the most promising anti-tumor activity (337)	[113]
**337**	C_22_H_18_O_5_	14	pale yellow gum					
**338**	C_11_H_12_O_4_	6	brown solids	*Phomopsis* sp. sh917	stems of *Isodon eriocalyx* var. *laxiflora*	Kunming, China		[43]
**339**	C_15_H_15_NO_3_	9	brown gum	*Dendrothyrium variisporum*	roots of the Algerian plant *Globularia alypum*	Algeria	Show the strongest activity against *Bacillus subtilis* and *Micrococcus luteus* (339)	[60]
**340**	C_14_H_13_NO_2_	9						
**341**	C_12_H_17_NO_3_	5						
**342**	C_15_H_18_N_2_O_4_	8	light yellow gum	*Trichoderma atroviride*	bulb of *Lycoris radiata*	china		[11]
**343**	C_32_H_34_N_2_O_4_	17	yellow crystal.	*Penicillium chrysogenum* V11	vein of *Myoporum bontioides* A. Gray	Leizhou Peninsula, China	Display significant antifungal activity and remarkable cytotoxicities	[114]
**344**	C_14_H_15_NO	8	yellow crystal	*Eurotium chevalieri* KUFA 0006	healthy twig of *Rhizophora mucronata* Poir.	Chanthaburi Province, Eastern Thailand	Show inhibition of biofilm production (344–345)	[62]
**345**	C_14_H_15_NO	8	yellowish viscous liquid					
**346**	C_13_H_15_NO_3_	7						
**347**	C_18_H_18_O_6_	10	colorless solid	*Simplicillium* sp. PSU-H41	leaf of *Hevea brasiliensis* (Euphorbiaceae)	Songkhla, Thailand	Display weak antibacterial against *Staphylococcus aureus* (347) Exhibit weak antifungal activity against *Cryptococcus neoformans* (349)	[63]
**348**	C_19_H_20_O_6_	10	pale yellow solid					
**349**	C_20_H_20_O_6_	11						
**350**	C_25_H_24_O_7_	14						
**351**								
**352**	C_25_H_22_O_8_	15	yellow gum					
**353**	C_24_H_26_O_7_	12	pale yellow gum					
**354**	C_34_H_30_O_11_	20	colorless solid					
**355**	C_31_H_28_O_8_	18	pale yellow gum					
**356**	C_17_H_24_N_2_O_6_	7	colorless viscous oil	*Phoma herbarum* PSU-H256	leaf of *Hevea brasiliensis*	Songkhla, Thailand		[115]
**357**	C_12_H_13_NO_6_	7						
**358**	C_16_H_19_NO_7_	8						
**359**	C_15_H_17_NO_5_	8						
**360**	C_7_H_12_N_2_O_3_	3						
**361**	C_14_H_14_N_2_O_5_	9						
**362**	C_11_H_12_O_3_	6	white amorphous solid.	*Penicillium* sp.	leaf of *Senecio flavus* (Asteraceae)	Al-Azhar University Egypt	Show antifungal activity and cytotoxic activity	[116]
**363**	C_30_H_37_NO_7_	13	white amorphous powder	*R. sanctae-cruciana*	leaves of the medicinal plant *A. lebbeck*.	India	Show considerable cytotoxic potential	[117]
**364**	C_24_H_30_O_4_	10	yellowish oil	*Arthrinium arundinis* TE-3	fresh leaves of cultivated tobacco	Hubei Province China	Show selective antifungal activity (364–365) Display moderate in vitro cytotoxicity (365)	[118]
**365**	C_20_H_24_O_4_	9						
**366**	C_20_H_24_O_3_	9						
**367**	C_23_H_24_O_5_	12	brown powder	*Aspergillus flavipes* Y-62	stems of plant *Suaeda glauca* (Bunge) Bunge	Zhejiang province, East China	Show weak antimicrobial activity	[119]
**368**	C_16_H_14_O_6_	9	colorless crystals	*Mycosphaerella* sp. (UFMGCB2032)	healthy leaves of *Eugenia bimarginata*	Atlanta, GA, USA	Exhibit moderate antifungal activities	[120]
**369**	C_17_H_18_O_9_	9	colorless solid					
**370**	C_20_H_16_O_5_	13	off-white gum	*Anteaglonium* sp. FL0768	Living photosynthetic tissue of sand spikemoss (*Selaginella arenicola*; Selaginellaceae)			[121]
**371**	C_28_H_26_N_2_O_5_	17	amorphous light yellow powder	*Penicillium janthinellum* SYPF 7899	three-year-old healthy *P. notoginseng*	Yunnan province, China	Exhibit significant inhibitory activities (371–373)	[122]
**372**								
**373**	C_15_H_19_NO_6_	7	brown oil					
**374**	C_14_H_24_O_4_	3	colorless oil	*Phaeophleospora vochysiae* sp. nov	*Vochysia divergens*	wetland in Brazil	Show considerable antimicrobial activity	[123]
**375**	C_12_H_17_NO_6_	5	colorless oil	*Bionectria* sp.	fresh seeds of *R. teadigera*	Haut Plateaux region, Cameroon		[70]
**376**	C_18_H_14_N_2_O_6_	13	white powder					
**377**	C_13_H_19_NO_4_	5	yellowish oil	*Trichoderma atroviride* S361	bark of *Cephalotaxus fortunei*	Zhejiang province, China		[17]
**378**								
**379**	C_18_H_20_O_7_	9	amorphous powder	*Xylaria* sp. SYPF 8246	root of *Panax notoginseng*	Wenshan, Yunnan, China	Display significant inhibitory activities against human carboxylesterase 2 (hCE 2) (379,383–385)	[94]
**380**	C_12_H_10_O_5_	8	colorless oil					
**381**	C_12_H_18_O_6_	4						
**382**	C_12_H_20_O_5_	3						
**383**	C_19_H_22_O_7_	9						
**384**	C_19_H_21_O_7_Cl	9						
**385**	C_18_H_19_O_7_Cl	9						
**386**	C_32_H_42_O_4_	12	brown oil	*Byssochlamys spectabilis*	leaf tissue of the medicinal plant *Edgeworthia chrysantha*	Zhejiang Province, China	weakly active against *Escherichia coli* and *Staphyloccocus aureus* (388) Display selective inhibitory effects toward hCE2-mediated FD hydrolysis (386)	[124]
**387**	C_16_H_22_O_3_	7	yellow oil					
**388**	C_16_H_26_O_2_	5						
**389**	C_20_H_29_N_5_O_6_	9	white amorphous powder	*Fusarium chlamydosporium*	*Anvillea garcinii* (Burm.f.) DC. leaves	Egypt	Exhibit selective antifungal activity and cytotoxic effect possess high antibacterial potential	[125]
**390**	C_15_H_16_N_2_O_2_	9		*Annulohypoxylon stygium*	red seaweed *Bostrychia radicans*	Ubatuba city, São Paulo State, Brazil		[126]
**391**	C_23_H_16_O_2_N_2_	17	purple-red powder	*Alternaria alternata* Shm-1	fresh wild body of *Phellinus igniarius*	Shanxi Province, China		[127]
**392**	C_10_H_12_O_6_	5	colorless crystals	*Cladosporium* sp. JS1–2	mangrove *Ceriops tagal*	Hainan Province in China	Show moderate antibacterial activities (392–393) Showed growth inhibition activities against newly hatched larvae of H. armigera Hubner (392–393)	[128]
**393**	C_10_H_14_N_2_O_2_	5	yellow powder					
**394**	C_8_H_13_NO_4_	3	white solid	*Diaporthe vochysiae* sp. nov. (LGMF1583)	medicinal plant *Vochysia divergens*		Display considerable antibacterial activity (395) Show low to moderate cytotoxic activity (394–395)	[129]
**395**	C_11_H_17_NO_4_	4	white solid					
**396**	C_28_H_40_O_6_	9	yellow oil	*Diaporthe lithocarpus* A740	Twigs of medicinal plant *Morinda officinalis*	Guangdong province , China	Show weak cytotoxic activity (396–397)	[41]
**397**	C_28_H_40_O_6_	9						
**398**	C_30_H_37_O_7_N	13	colorless powder	*Xylaria longipes*		Ailao Moutain		[130]
**399**	C_30_H_39_O_9_N	12						
**400**	C_32_H_41_O_8_N	13						
**401**								
**402**	C_30_H_37_NO_7_	13						
**403**	C_18_H_18_O_7_	10		*Penicillium citrinum*	*Parmotrema* sp.	Hakgala montane forest in Sri Lanka	Show moderate antioxidant activity	[131]
**404**	C_11_H_11_ClO_5_	6		*Periconia macrospinosa* KT3863	a terrestrial herbaceous plant	Kanagawa prefecture, Japan		[132]
**405**	C_12_H_13_ClO_4_	6						
**406**	C_7_H_12_O_3_	2	light yellow liquid	*Lasiosdiplodia pseudotheobromae*			Exhibite XO inhibition (407) oxidized form of 406 show high XO inhibition	[133]
**407**	C_13_H_22_O_3_	3						
**408**	C_17_H_16_O_8_	10	pale-yellow needles	*Pleosporales* sp. SK7	leaves of the mangrove plant *Kandelia candel*	Guangxi Province, China		[23]
**409**	C_15_H_19_N_2_O_2_	8	faint yellow oil	*Aspergillus* sp. AV-2	inner healthy leaves of mangrove plant *Avicennia marina*	Hurghada, Egypt		[134]
**410**	C_19_H_22_O_5_	9	yellow powder					
**411**	C_10_H_14_O_3_	4	yellowish oil	*Irpex lacteus* DR10-1	Roots of waterlogging tolerant plant *Distylium chinense*	Chongqing in the TGR area, China	Exhibit strong antioxidant activity (413) Show moderate antibacterial activity (411–413)	[24]
**412**	C_10_H_14_O_3_	4						
**413**	C_12_H_16_O_4_	5	brown flaky solid					
**414**	C_33_H_50_O_6_	9	pale yellow oil	*Penicillium crustosum* PRB-2 & *Xylaria* sp. HDN13-249	*Xylaria* sp. HDN13-249: root of *Sonneratia caseolaris*	Hainan province, China	Show antibacterial activity (415–416) Show promising activity against *M. phlei* (416)	[135]
**415**	C_33_H_50_O_9_S	9						
**416**	C_24_H_40_O_5_	5	pale yellow oils	*Xylaria* sp. HDN13-249				
**417**	C_24_H_40_O_8_S	5						
**418**	C_9_H_14_O_2_	3	colorless oil	*Aspergillus terreus* EN-539 & *Paecilomyces lilacinus* EN-531	inner tissues of the marine red alga *Laurencia okamurai*	China	Exhibit inhibitory activity against bacteria and fungi	[136]
**419**	C_23_H_20_O_5_	14	white powder	*Diaporthe lithocarpus*	leaves of *Artocarpus heterophyllus*	Dortmund, Germany		[137]
**420**	C_16_H_20_N_2_O_4_	8	colourless oil	*Aspergillus aculeatus* F027	fresh leaves of *Ophiopogon japonicus* (Linn. f.) Ker-Gawl	Hubei province of China		[138]
**421**	C_17_H_20_O_6_	8	reddish oil	*Fusarium solani* B-18	inner tissue of the unidentifified forest litters	Mount Merapi area Sleman, Yogyakarta, Indonesia.	Activat a signaling pathway in osteoclastic differentiation of murine macrophage (421)	[139]
**422**	C_17_H_20_O_6_	8	yellow oil					
**423**	C_15_H_18_O_5_	7	reddish oil					
**424**	C_15_H_18_O_5_	7	pale-yellow oil					
**425**	C_16_H_20_O_5_	7	amorphous powder	*Hypoxylon fuscu*	lichen *Usnea* sp.	Lilong Snow Mountain in Lijiang, Yunnan, China	Exhibit moderate cytotoxicity (426–427)	[86]
**426**	C_21_H_36_O_6_	4	white solid					
**427**	C_18_H_30_O_7_	4	white powder					
**428**	C_18_H_28_O_6_	5						
**429**	C_25_H_24_O_6_	14	colorless gum	*Simplicillium lanosoniveum* (J.F.H. Beyma) Zare & W. Gams PSU-H168 and PSU-H261	leaves of *Hevea brasiliensis*	Songkhla Province, Thailand	Exhibit antibacterial activity (430) Display antifungal activity (430–431)	[96]
**430**	C_32_H_34_O_8_	16						
**431**	C_16_H_14_O_7_	10	yellow gum					
**432**	C_14_H_20_O_4_	5	white amorphous powder	*Clonostachys rosea* B5-2	mangrove plants	Garut, Indonesia	Exhibit phytotoxicity against lettuce seedlings (432)	[140]
**433**	C_7_H_10_O_3_	3	colourless oil					
**434**	C_9_H_12_O_3_	4	white amorphous powder					
**435**	C_9_H_14_O_4_	3						
**436**	C_26_H_32_O_12_	11	white powder	*Talaromyces purpurogenus*	fresh leaves of the toxic medicinal plant *Tylophora ovata*	Guangxi Province, China	Show significant inhibitory activity against NO production in LPS-induced RAW264.7 cells (436) Show moderate inhibitory activities toward XOD and PTP1b (437,441)	[87]
**437**	C_26_H_38_O_11_	8	white powder					
**438**	C_27_H_28_O_8_	14	white powders					
**439**	C_29_H_40_O_9_	10						
**440**	C_27_H_40_O_7_	8						
**441**	C_26_H_34_O_7_	10						
**442**	C_22_H_32_N_4_O_5_	9	white powder	*Phomopsis* sp. D15a2a	leaves of *Alternanthera bettzickiana* (Amaranthaceae)	Anambra state of Nigeria		[55]
**443**	C_8_H_13_NO_5_	3						
**444**	C_20_H_38_O_7_	2	colorless oil	*Aureobasidium pullulans* AJF1	flower of *Aconitum carmichaeli*,	Jangbaek Mountain, Gangwon-do, Korea		[141]
**445**	C_30_H_56_O_10_	3						
**446**	C_16_H_14_O_8_	10	yellow amorphous powder	*Alternaria alternata* JS0515	*Vitex rotundifolia* (beach vitex)	Suncheon, Korea		[142]
**447**	C_23_H_27_O_5_Cl	10	colorless oil	*Armillaria* sp. & *Epicoccum* sp. YUD17002	YUD17002: rhizomes of the underground portion of *Gastrodia elata*	Yunnan Province, China	Exhibit moderate in vitro cytotoxic activities (447) Show weak acetylcholinesterase Inhibitory activity (447)	[27]
**448**	C_10_H_10_O_4_	6	white amorphous powder					
**449**	C_14_H_20_O_9_	5	light-yellow oil					

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
