# Peer review of "Biological Activities of Some New Secondary Metabolites Isolated from Endophytic Fungi: A Review Study"

_ijms, 2021, doi:10.3390/ijms22020959_

Round 1

Reviewer 1 Report

From reading the text no major problems emerge, but I realize that my correction is only partial, because I am not able to verify the correctness of the chemical formulas; it is not, in fact, my field of study. So I entrust the final judgment to my other colleagues. Some corrections are noted in the conclusion.

However, I attach the file with my notes.

Author Response

1 I have revised the singular and plural, capitalization, grammar and other details of words.

2 In addition, I also reorganized the figures

3 “These compounds have a variety of unique structures, structural diversity decide the multiformity biological activities of compounds, especially some of those metabolites displayed significant antimicrobial effects, cytotoxic activities, antioxidant activities and other biological activities, which indicate that they have potential to be agents to treat some diseases.”

  I reorganized this sentence as “These compounds have a variety of unique structures, the difference in structure leads to different  biological activities of these compounds. Some of these metabolites display significant antimicrobial effects, cytotoxic activities, antioxidant activities and other biological activities, which indicate that they have potential to be agents to treat some diseases.”

4 I found you label the (fig2) many times, and it can not correspond to the Fig2. Thank you for pointing out the error. I corrected this problem.

Reviewer 2 Report

My revision has been made in the attached pdf file.

Please check all scientific names and their abbreviations.

Font and characters are not similar along the manuscript.

Some recent publications reported very interesting results of secondary metabolites ion general either from bacteria and fungi showed bioactivity against several phytopathogens , you should start your introduction  with some general information about the secondly metabolites in different microorganism then you can specialize only to fungi then to endophytic fungi.

I suggest to change the title. Such as: Biological activities of some new secondary metabolites isolated from endophytic fungi. Review study.

Author Response

1 First of all, thank you for your constructive comments. Because my whole article is a summary of the metabolites and biological activities produced by endophytes from plants (endophytes also exist in animals), so according to your enlighten, I have modified the introduction, and the revised content is as follows: starting from the introduction of the definition of endophytes, introducing its classes, biological activities and research significance, and then introducing a very important category: endophytic fungi.

2 using “Biological activities of some new secondary metabolites isolated from endophytic fungi: A review study” replace original title “A review: new metabolites with biological activities isolated from plant endophytic fungi”

 I did use a longer length to describe the biological activities. besides, The purpose of studying endophytic fungi is to obtain compounds with good biological activities, so it is necessary to focus on the biological activities

3 “New secondary metabolites” in keywords, i prefer to keep the word new to echo the title. I don’t know if the editor thinks it is feasible?

4 The names of the three fungi are enclosed in brackets in page 58 in order to make the whole sentence read more smoothly.

5 The error like font slanted you pointed out have been corrected.

6 All scientific names and their abbreviations in the review were checked and confirmed.

Reviewer 3 Report

Zheng et al. describe novel compounds isolated from endophytic fungi between 2017 and 2019. The work includes descriptions of the many compounds as well as their potential biological activities. The breadth of the review is its biggest strength. There are some issues with the form and message of the review. As it stands, the review is more of a collection of chemical structure descriptions as well as a table of biological activities; however, there is no main narrative to explain where the field has been, where the field is, and where the field is going. The following are more specific issues that I have with the manuscript.

The review includes statements that suggest structural elements are rare or surprising. For example, “It is of particular interest that 11 and 12 contain the epoxycyclohexenone moiety instead of the benzaldehyde group[10].” It’s unclear why this is of particular interest. Statements like these require elaboration and narrative.

Many statements are unclear or include grammatical errors which must be rewritten. The following are just a few examples.

“ Endophytes refer to the microorganisms that exist in all or part of their life cycle in various organs, tissues and intercellular Spaces of plants, and host plants generally do not show any symptoms of infection, including endophytic fungi, endophytic bacteria and endophytic actinomycetes[1].”

“Being a very important microbial resource, endophytes can be found in nature widely.”

“In recent years, the metabolites isolated from the fermentation products of endophytic fungus include alkaloids, steroids, terpenes, anthraquinones, cyclic peptides, flavonoids and so on [3].” Vague

“It is of particular interest that 11 and 12 contain the epoxycyclohexenone moiety instead of the benzaldehyde group[10].” Why is this of particular interest?

“Through this review, i hope to arouse more people's interest and attention in this field and create a better life for mankind by utilizing endophytic resources.”

Author Response

1 The definition of endophytes was re-described,The sentences have also been rearranged

2 “Through this review, i hope to arouse more people's interest and attention in this field and create a better life for mankind by utilizing endophytic resources.”

I explain how to create a better life for people

3 As requested by the editor, I have explained this sentence

“It is of particular interest that 11 and 12 contain the epoxycyclohexenone moiety instead of the benzaldehyde group[10].” Why is this of particular interest?

4 “however, there is no main narrative to explain where the field has been, where the field is, and where the field is going”

.   Regarding this revise opinion, I have added something in the conclusion. details as follows:

 “Since the endophyte resources were discovered, more and more researches have been conducted on them. Just from my review article, I have counted the new secondary metabolites isolated from plant endophytes during the three years from 2017 to 2019. Among them, 38 articles were published in 2017, 136 new compounds were obtained; 39 articles were published in 2018, 117 new compounds were obtained; 57 articles were published in 2019, and 196 new compounds were obtained. It can be discovered that in the past three years, the research trend of plant endophytes and their metabolites has increased year by year. The more new compounds obtained, the greater the possibility of screening compounds with excellent biological activity, This is also an important significance for researchers to study plant endophytes. ”

5“In recent years, the metabolites isolated from the fermentation products of endophytic fungus include alkaloids, steroids, terpenes, anthraquinones, cyclic peptides,flavonoids and so on "

 I have modified this sentence as “The multiformity of endophytes enable they can produce a variety of secondary metabolites. In recent years, the metabolites isolated from the endophytic fungi include alkaloids, steroids, terpenes, anthraquinones, cyclic peptides, flavonoids commonly”

Round 2

Reviewer 2 Report

 The authors revised well the manuscript.

Only two small minor revisions:

1- Add the following publication in introduction related to some metabolites produced by T. harzianum and these metabolites having biodegradation effect against some soil contaminants

"Elshafie H.S., Camele I., Sofo A., Mazzone G., Caivano M., Masi S. and Caniani D. 2020. Mycoremediation effect of Trichoderma harzianum strain T22 combined with ozonation in diesel-contaminated sand. Chemosphere 252, 126597. DOI: 10.1016/j.chemosphere.2020.126597."

2- Add one or two small phrases in the beginning of introduction related to some metabolites pruced in general from microorganisms not only fungi such as some important bacteria phytopathogenic, and you can use the following two references:

a- Elshafie H.S., Viggiani L., Mostafa M.S., El-Hashash M.A., Bufo S.A. and Camele I. 2017. Biological activity and chemical identification of ornithine lipid produced by Burkholderia gladioli agaricicola ICMP 11096 using LC-MS and NMR analyses. J. Biol. Res. 90: (6534), 96-103. DOI: 10.4081/jbr.2017.6534.

b- Camele I., Elshafie H.S., Caputo L., Sakr S.H. and De Feo V. 2019. Bacillus mojavensis: Biofilm formation and biochemical investigation of its bioactive metabolites. J. Biol. Res. 92: (8296), 39-45. DOI:4081/jbr.2019.8296.

Author Response

Thanks very much for your attention to our paper. I have revised the manuscript as your guidance, details as follows:

1 According to the suggestion of one of the reviewers, I checked the whole review article carefully, and all issues in the review have been revised, such as grammatical issues, initial capitalization issues, singular and plural number issues, special term italics issues, and punctuation issues, etc.

2 According to the suggestion of the other reviewers, i have added some description in the introduction to make the whole article more comprehensive. At the beginning, i added two small phrases to describe microorganisms and their secondary metabolites. In the middle part, I added a short paragraph to describe the protective effect of endophytes on the ecological environment by reference specifically.

At last, I want to thank you sincerely for your suggestions again and I feel so sorry that so much of your precious time was wasted on our paper revision.

Reviewer 3 Report

The document is improved as a reference or resource. There remain English language issues that must be corrected prior to publication.

Author Response

(The authors gave the same response as above.)
